# Efficient Clustering for Stretched Mixtures: Landscape and Optimality

**Kaizheng Wang**
Columbia University
`kaizheng.wang@columbia.edu`

**Yuling Yan**
Princeton University
`yulingy@princeton.edu`

**Mateo Díaz**
Cornell University
`md825@cornell.edu`

## Abstract

This paper considers a canonical clustering problem where one receives unlabeled samples drawn from a balanced mixture of two elliptical distributions and aims for a classifier to estimate the labels. Many popular methods including PCA and k-means require individual components of the mixture to be somewhat spherical, and perform poorly when they are stretched. To overcome this issue, we propose a non-convex program seeking for an affine transform to turn the data into a one-dimensional point cloud concentrating around $-1$ and $1$, after which clustering becomes easy. Our theoretical contributions are two-fold: (1) we show that the non-convex loss function exhibits desirable geometric properties when the sample size exceeds some constant multiple of the dimension, and (2) we leverage this to prove that an efficient first-order algorithm achieves near-optimal statistical precision without good initialization. We also propose a general methodology for clustering with flexible choices of feature transforms and loss objectives.

## 1 Introduction

Clustering is a fundamental problem in data science, especially in the early stages of knowledge discovery. In this paper, we consider a binary clustering problem where the data come from a mixture of two elliptical distributions. Suppose that we observe i.i.d. samples $\{X_i\}_{i=1}^n \subseteq \mathbb{R}^d$ from the latent variable model

$$X_i = \mu_0 + \mu Y_i + \Sigma^{1/2} Z_i, \qquad i \in [n]. \tag{1}$$

Here $\mu_0$, $\mu \in \mathbb{R}^d$ and $\Sigma \succ 0$ are deterministic; $Y_i \in \{\pm 1\}$ and $Z_i \in \mathbb{R}^d$ are independent random quantities; $\mathbb{P}(Y_i = -1) = \mathbb{P}(Y_i = 1) = 1/2$, and $Z_i$ is an isotropic random vector whose distribution is spherically symmetric with respect to the origin. $X_i$ is elliptically distributed (Fang et al., 1990) given $Y_i$. The goal of clustering is to estimate $\{Y_i\}_{i=1}^n$ from $\{X_i\}_{i=1}^n$. Moreover, it is desirable to build a classifier with straightforward out-of-sample extension that predicts labels for future samples.

As a warm-up example, assume for simplicity that $Z_i$ has density and $\mu_0 = 0$. The Bayes-optimal classifier is

$$\varphi_{\beta^\star}(x) = \text{sgn}(\beta^{\star\top} x) = \begin{cases} 1 & \text{if } \beta^{\star\top} x \geq 0 \\ -1 & \text{otherwise} \end{cases},$$

with any $\beta^\star \propto \Sigma^{-1}\mu$. A natural strategy for clustering is to learn a linear classifier $\varphi_\beta(x) = \text{sgn}(\beta^\top x)$ with discriminative coefficients $\beta \in \mathbb{R}^d$ estimated from the samples. Note that

$$\beta^\top X_i = (\beta^\top \mu) Y_i + \beta^\top \Sigma^{1/2} Z_i \overset{d}{=} (\beta^\top \mu) Y_i + \sqrt{\beta^\top \Sigma \beta} Z_i,$$

where $Z_i = e_1^\top Z_i$ is the first coordinate of $Z_i$. The transformed data $\{\beta^\top X_i\}_{i=1}^n$ are noisy observations of scaled labels $\{(\beta^\top \mu)Y_i\}_{i=1}^n$. A discriminative feature mapping $x \mapsto \beta^\top x$ results in high signal-to-noise ratio $(\beta^\top \mu)^2/\beta^\top \Sigma \beta$, turning the data into two well-separated clusters in $\mathbb{R}$.

When the clusters are almost spherical ($\Sigma \approx I$) or far apart ($\|\mu\|_2^2 \gg \|\Sigma\|_2$), the mean vector $\mu$ has reasonable discriminative power and the leading eigenvector of the overall covariance matrix $\mu\mu^\top + \Sigma$ roughly points that direction. This helps develop and analyze various spectral methods (Vempala and Wang, 2004; Ndaoud, 2018) based on Principal Component Analysis (PCA). $k$-means (Lu and Zhou, 2016) and its semidefinite relaxation (Mixon et al., 2017; Royer, 2017; Fei and Chen, 2018; Giraud and Verzelen, 2018; Chen and Yang, 2018) are also closely related. As they are built upon the Euclidean distance, a key assumption is the existence of well-separated balls each containing the bulk of one cluster. Existing works typically require $\|\mu\|_2^2/\|\Sigma\|_2$ to be large under models like (1). Yet, the separation is better measured by $\mu^\top \Sigma^{-1} \mu$, which always dominates $\|\mu\|_2^2/\|\Sigma\|_2$. Those methods may fail when the clusters are separated but "stretched". As a toy example, consider a Gaussian mixture $\frac{1}{2}N(\mu, \Sigma) + \frac{1}{2}N(-\mu, \Sigma)$ in $\mathbb{R}^2$ where $\mu = (1, 0)^\top$ and the covariance matrix $\Sigma = \mathrm{diag}(0.1, 10)$ is diagonal. Then the distribution consists of two separated but stretched ellipses. PCA returns the direction $(0, 1)^\top$ that maximizes the variance but is unable to tell the clusters apart.

To get high discriminative power under general conditions, we search for $\beta$ that makes $\{\beta^\top X_i\}_{i=1}^n$ concentrate around the label set $\{\pm 1\}$, through the following optimization problem:

$$\min_{\beta \in \mathbb{R}^d} \sum_{i=1}^n f(\beta^\top X_i). \tag{2}$$

Here $f : \mathbb{R} \to \mathbb{R}$ attains its minimum at $\pm 1$, e.g. $f(x) = (x^2 - 1)^2$. We name this method as "Clustering via Uncoupled REgression", or CURE for short. Here $f$ penalizes the discrepancy between predictions $\{\beta^\top X_i\}_{i=1}^n$ and labels $\{Y_i\}_{i=1}^n$. In the unsupervised setting, we have no access to the one-to-one correspondence but can still enforce proximity on the distribution level, i.e.

$$\frac{1}{n} \sum_{i=1}^n \delta_{\beta^\top X_i} \approx \frac{1}{2}\delta_{-1} + \frac{1}{2}\delta_1. \tag{3}$$

A good approximate solution to (2) leads to $|\beta^\top X_i| \approx 1$. That is, the transformed data form two clusters around $\pm 1$. The symmetry of the mixture distribution automatically ensures balance between the clusters. Thus (2) is an uncoupled regression problem based on (3). Above we focus on the centered case ($\mu_0 = 0$) merely to illustrate main ideas. Our general methodology

$$\min_{\alpha \in \mathbb{R},\ \beta \in \mathbb{R}^d} \left\{ \frac{1}{n} \sum_{i=1}^n f(\alpha + \beta^\top X_i) + \frac{1}{2}(\alpha + \beta^\top \hat{\mu}_0)^2 \right\}, \tag{4}$$

where $\hat{\mu}_0 = \frac{1}{n}\sum_{i=1}^n X_i$, deals with arbitrary $\mu_0$ by incorporating an intercept term $\alpha$.

**Main contributions.** We propose a clustering method through (4) and study it under the model (1) without requiring the clusters to be spherical. Under mild assumptions, we prove that an efficient algorithm achieves near-optimal statistical precision even in the absence of a good initialization.

- (**Loss function design**) We construct an appropriate loss function $f$ by clipping the growth of the quartic function $(x^2 - 1)^2/4$ outside some interval centered at 0. As a result, $f$ has two "valleys" at $\pm 1$ and does not grow too fast, which is beneficial to statistical analysis and optimization.

- (**Landscape analysis**) We characterize the geometry of the empirical loss function when $n/d$ exceeds some constant. In particular, all second-order stationary points, where the smallest eigenvalues of Hessians are not significantly negative, are nearly optimal in the statistical sense.

- (**Efficient algorithm with near-optimal statistical property**) We show that with high probability, a perturbed version of gradient descent algorithm starting from $0$ yields a solution with near-optimal statistical property after $\tilde{O}(n/d + d^2/n)$ iterations (up to polylogarithmic factors).

The formulation (4) is uncoupled linear regression for binary clustering. Beyond that, we introduce a unified framework which learns feature transforms to identify clusters with possibly non-convex shapes. That provides a principled way of designing flexible unsupervised learning algorithms.

We introduce the model and methodology in Section 2, conduct theoretical analysis in Section 3, present numerical results in Section 4, and finally conclude the paper with a discussion in Section 5.

**Related work.** Methodologies for clustering can be roughly categorized as generative and discriminative ones. Generative approaches fit mixture models for the joint distribution of features $\boldsymbol{X}$ and label $Y$ to make predictions (Moitra and Valiant, 2010; Kannan et al., 2005; Anandkumar et al., 2014). Their success usually hinges on well-specified models and precise estimation of parameters. Since clustering is based on the conditional distribution of $Y$ given $\boldsymbol{X}$, it only involves certain functional of parameters. Generative approaches often have high overhead in terms of sample size and running time. On the other hand, discriminative approaches directly aim for predictive classifiers. A common strategy is to learn a transform to turn the data into a low-dimensional point cloud that facilitates clustering. Statistical analysis of mixture models lead to information-based methods (Bridle et al., 1992; Krause et al., 2010), analogous to the logistic regression for supervised classification. Geometry-based methods uncover latent structures in an intuitive way, similar to the support vector machine. Our method CURE belongs to this family. Other examples include projection pursuit (Friedman and Tukey, 1974; Peña and Prieto, 2001), margin maximization (Ben-Hur et al., 2001; Xu et al., 2005), discriminative $k$-means (Ye et al., 2008; Bach and Harchaoui, 2008), graph cut optimization by spectral methods (Shi and Malik, 2000; Ng et al., 2002) and semidefinite programming (Weinberger and Saul, 2006). Discriminative methods are easily integrated with modern tools such as deep neural networks (Springenberg, 2015; Xie et al., 2016). The list above is far from exhaustive.

The formulation (4) is invariant under invertible affine transforms of data and thus tackles stretched mixtures which are catastrophic for many existing approaches. A recent paper Kushnir et al. (2019) uses random projections to tackle such problem but requires the separation between two clusters to grow at the order of $\sqrt{d}$, where $d$ is the dimension. There have been provable algorithms dealing with general models with multiple classes and minimal separation conditions (Brubaker and Vempala, 2008; Kalai et al., 2010; Belkin and Sinha, 2015). However, their running time and sample complexity are large polynomials in the dimension and desired precision. In the class of two-component mixtures we consider, CURE has near-optimal (linear) sample complexity and runs fast in practice. Another relevant area of study is clustering under sparse mixture models (Azizyan et al., 2015; Verzelen and Arias-Castro, 2017), where additional structures help handle non-spherical clusters efficiently.

The vanilla version of CURE in (2) is closely related to the Projection Pursuit (PP) (Friedman and Tukey, 1974) and Independent Component Analysis (ICA) (Hyvärinen and Oja, 2000). PP and ICA find the most nontrivial direction by maximizing the deviation of the projected data from some null distribution (e.g. Gaussian). Their objective functions are designed using key features of that. Notably, Peña and Prieto (2001) propose clustering algorithms based on extreme projections that maximize and minimize the kurtosis; Verzelen and Arias-Castro (2017) use the first absolute moment and skewness to construct objective functions in pursuit of projections for clustering. On the contrary, CURE stems from uncoupled regression and minimizes the discrepancy between the projected data and some target distribution. This makes it generalizable beyond linear feature transforms with flexible choices of objective functions. Moreover, CURE has nice computational guarantees while only a few algorithms for PP and ICA do. The formulation (2) with double-well loss $f$ also appears in the real version of Phase Retrieval (PR) (Candes et al., 2015) for recovering a signal $\boldsymbol{\beta}$ from noisy quadratic measurements $Y_i \approx (\boldsymbol{X}_i^\top \boldsymbol{\beta})^2$. In both CURE and PR, one observes the magnitudes of labels/outputs without sign information. However, algorithmic study of PR usually require $\{\boldsymbol{X}_i\}_{i=1}^n$ to be isotropic Gaussian; most efficient algorithms need good initializations by spectral methods. Those cannot be easily adapted to clustering. Our analysis of CURE could provide a new way of studying PR under more general conditions.

**Notation.** Let $[n] = \{1, 2, \cdots, n\}$. Denote by $|\cdot|$ the absolute value of a real number or cardinality of a set. For real numbers $a$ and $b$, let $a \wedge b = \min\{a, b\}$ and $a \vee b = \max\{a, b\}$. For nonnegative sequences $\{a_n\}_{n=1}^\infty$ and $\{b_n\}_{n=1}^\infty$, we write $a_n \lesssim b_n$ or $a_n = O(b_n)$ if there exists a positive constant $C$ such that $a_n \leq C b_n$. In addition, we write $a_n = \tilde{O}(b_n)$ if $a_n = O(b_n)$ holds up to some logarithmic factor; $a_n \asymp b_n$ if $a_n \lesssim b_n$ and $b_n \lesssim a_n$. We let $\mathbf{1}_S$ be the indicator function of a set $S$. We equip $\mathbb{R}^d$ with the inner product $\langle \boldsymbol{x}, \boldsymbol{y} \rangle = \boldsymbol{x}^\top \boldsymbol{y}$, Euclidean norm $\|\boldsymbol{x}\|_2 = \sqrt{\langle \boldsymbol{x}, \boldsymbol{x} \rangle}$ and canonical bases $\{\boldsymbol{e}_j\}_{j=1}^d$. Let $\mathbb{S}^{d-1} = \{\boldsymbol{x} \in \mathbb{R}^d : \|\boldsymbol{x}\|_2 = 1\}$, $B(\boldsymbol{x}, r) = \{\boldsymbol{y} \in \mathbb{R}^d : \|\boldsymbol{y} - \boldsymbol{x}\|_2 \leq r\}$, and $\mathrm{dist}(\boldsymbol{x}, S) = \inf_{\boldsymbol{y} \in S} \|\boldsymbol{x} - \boldsymbol{y}\|_2$ for $S \subseteq \mathbb{R}^d$. For a matrix $\boldsymbol{A}$, we define its spectral norm $\|\boldsymbol{A}\|_2 = \sup_{\|\boldsymbol{x}\|_2 = 1} \|\boldsymbol{A}\boldsymbol{x}\|_2$. For a symmetric matrix $\boldsymbol{A}$, we use $\lambda_{\max}(\boldsymbol{A})$ and $\lambda_{\min}(\boldsymbol{A})$ to represent its largest and smallest eigenvalues, respectively. For a positive definite matrix $\boldsymbol{A} \succ 0$, let $\|\boldsymbol{x}\|_{\boldsymbol{A}} = \sqrt{\boldsymbol{x}^\top \boldsymbol{A}\boldsymbol{x}}$.

Denote by $\delta_{\boldsymbol{x}}$ the point mass at $\boldsymbol{x}$. Define $\|X\|_{\psi_2} = \sup_{p \geq 1} p^{-1/2} \mathbb{E}^{1/p} |X|^p$ for random variable $X$ and $\|\boldsymbol{X}\|_{\psi_2} = \sup_{\|\boldsymbol{u}\|_2 = 1} \|\langle \boldsymbol{u}, \boldsymbol{X} \rangle\|_{\psi_2}$ for random vector $\boldsymbol{X}$.

## 2  Problem setup

### 2.1  Elliptical mixture model

**Model 1.** *Let $\boldsymbol{X} \in \mathbb{R}^d$ be a random vector with the decomposition*

$$\boldsymbol{X} = \boldsymbol{\mu}_0 + \boldsymbol{\mu} Y + \boldsymbol{\Sigma}^{1/2} \boldsymbol{Z}.$$

*Here $\boldsymbol{\mu}_0, \boldsymbol{\mu} \in \mathbb{R}^d$ and $\boldsymbol{\Sigma} \succ 0$ are deterministic; $Y \in \{\pm 1\}$ and $\boldsymbol{Z} \in \mathbb{R}^d$ are random and independent. Let $Z = \boldsymbol{e}_1^\top \boldsymbol{Z}$, $\rho$ be the distribution of $\boldsymbol{X}$ and $\{\boldsymbol{X}_i\}_{i=1}^n$ be i.i.d. samples from $\rho$.*

- **(Balanced classes)** $\mathbb{P}(Y = -1) = \mathbb{P}(Y = 1) = 1/2$;

- **(Elliptical sub-Gaussian noise)** $\boldsymbol{Z}$ is sub-Gaussian with $\|\boldsymbol{Z}\|_{\psi_2}$ bounded by some constant $M$, $\mathbb{E}\boldsymbol{Z} = \boldsymbol{0}$ and $\mathbb{E}(\boldsymbol{Z}\boldsymbol{Z}^\top) = \boldsymbol{I}_d$; its distribution is spherically symmetric with respect to $\boldsymbol{0}$;

- **(Leptokurtic distribution)** $\mathbb{E}Z^4 - 3 > \kappa_0$ holds for some constant $\kappa_0 > 0$;

- **(Regularity)** $\|\boldsymbol{\mu}_0\|_2$, $\|\boldsymbol{\mu}\|_2$, $\lambda_{\max}(\boldsymbol{\Sigma})$ and $\lambda_{\min}(\boldsymbol{\Sigma})$ are bounded away from 0 and $\infty$ by constants.

We aim to build a classifier $\mathbb{R}^d \to \{\pm 1\}$ based solely on the samples $\{\boldsymbol{X}_i\}_{i=1}^n$ from a mixture of two elliptical distributions. For simplicity, we assume that the two classes are balanced and focus on the well-conditioned case where the signal strength and the noise level are of constant order. This is already general enough to include stretched clusters incapacitating many popular methods including PCA, $k$-means and semi-definite relaxations (Brubaker and Vempala, 2008). One may wonder whether it is possible to transform the data into what they can handle. While multiplication by $\boldsymbol{\Sigma}^{-1/2}$ yields spherical clusters, precise estimation of $\boldsymbol{\Sigma}^{-1/2}$ or $\boldsymbol{\Sigma}$ is no easy task under the mixture model. Dealing with those $d \times d$ matrices causes overhead expenses in computation and storage. The assumption on positive excess kurtosis prevents the loss function from having undesirable degenerate saddle points and facilitates the proof of algorithmic convergence. It rules out distributions whose kurtoses do not exceed that of the normal distribution, and it is not clear whether there exists an easy fix for that. The last assumption in Model 1 makes the loss landscape regular, helps avoid undesirable technicalities, and is commonly adopted in the study of parameter estimation in mixture models. The Bayes optimal classification error is of constant order, and we want to achieve low excess risk.

### 2.2  Clustering via Uncoupled Regression

Under Model 1, the Bayes optimal classifier for predicting $Y$ given $\boldsymbol{X}$ is

$$\hat{Y}^{\text{Bayes}}(\boldsymbol{X}) = \text{sgn}\left(\alpha^{\text{Bayes}} + \boldsymbol{\beta}^{\text{Bayes}\top}\boldsymbol{X}\right),$$

where $\left(\alpha^{\text{Bayes}}, \boldsymbol{\beta}^{\text{Bayes}}\right) = \left(-\boldsymbol{\mu}_0^\top \boldsymbol{\Sigma}^{-1}\boldsymbol{\mu}, \boldsymbol{\Sigma}^{-1}\boldsymbol{\mu}\right)$. On the other hand, it is easily seen that the following (population-level) least squares problem $\mathbb{E}[(\alpha + \boldsymbol{\beta}^\top \boldsymbol{X}) - Y]^2$ has a unique solution $(\alpha^{\text{LR}}, \boldsymbol{\beta}^{\text{LR}}) = (-c\boldsymbol{\mu}_0^\top \boldsymbol{\Sigma}^{-1}\boldsymbol{\mu}, c\boldsymbol{\Sigma}^{-1}\boldsymbol{\mu})$ for some $c > 0$. For the supervised classification problem where we observe $\{(\boldsymbol{X}_i, Y_i)\}_{i=1}^n$, the optimal feature transform can be estimated via linear regression

$$\frac{1}{n}\sum_{i=1}^n [(\alpha + \boldsymbol{\beta}^\top \boldsymbol{X}_i) - Y_i]^2. \tag{5}$$

This is closely related to Fisher's Linear Discriminant Analysis (Friedman et al., 2001).

In the unsupervised clustering problem, we no longer observe individual labels $\{Y_i\}_{i=1}^n$ associated with $\{\boldsymbol{X}_i\}_{i=1}^n$ but have population statistics of labels, as the classes are balanced. While (5) directly forces $\alpha + \boldsymbol{\beta}^\top \boldsymbol{X}_i \approx Y_i$ thanks to supervision, here we relax such proximity to the population level:

$$\frac{1}{n}\sum_{i=1}^n \delta_{\alpha + \boldsymbol{\beta}^\top \boldsymbol{X}_i} \approx \frac{1}{2}\delta_{-1} + \frac{1}{2}\delta_1. \tag{6}$$

Thus the regression should be conducted in an uncoupled manner using marginal information about $\boldsymbol{X}$ and $Y$. We seek for an affine transformation $\boldsymbol{x} \mapsto \alpha + \boldsymbol{\beta}^\top \boldsymbol{x}$ to turn the samples $\{\boldsymbol{X}_i\}_{i=1}^n$ into two balanced clusters around $\pm 1$, after which $\mathrm{sgn}(\alpha + \boldsymbol{\beta}^\top \boldsymbol{X})$ predicts $Y$ up to a global sign flip. It is also supported by the geometric intuition in Section 1 based on projections of the mixture distribution.

Clustering via Uncoupled REgression (CURE) is formulated as an optimization problem:

$$\min_{\alpha \in \mathbb{R}, \ \boldsymbol{\beta} \in \mathbb{R}^d} \left\{ \frac{1}{n} \sum_{i=1}^n f(\alpha + \boldsymbol{\beta}^\top \boldsymbol{X}_i) + \frac{1}{2}(\alpha + \boldsymbol{\beta}^\top \hat{\boldsymbol{\mu}}_0)^2 \right\}, \tag{7}$$

where $\hat{\boldsymbol{\mu}}_0 = \frac{1}{n} \sum_{i=1}^n \boldsymbol{X}_i$. $f$ attains its minimum at $\pm 1$. Minimizing $\frac{1}{n} \sum_{i=1}^n f(\alpha + \boldsymbol{\beta}^\top \boldsymbol{X}_i)$ makes the transformed data $\{\alpha + \boldsymbol{\beta}^\top \boldsymbol{X}_i\}_{i=1}^n$ concentrate around $\{\pm 1\}$. However, there are always two trivial minimizers $(\alpha, \boldsymbol{\beta}) = (\pm 1, \boldsymbol{0})$, each of which maps the entire dataset to a single point. What we want are two balanced clusters around $-1$ and $1$. The centered case ($\boldsymbol{\mu}_0 = \boldsymbol{0}$) discussed in Section 1 does not have such trouble as $\alpha$ is set to be 0 and the symmetry of the mixture automatically balance the two clusters. For the general case, we introduce a penalty term $(\alpha + \boldsymbol{\beta}^\top \hat{\boldsymbol{\mu}}_0)^2/2$ in (7) to drive the center of the transformed data towards 0. The idea comes from moment-matching and is similar to that in Flammarion et al. (2017). If $\frac{1}{n} \sum_{i=1}^n f(\alpha + \boldsymbol{\beta}^\top \boldsymbol{X}_i)$ is small, then $|\alpha + \boldsymbol{\beta}^\top \boldsymbol{X}_i| \approx 1$ and

$$\frac{1}{n} \sum_{i=1}^n \delta_{\alpha + \boldsymbol{\beta}^\top \boldsymbol{X}_i} \approx \frac{|\{i : \ \alpha + \boldsymbol{\beta}^\top \boldsymbol{X}_i \geq 0\}|}{n} \delta_1 + \frac{|\{i : \ \alpha + \boldsymbol{\beta}^\top \boldsymbol{X}_i < 0\}|}{n} \delta_{-1}.$$

Then, in order to get (6), we simply match the expectations on both sides. This gives rise to the quadratic penalty term in (7). The same idea generalizes beyond the balanced case. When the two classes 1 and $-1$ have probabilities $p$ and $(1-p)$, we can match the mean of $\{\alpha + \boldsymbol{\beta}^\top \boldsymbol{X}_i\}_{i=1}^n$ with that of a new target distribution $p\delta_1 + (1-p)\delta_{-1}$, and change the quadratic penalty to $[(\alpha + \boldsymbol{\beta}^\top \hat{\boldsymbol{\mu}}_0) - (2p-1)]^2$. When $p$ is unknown, (7) can always be a surrogate as it seeks for two clusters around $\pm 1$ and uses the quadratic penalty to prevent any of them from being vanishingly small.

The function $f$ in (7) requires careful design. To facilitate statistical and algorithmic analysis, we want $f$ to be twice continuously differentiable and grow slowly. That makes the empirical loss smooth and concentrate well around its population counterpart. In addition, the coercivity of $f$, i.e. $\lim_{|x| \to \infty} f(x) = +\infty$, confines all minimizers within some ball of moderate size. Similar to the construction of Huber loss (Huber, 1964), we start from $h(x) = (x^2 - 1)^2/4$, keep its two valleys around $\pm 1$, clip its growth using linear functions and interpolate in between using cubic splines:

$$f(x) = \begin{cases} h(x), & |x| \leq a \\ h(a) + h'(a)(|x| - a) + \frac{h''(a)}{2}(|x| - a)^2 - \frac{h''(a)}{6(b-a)}(|x| - a)^3, & a < |x| \leq b \ . \\ f(b) + [h'(a) + \frac{b-a}{2}h''(a)](|x| - b), & |x| > b \end{cases} \tag{8}$$

Here $b > a > 1$ are constants to be determined later. $f$ is clearly not convex, and neither is the loss function in (7). Yet we can find a good approximate solution efficiently by taking advantage of statistical assumptions and recent advancements in non-convex optimization (Jin et al., 2017).

## 2.3 Generalization

The aforementioned procedure seeks for a one-dimensional embedding of the data that facilitates clustering. It searches for the best affine function such that the transformed data look like a two-point distribution. The idea of uncoupled linear regression can be easily generalized to any suitable target probability distribution $\nu$ over a space $\mathcal{Y}$, class of feature transforms $\mathcal{F}$ from the original space $\mathcal{X}$ to $\mathcal{Y}$, discrepancy measure $D$ that quantifies the difference between the transformed data distribution and $\nu$, and classification rule $g : \ \mathcal{Y} \to [K]$. CURE for Model 1 above uses $\mathcal{X} = \mathbb{R}^d$, $\mathcal{Y} = \mathbb{R}$, $\nu = \frac{1}{2}\delta_{-1} + \frac{1}{2}\delta_1$, $\mathcal{F} = \{\boldsymbol{x} \mapsto \alpha + \boldsymbol{\beta}^\top \boldsymbol{x} : \ \alpha \in \mathbb{R}, \ \boldsymbol{\beta} \in \mathbb{R}^d\}$, $g(y) = \mathrm{sgn}(y)$ and

$$D(\mu, \nu) = |\mathbb{E}_{X \sim \mu} f(X) - \mathbb{E}_{X \sim \nu} f(X)| + \frac{1}{2}|\mathbb{E}_{X \sim \mu} X - \mathbb{E}_{X \sim \nu} X|^2 \tag{9}$$

for any probability distribution $\mu$ over $\mathbb{R}$. Here we briefly show why (9) is true. Fix any $f : \ \boldsymbol{x} \mapsto \alpha + \boldsymbol{\beta}^\top \boldsymbol{x}$ in $\mathcal{F}$ and let $\mu = \frac{1}{n} \sum_{i=1}^n \delta_{\alpha + \boldsymbol{\beta}^\top \boldsymbol{X}_i}$ be the transformed data distribution. From $f(-1) =$

**Algorithm 1** Clustering via Uncoupled REgression (meta-algorithm)

---

**Input:** Data $\{\boldsymbol{X}_i\}_{i=1}^n$ in a feature space $\mathcal{X}$, embedding space $\mathcal{Y}$, target distribution $\nu$ over $\mathcal{Y}$, discrepancy measure $D$, function class $\mathcal{F}$, classification rule $g$.
**Embedding:** find an approximation solution $\hat{\varphi}$ to $\min_{\varphi \in \mathcal{F}} D(\varphi_\# \hat{\rho}_n, \nu)$.
**Output:** $\hat{Y}_i = g[\hat{\varphi}(\boldsymbol{X}_i)]$ for $i \in [n]$.

---

**Algorithm 2** Perturbed gradient descent

---

**Initialize** $\boldsymbol{\gamma}^0 = \boldsymbol{0}$.
**For** $t = 0, 1, \ldots$ **do**
    **If** perturbation condition holds:        Perturb $\boldsymbol{\gamma}^t \leftarrow \boldsymbol{\gamma}^t + \boldsymbol{\xi}^t$ with $\boldsymbol{\xi}^t \sim \mathcal{U}(B(\boldsymbol{0}, r))$
    **If** termination condition holds:       **Return** $\boldsymbol{\gamma}^t$
    **Update** $\boldsymbol{\gamma}^{t+1} \leftarrow \boldsymbol{\gamma}^t - \eta \nabla \hat{L}_1(\boldsymbol{\gamma}^t)$.

---

$f(1) = 0$ and $\mathbb{E}_{X \sim \nu} X = 0$ we see

$$|\mathbb{E}_{X \sim \mu} f(X) - \mathbb{E}_{X \sim \nu} f(X)| = \mathbb{E}_{X \sim \mu} f(X) = \frac{1}{n} \sum_{i=1}^n f(\alpha + \boldsymbol{\beta}^\top \boldsymbol{X}_i),$$

$$|\mathbb{E}_{X \sim \mu} X - \mathbb{E}_{X \sim \nu} X|^2 = \left( \frac{1}{n} \sum_{i=1}^n (\alpha + \boldsymbol{\beta}^\top \boldsymbol{X}_i) \right)^2 = (\alpha + \boldsymbol{\beta}^\top \hat{\boldsymbol{\mu}}_0),$$

$$D(\mu, \nu) = \frac{1}{n} \sum_{i=1}^n f(\alpha + \boldsymbol{\beta}^\top \boldsymbol{X}_i) + \frac{1}{2}(\alpha + \boldsymbol{\beta}^\top \hat{\boldsymbol{\mu}}_0).$$

On top of that, we propose a general framework for clustering (also named as CURE) and describe it at a high level of abstraction in Algorithm 1. Here $\hat{\rho}_n = \frac{1}{n} \sum_{i=1}^n \delta_{\boldsymbol{X}_i}$ is the empirical distribution of data and $\varphi_\# \hat{\rho}_n = \frac{1}{n} \sum_{i=1}^n \delta_{\varphi(\boldsymbol{X}_i)}$ is the push-forward distribution. The general version of CURE is a flexible framework for clustering based on uncoupled regression (Rigollet and Weed, 2019). For instance, we may set $\mathcal{Y} = \mathbb{R}^K$ and $\nu = \frac{1}{K} \sum_{k=1}^K \delta_{\boldsymbol{e}_k}$ when there are $K$ clusters; choose $\mathcal{F}$ to be the family of convolutional neural networks for image clustering; let $D$ be the Wasserstein distance or some divergence. CURE is easily integrated with other tools, see Section A.2 in the supplementary material.

## 3 Theoretical analysis

### 3.1 Main results

Let $\hat{L}_1(\alpha, \boldsymbol{\beta})$ denote the objective function of CURE in (7). Our main result (Theorem 1) shows that with high probability, a perturbed version of gradient descent (Algorithm 2) applied to $\hat{L}_1$ returns an approximate minimizer that is nearly optimal in the statistical sense, within a reasonable number of iterations. Here $\mathcal{U}(B(\boldsymbol{0}, r))$ refers to the uniform distribution over $B(\boldsymbol{0}, r)$. We omit technical details of the algorithm and defer them to Appendix B.4, see Algorithm 1 and Theorem 3 therein. For notational simplicity, we write $\boldsymbol{\gamma} = (\alpha, \boldsymbol{\beta}) \in \mathbb{R} \times \mathbb{R}^d$ and $\boldsymbol{\gamma}^{\text{Bayes}} = (\alpha^{\text{Bayes}}, \boldsymbol{\beta}^{\text{Bayes}}) = (-\boldsymbol{\mu}^\top \boldsymbol{\Sigma}^{-1} \boldsymbol{\mu}_0, \boldsymbol{\Sigma}^{-1} \boldsymbol{\mu})$. $\boldsymbol{\gamma}^{\text{Bayes}}$ defines the Bayes-optimal classifier $\boldsymbol{x} \mapsto \text{sgn}(\alpha^{\text{Bayes}} + \boldsymbol{\beta}^{\text{Bayes}\top} \boldsymbol{x})$ for Model 1.

**Theorem 1** (Main result). *Let $\boldsymbol{\gamma}_0, \boldsymbol{\gamma}_1, \cdots$ be the iterates of Algorithm 2 starting from $\boldsymbol{0}$. Under Model 1 there exist constants $c, C, C_0, C_1, C_2 > 0$ independent of $n$ and $d$ such that if $n \geq Cd$ and $b \geq 2a \geq C_0$, then with probability at least $1 - C_1[(d/n)^{C_2 d} + e^{-C_2 n^{1/3}} + n^{-10}]$, Algorithm 2 terminates within $\tilde{O}(n/d + d^2/n)$ iterations and the output $\hat{\boldsymbol{\gamma}}$ satisfies*

$$\min_{s = \pm 1} \|s \hat{\boldsymbol{\gamma}} - c \boldsymbol{\gamma}^{\text{Bayes}}\|_2 \lesssim \sqrt{\frac{d}{n} \log \left( \frac{n}{d} \right)}.$$

Up to a $\sqrt{\log(n/d)}$ factor, this matches the optimal rate of convergence $O(\sqrt{d/n})$ for the supervised problem with $\{Y_i\}_{i=1}^n$ observed, which is even easier than the current one. Theorem 1 asserts that we

can achieve a near-optimal rate efficiently without good initialization, although the loss function is non-convex. The two terms $n/d$ and $d^2/n$ in the iteration complexity have nice interpretations. When $n$ is large, we want a small computational error in order to achieve statistical optimality. The term $n/d$ reflects the cost for this. When $n$ is small, the empirical loss function does not concentrate well and is not smooth enough either. Hence we choose a conservative step-size and pay the corresponding price $d^2/n$. A byproduct of Theorem 1 is the following corollary which gives a tight bound for the excess risk. Here $\|g\|_\infty = \sup_{x\in\mathbb{R}} |g(x)|$ for any $g: \mathbb{R} \to \mathbb{R}$. The proof is deferred to Appendix I.

**Corollary 1** (Misclassification rate). *Consider the settings in Theorem 1 and suppose that $Z = e_1^\top Z$ has density $p \in C^1(\mathbb{R})$ satisfying $\|p\|_\infty \leq C_3$ and $\|p'\|_\infty \leq C_3$ for some constant $C_3 > 0$. For $\gamma = (\alpha, \boldsymbol{\beta}) \in \mathbb{R} \times \mathbb{R}^d$, define its misclassification rate (up to a global sign flip) as*

$$\mathcal{R}(\gamma) = \min_{s=\pm 1} \mathbb{P}\left(s\,\mathrm{sgn}\left(\alpha + \boldsymbol{\beta}^\top \boldsymbol{X}\right) \neq Y\right).$$

*There exists a constant $C_4$ such that*

$$\mathbb{P}\left(\mathcal{R}(\hat{\gamma}) \leq \mathcal{R}(\gamma^{\mathrm{Bayes}}) + \frac{C_4 d \log(n/d)}{n}\right) \geq 1 - C_1[(d/n)^{C_2 d} + e^{-C_2 n^{1/3}} + n^{-10}].$$

## 3.2 Sketch of proof

The loss function $\hat{L}_1$ is non-convex in general. To find an approximate minimizer efficiently without good initialization, we need $\hat{L}_1$ to exhibit benign geometric properties that can be exploited by a simple algorithm. Our choice is the perturbed gradient descent algorithm in Jin et al. (2017), see Algorithm 1 in Appendix B.4 for more details. Provided that the function is smooth enough, it provably converges to an approximate second-order stationary point where the norm of gradient is small and the Hessian matrix does not have any significantly negative eigenvalue. Then it boils down to landscape analysis of $\hat{L}_1$ with precise characterizations of approximate stationary points. To begin with, define the population version of $\hat{L}_1$ as

$$L_1(\alpha, \boldsymbol{\beta}) = \mathbb{E}_{\boldsymbol{X}\sim\rho} f(\alpha + \boldsymbol{\beta}^\top \boldsymbol{X}) + \frac{1}{2}(\alpha + \boldsymbol{\beta}^\top \boldsymbol{\mu}_0)^2.$$

**Proposition 1.** *There exist positive constants $c, \varepsilon, \delta, \eta$ and a set $S \subseteq \mathbb{R} \times \mathbb{R}^d$ such that*

*1. The only two local minima of $L_1$ are $\pm\gamma^\star$ with $\gamma^\star = -c\gamma^{\mathrm{Bayes}}$;*

*2. All the other first-order critical points (i.e. with zero gradient) are within $\delta$ distance to $S$;*

*3. $\|\nabla L_1(\gamma)\|_2 \geq \varepsilon$ if $\mathrm{dist}(\gamma, \{\pm\gamma^\star\} \cup S) \geq \delta$;*

*4. $\nabla^2 L_1(\gamma) \succeq \eta\boldsymbol{I}$ if $\mathrm{dist}(\gamma, \{\pm\gamma^\star\}) \leq \delta$, and $\lambda_{\min}[\nabla^2 L_1(\gamma)] \leq -\eta$ if $\mathrm{dist}(\gamma, S) \leq \delta$.*

Proposition 1 shows that all of the approximate second-order critical points of $L_1$ are close to that corresponding to the Bayes-optimal classifier. Then we will prove similar results for the empirical loss $\hat{L}_1$ using concentration inequalities, which leads to the following proposition translating approximate second-order stationarity to estimation error.

**Proposition 2.** *There exists a constant $C$ such that the followings happen with high probability: for any $\gamma \in \mathbb{R} \times \mathbb{R}^d$ satisfying $\|\nabla\hat{L}_1(\gamma)\|_2 \leq \varepsilon/2$ and $\lambda_{\min}[\nabla^2\hat{L}_1(\gamma)] > -\eta/2$,*

$$\min_{s=\pm 1} \|s\gamma - \gamma^\star\|_2 \leq C\left(\left\|\nabla\hat{L}_1(\gamma)\right\|_2 + \sqrt{\frac{d}{n}\log\left(\frac{n}{d}\right)}\right).$$

To achieve near-optimal statistical error (up to a $\sqrt{\log(n/d)}$ factor), Proposition 2 asserts that it suffices to find any $\hat{\gamma}$ such that $\|\nabla\hat{L}_1(\hat{\gamma})\|_2 \lesssim \sqrt{d/n}$ and $\lambda_n[\nabla^2\hat{L}_1(\hat{\gamma})] > -\eta/2$. Here the perturbed gradient descent algorithm comes into play, and we see the light at the end of the tunnel. It remains to estimate the Lipschitz smoothness of $\nabla\hat{L}_1$ and $\nabla^2\hat{L}_1$ with respect to the Euclidean norm. Once this is done, we can directly apply the convergence theorem in Jin et al. (2017) for the perturbed gradient descent. A more comprehensive outline of the proof and all the details are deferred to the Appendix.

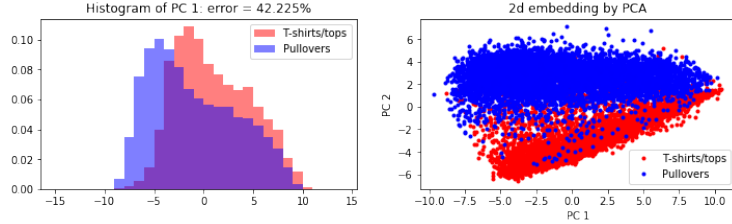

Figure 1: Visualization of the dataset via PCA. The left plot shows the transformed data via PCA. The right polt is a 2-dimensional visualization of the dataset using PCA.

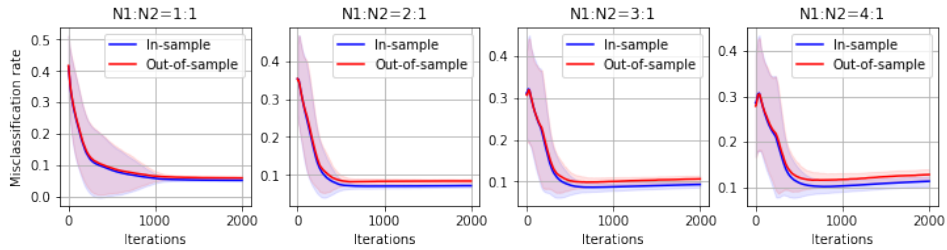

Figure 2: In sample and out-of-sample misclassification rate (with error bar quantifying one standard deviation) vs. iteration count for CURE over 50 independent trials. The four plots corresponds to $N_2 = 6000, 3000, 2000$ and $1500$ respectively, while $N_1$ is always fixed to be 6000.

## 4    Numerical experiments

In this section, we conduct numerical experiments on a real dataset. We randomly select $N_1$ (resp. $M_1$) T-shirts/tops and $N_2$ (resp. $M_2$) pullovers from the Fashion-MNIST (Xiao et al., 2017) training (resp. testing) dataset, each of which is a $28 \times 28$ grayscale image represented by a vector in $[0, 1]^{28 \times 28}$. The goal is clustering, i.e. learning from those $N = N_1 + N_2$ unlabeled images to predict the class labels of both $N$ training samples and $M = M_1 + M_2$ testing samples. The inputs for CURE and other methods are raw images and their pixel-wise centered versions, respectively. To get a sense why this problem is difficult, we set $N_1 = N_2 = 6000$ and plot the transformed data via PCA in the left panel of Figure 1: the transformation does not give meaningful clustering information, and the misclassification rate is $42.225\%$. A 2-dimensional visualization of the dataset using PCA (right panel of Figure 1) shows two stretched clusters, which cause the PCA to fail. In this dataset, the bulk of a image corresponds to the belly part of clothing with different grayscales, logos and hence contributes to the most of variability. However, T-shirts and Pullovers are distinguished by sleeves. Hence the two classes can be separated by a linear function that is not related to the leading principle component of data. CURE aims for such direction onto which the projected data exhibit cluster structures.

To show that CURE works beyond our theory, we set $N_1$ to be 6000 and choose $N_2$ from $\{6000, 3000, 2000, 1500\}$ to include unbalanced cases. We set $M_1$ to be 1000 and choose $M_2$ from $\{1000, 500, 333, 250\}$. We use gradient descent with random initialization from the unit sphere and learning rate $10^{-3}$ (instead of perturbed gradient descent) to solve (7) as that requires less tuning. Figure 2 shows the learning curves of CURE over 50 independent trials. Even when the classes are unbalanced, CURE still reliably achieves low misclassification rates. Figure 3 presents histograms of testing data under the feature transform learned by the last (50th) trial of CURE, showing two seperated clusters around $\pm 1$ corresponding to the two classes. To demonstrate the efficacy of CURE, we compare its misclassification rates with those of K-means and spectral methods on the training sets. We include the standard deviation over 50 independent trials for CURE due to its random initializations; other methods use the default settings (in Python) and thus are regarded as deterministic algorithms. As is shown in Table 1, CURE has the best performance under all settings.

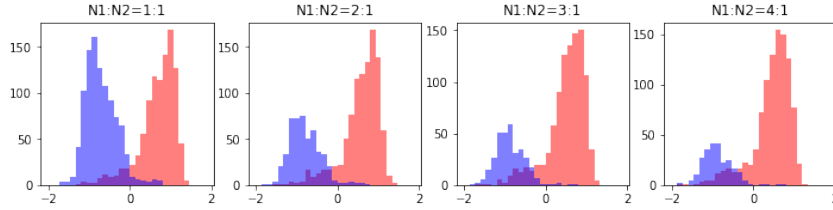

Figure 3: Histograms of transformed out-of-sample data for CURE. The red bins correspond to T-shirts/tops, and the blue bins correspond to pullovers.

Table 1: Misclassification rate of CURE and other methods.

| $N_1 : N_2$ <br> Method | $1 : 1$ | $2 : 1$ | $3 : 1$ | $4 : 1$ |
|---|---|---|---|---|
| CURE | $5.2 \pm 0.2\%$ | $7.1 \pm 0.4\%$ | $9.3 \pm 0.7\%$ | $11.3 \pm 1.1\%$ |
| K-means | $45.1\%$ | $49.7\%$ | $46.8\%$ | $45.1\%$ |
| Spectral method (vanilla) | $42.2\%$ | $46.9\%$ | $49.7\%$ | $49.0\%$ |
| Spectral method (Gaussian kernel) | $49.9\%$ | $33.4\%$ | $25.0\%$ | $20.0\%$ |

## 5   Discussion

Motivated by the elliptical mixture model (Model 1), we propose a discriminative clustering method CURE and establish near-optimal statistical guarantees for an efficient algorithm. It is worth pointing out that CURE learns a classification rule that readily predicts labels for any new data. This is an advantage over many existing approaches for clustering and embedding whose out-of-sample extensions are not so straightforward. We impose several technical assumptions (spherical symmetry, constant condition number, positive excess kurtosis, etc.) to simplify the analysis, which we believe can be relaxed. Achieving Bayes optimality in multi-class clustering is indeed very challenging. Under parametric models such as Gaussian mixtures, one may construct suitable loss functions for CURE based on likelihood functions and obtain statistical guarantees. Other directions that are worth exploring include the optimal choice of the target distribution and the discrepancy measure, high-dimensional clustering with additional structures, estimation of the number of clusters, to name a few. We also hope to further extend our methodology and theory to other tasks in unsupervised learning and semi-supervised learning.

The general CURE (Algorithm 1) provides versatile tools for clustering problems. In fact, it is related to several methods in the deep learning literature (Springenberg, 2015; Xie et al., 2016; Yang et al., 2017). When we were finishing the paper, we noticed that Genevay et al. (2019) develop a deep clustering algorithm based on $k$-means and use optimal transport to incorporate prior knowledge of class proportions. Those methods are built upon certain network architectures (function classes) or loss functions while CURE offers more choices. In addition to the preliminary numerical results, it would be nice to see how CURE tackles more challenging real data problems.

## Broader Impact

This work presents a framework CURE for solving clustering problems which are ubiquitous in data science problems, especially in early stages of knowledge discovery. Thanks to its flexibility, CURE has potential applications in numerous fields including science, engineering, economics, sociology and so on. It can be easily integrated with other tools in machine learning and can be adapted to meet ethical and societal standards. Our theoretical analysis under a canonical model establishes guarantees for CURE and provides useful guidances to practitioners. Numerical experiments on image data demonstrate the remarkable efficacy of CURE. For better deployment of the system in sensitive real-world problems, we still need to ensure the reliability, quantify the uncertainty and develop diagnosis procedures in case of failure. These are fundamental questions worth investigation in the future.

## Acknowledgments and Disclosure of Funding

We thank Philippe Rigollet and Damek Davis for insightful and stimulating discussions. Kaizheng Wang acknowledges support from the Harold W. Dodds Fellowship at Princeton University where part of the work was done. Yuling Yan is supported in part by the AFOSR grant FA9550-19-1-0030. Mateo Díaz would like to thank his advisor, Damek Davis, for research funding during the completion of this work.

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
