[Supplementary Material]

# Supplement to "Efficient Clustering for Stretched Mixtures: Landscape and Optimality"

**Kaizheng Wang**
Columbia University
kaizheng.wang@columbia.edu

**Yuling Yan**
Princeton University
yulingy@princeton.edu

**Mateo Díaz**
Cornell University
md825@cornell.edu

## A  Additional numerical experiments

### A.1  Two classes

In this section, we provide additional numerical experiments to compare CURE in (7) with other clustering methods on the same real dataset as Section 4. We focus on six methods: (i) discriminative K-means (DisKmeans) in Ye et al. (2008); (ii) a discriminative clustering formulation described in Bach and Harchaoui (2008); Flammarion et al. (2017); (iii) Model-based clustering (Mclust) in Fraley and Raftery (1999); (iv) Projection Pursuit (PP) in Peña and Prieto (2001); (v) Adaptive LDA-guided K-means Clustering in Ding and Li (2007); and (vi) Minimum Density Hyperplane (MDH) in Pavlidis et al. (2016).

As suggested by Ye et al. (2008), the regularization parameter $\lambda$ therein has a significant impact on the performance of DisKmeans. To resolve this issue, they provide an automatic tuning framework. Here we provide a comparison between CURE and DisKmeans. For the DisKmeans, we consider pre-chosen $\lambda \in \{0, 1, 10, 100\}$ as well as $\lambda$ from the automatic tuning procedure suggested by Ye et al. (2008), initialized from 1. Due to high computational cost of DisKmeans with automatic tuning (which includes eigendecomposition of $(N_1 + N_2) \times (N_1 + N_2)$ matrix in each iteration), we conduct the experiment on smaller dataset: we fix $N_1 = 1000$ and choose $N_2$ from $\{1000, 500, 333, 250\}$. As is shown in Table 1, CURE has lower misclassification rate under all settings. It is also worth mentioning that the automatic tuning procedure sends $\lambda \to \infty$, in which case DisKmeans is equivalent to classical K-means.

Table 1: Misclassification rate of CURE and discriminative K-means.

| $N_1 : N_2$ / Method | | $1:1$ | $2:1$ | $3:1$ | $4:1$ |
|---|---|---|---|---|---|
| CURE | / | $5.2 \pm 0.3\%$ | $6.7 \pm 0.6\%$ | $9.1 \pm 0.9\%$ | $11.2 \pm 1.2\%$ |
| Discriminative K-means Ye et al. (2008) | $\lambda = 0$ | $49.9\%$ | $49.5\%$ | $49.5\%$ | $47.7\%$ |
| | $\lambda = 1$ | $48.8\%$ | $46.6\%$ | $49.4\%$ | $48.3\%$ |
| | $\lambda = 10$ | $46.5\%$ | $44.2\%$ | $47.4\%$ | $41.8\%$ |
| | $\lambda = 100$ | $6.6\%$ | $49.4\%$ | $46.5\%$ | $27.2\%$ |
| | automatic tuning | $43.3\%$ | $49.4\%$ | $47.5\%$ | $45.8\%$ |

For experiments comparing CURE with other five methods, we still adopt the usual setting of sample size: we fix $N_1 = 6000$ and choose $N_2$ from $\{6000, 3000, 2000, 1500\}$. Model-based clustering (Mclust) in Fraley and Raftery (1999), Projection Pursuit (PP) in Peña and Prieto (2001) and Minimum

Density Hyperplane (MDH) in Pavlidis et al. (2016) are implemented using open-source R packages with default settings. In addition:

1. The discriminative clustering method appeared in Bach and Harchaoui (2008); Flammarion et al. (2017) stems from the optimization problem

$$\min_{\boldsymbol{v}\in\mathbb{R}^d,\boldsymbol{y}\in\{\pm 1\}^d} \|\boldsymbol{y} - \boldsymbol{X}\boldsymbol{v}\|_2^2, \tag{1}$$

where $\boldsymbol{X}$ is the centered data matrix. We adopt the alternating minimization scheme: given $\boldsymbol{v}$, the optimal $\boldsymbol{y}$ is obtained by $\mathrm{sgn}(\boldsymbol{X}\boldsymbol{v})$ (or by running K-means on $\boldsymbol{X}\boldsymbol{v}$, which has similar empirical performance) while given $\boldsymbol{y}$, the optimal $\boldsymbol{v}$ is obtained from solving a least squares problem. In the first step, $\boldsymbol{v}$ is initialized from a uniform distribution over the unit sphere. The iterative algorithm is terminated when $\boldsymbol{y}$, the predicted label, no longer changes.

2. Following the instructions in Ding and Li (2007), we implement the adaptive LDA-guided K-means clustering algorithm (Algorithm 1 therein) by alternating between linear discriminant analysis and K-means until convergence.

Table 2 shows the misclassification rate and the standard deviation of CURE and the other five methods over 50 independent trials. It is clear that CURE is more accurate and stable than these five methods under all settings.

Table 2: Misclassification rate of CURE and Method (1).

| $N_1 : N_2$ \ Method | 1 : 1 | 2 : 1 | 3 : 1 | 4 : 1 |
|---|---|---|---|---|
| CURE | $5.2 \pm 0.2\%$ | $7.1 \pm 0.4\%$ | $9.3 \pm 0.7\%$ | $11.3 \pm 1.1\%$ |
| Method (1) | $31.1 \pm 13.8\%$ | $32.9 \pm 13.3\%$ | $34.7 \pm 12.7\%$ | $36.8 \pm 11.2\%$ |
| Mclust | $48.7 \pm 1.3\%$ | $39.1 \pm 4.8\%$ | $34.1 \pm 8.0\%$ | $28.2 \pm 7.8\%$ |
| Projection Pursuit | $36.9 \pm 9.8\%$ | $37.4 \pm 9.6\%$ | $39.7 \pm 6.9\%$ | $40.6 \pm 7.3\%$ |
| LDA-guided K-means | $45.9\%$ | $49.0\%$ | $45.6\%$ | $44.3\%$ |
| MDH | $48.6\%$ | $43.1\%$ | $38.3\%$ | $35.2\%$ |

## A.2 Multiple classes

To illustrate how the general CURE in Section 2.3 works, we consider the clustering problem with the first 4 classes in Fashion-MNIST (T-shirt/top, Trouser, Pullover, Dress), each of which has 6000 training samples and 1000 testing samples. Our training process only uses features of training samples and does not touch any labels.

We let the number of classes $K$ be 4, the embedding space $\mathcal{Y}$ be $\mathbb{R}^K$, the target distribution $\nu$ be $\frac{1}{K}\sum_{j=1}^{K}\delta_{\boldsymbol{e}_j}$, the discrepancy measure $D$ be the Wasserstein-1 distance, and define the classification rule $g(\boldsymbol{y}) = \mathrm{argmin}_{j\in[K]}\|\boldsymbol{y} - \boldsymbol{e}_j\|_2$. We compare two classes $\mathcal{F}$ of feature mappings: linear functions and fully-connected neural networks with one hidden layer that has 100 nodes. Initial values All of the weight parameters are initialized using i.i.d. samples from $N(0, 0.05^2)$.

Let $f_{\boldsymbol{\theta}}$ be a feature transform in $\mathcal{F}$, parametrized by $\boldsymbol{\theta}$. Denote by $\{\boldsymbol{x}_i\}_{i=1}^n$ the samples, where $n = 4 \times 6000 = 24000$. The loss function is

$$L(\boldsymbol{\theta}) = W_1\left(\frac{1}{n}\sum_{i=1}^{n}\delta_{f_{\boldsymbol{\theta}}(\boldsymbol{x}_i)}, \nu\right) = \min_{\boldsymbol{P}\in[0,1]^{n\times K},\, \mathbf{1}_n^\top \boldsymbol{P}=\mathbf{1}_K^\top/K,\, \boldsymbol{P}\mathbf{1}_K=\mathbf{1}_n/n}\sum_{i=1}^{n}\sum_{j=1}^{K}p_{ij}|f_{\boldsymbol{\theta}}(\boldsymbol{x}_i) - \boldsymbol{e}_j|.$$

It is natural to optimize with respect to $\boldsymbol{P}$ and $\boldsymbol{\theta}$ in an alternating manner. We apply random sampling techniques to speedup computation. In the $t$-th iteration,

1. Draw $B = 200$ samples $\{\boldsymbol{x}_{ti}\}_{i=1}^B$ uniform at random (with replacement) from the dataset;

2. Use the Python function `ot.sinkhorn2` in library POT (Flamary and Courty, 2017) with `reg = 0.1` to obtain the solution $\boldsymbol{P}_t$ to an entropy-regularized version of

$$\min_{\boldsymbol{P}\in[0,1]^{B\times K},\, \mathbf{1}_B^\top \boldsymbol{P}=\mathbf{1}_K^\top/K,\, \boldsymbol{P}\mathbf{1}_K=\mathbf{1}_B/B}\sum_{i=1}^{B}\sum_{j=1}^{K}p_{ij}|f_{\boldsymbol{\theta}_t}(\boldsymbol{x}_{ti}) - \boldsymbol{e}_j|;$$

Figure 1: 4-class Fashion-MNIST: Testing errors of linear functions and neural networks, with error bar quantifying one standard deviation.

3. Update model parameters by $\boldsymbol{\theta}_{t+1} = \boldsymbol{\theta}_t - \eta\partial L_t(\boldsymbol{\theta}_t)$, where $\partial$ is the sub-differential operator, $\eta = 10^{-3}$ and

$$L_t(\boldsymbol{\theta}) = \sum_{j=1}^{K} \hat{p}_{ij} |f_{\boldsymbol{\theta}}(\boldsymbol{x}_{ti}) - \boldsymbol{e}_j|, \qquad \forall \boldsymbol{\theta}.$$

An epoch refers to $n/B = 12$ consecutive iterations. The learning curves in Figure 1 shows the advantage of neural network and demonstrates the flexibility of CURE with nonlinear function classes.

## B  Proof sketch of Theorem 1

### B.1  Step 1: properties of the test function $f$

We now investigate the function $f$ defined in (8) and relate it to $h(x) = (x^2 - 1)^2/4$. As Lemma 1 suggests, $|f'|$, $|f''|$ and $|f'''|$ are all bounded by constants determined by $a$ and $b$; $|f' - h'|$ and $|f'' - h''|$ are bounded by polynomials that are independent of $a$ and $b$. See Appendix D for a proof.

**Lemma 1.** *When $a$ is sufficiently large and $b \geq 2a$, $f$ has the following properties:*

1. *$f'$ is continuous with $F_1 \triangleq \sup_{x\in\mathbb{R}} |f'(x)| \leq 2a^2 b$ and $|f'(x) - h'(x)| \leq 7|x|^3 \mathbf{1}_{\{|x|\geq a\}}$;*

2. *$f''$ is continuous with $F_2 \triangleq \sup_{x\in\mathbb{R}} |f''(x)| \leq 3a^2$ and $|f''(x) - h''(x)| \leq 9x^2 \mathbf{1}_{\{|x|\geq a\}}$;*

3. *$f'''$ exists in $\mathbb{R} \setminus \{\pm a, \pm b\}$ with $F_3 \triangleq \sup_{x\in\mathbb{R}\setminus\{\pm a,\pm b\}} |f'''(x)| \leq 6a$.*

### B.2  Step 2: landscape analysis of the population loss

To kick off the landscape analysis we investigate the population version of $\hat{L}_1$, namely

$$L_1(\alpha, \boldsymbol{\beta}) = \mathbb{E}_{\boldsymbol{X}\sim\rho} f(\alpha + \boldsymbol{\beta}^\top \boldsymbol{X}) + \frac{1}{2}(\alpha + \boldsymbol{\beta}^\top \boldsymbol{\mu}_0)^2. \tag{2}$$

One of the main obstacles is the complicated piecewise definition of $f$, which prevent us from obtaining closed form formulae. We bypass this problem by relating the population loss with $f$ to that with the quartic function $h$. See Appendix E for a proof.

**Theorem 1** (Landscape of the population loss). *Consider Model 1 and assume that $b \geq 2a$. There exist positive constants $A, \varepsilon, \delta$ and $\eta$ determined by $M$, $\mathbb{E}Z^4$, $\|\boldsymbol{\mu}\|_2$, $\lambda_{\max}(\boldsymbol{\Sigma})$ and $\lambda_{\min}(\boldsymbol{\Sigma})$ but independent of $d$ and $n$, such that when $a > A$,*

1. *The only two global minima of $L_1$ are $\pm\boldsymbol{\gamma}^\star$, where $\boldsymbol{\gamma}^\star = (-c\boldsymbol{\beta}^{h\top}\boldsymbol{\mu}_0, c\boldsymbol{\beta}^h)$ for some $c \in (1/2, 2)$ and*

$$\boldsymbol{\beta}^h = \left( \frac{1 + 1/\|\boldsymbol{\mu}\|_{\boldsymbol{\Sigma}^{-1}}^2}{\|\boldsymbol{\mu}\|_{\boldsymbol{\Sigma}^{-1}}^4 + 6\|\boldsymbol{\mu}\|_{\boldsymbol{\Sigma}^{-1}}^2 + M_Z} \right)^{1/2} \boldsymbol{\Sigma}^{-1}\boldsymbol{\mu};$$

2. $\|\nabla L_1(\boldsymbol{\gamma})\|_2 \geq \varepsilon$ if $\mathrm{dist}(\boldsymbol{\gamma}, \{\pm\boldsymbol{\gamma}^\star\} \cup S) \geq \delta$, where $S = \{\mathbf{0}\} \cup \{(-\boldsymbol{\beta}^\top\boldsymbol{\mu}_0, \boldsymbol{\beta}) : \boldsymbol{\mu}^\top\boldsymbol{\beta} = \mathbf{0}, \ \boldsymbol{\beta}^\top\boldsymbol{\Sigma}\boldsymbol{\beta} = 1/M_Z\}$;

3. $\nabla^2 L_1(\boldsymbol{\gamma}) \succeq \eta\boldsymbol{I}$ if $\mathrm{dist}(\boldsymbol{\gamma}, \{\pm\boldsymbol{\gamma}^\star\}) \leq \delta$, and $\boldsymbol{u}^\top\nabla^2 L_1(\boldsymbol{\gamma})\boldsymbol{u} \leq -\eta$ if $\mathrm{dist}(\boldsymbol{\gamma}, S) \leq \delta$ with $\boldsymbol{u} = (0, \boldsymbol{\Sigma}^{-1}\boldsymbol{\mu}/\|\boldsymbol{\Sigma}^{-1}\boldsymbol{\mu}\|_2)$.

Theorem 1 precisely characterizes the landscape of $L_1$. In particular, all of its critical points make up the set $\{\pm\boldsymbol{\gamma}^\star\} \cup S$, where $\pm\boldsymbol{\gamma}^\star$ are global minima and $S$ consists of strict saddles. The local geometry around critical points is also desirable.

### B.3 Step 3: landscape analysis of the empirical loss

Based on geometric properties of the population loss $L_1$, we establish similar results for the empirical loss $\hat{L}_1$ through concentration analysis. See Appendix F for a proof.

**Theorem 2** (Landscape of the empirical loss). *Consider Model 1 and assume that $b \geq 2a \geq 4$. Let $\boldsymbol{\gamma}^\star$ and $S$ be defined as in Theorem 1. There exist positive constants $A, C_0, C_1, C_2, M_1, \varepsilon, \delta$ and $\eta$ determined by $M, M_Z, \|\boldsymbol{\mu}\|_2, \lambda_{\max}(\boldsymbol{\Sigma})$ and $\lambda_{\min}(\boldsymbol{\Sigma})$ but independent of $d$ and $n$, such that when $a \geq A$ and $n \geq C_0 d$, the followings hold with probability exceeding $1 - C_1(d/n)^{C_2 d} - C_1\exp(-C_2 n^{1/3})$:*

1. $\|\nabla\hat{L}_1(\boldsymbol{\gamma})\|_2 \geq \varepsilon$ if $\mathrm{dist}(\boldsymbol{\gamma}, \{\pm\boldsymbol{\gamma}^\star\} \cup S) \geq \delta$;

2. $\boldsymbol{u}^\top\nabla^2\hat{L}_1(\boldsymbol{\gamma})\boldsymbol{u} \leq -\eta$ if $\mathrm{dist}(\boldsymbol{\gamma}, S) \leq \delta$, with $\boldsymbol{u} = (0, \boldsymbol{\Sigma}^{-1}\boldsymbol{\mu}/\|\boldsymbol{\Sigma}^{-1}\boldsymbol{\mu}\|_2)$;

3. $\|\nabla\hat{L}_1(\boldsymbol{\gamma}_1) - \nabla\hat{L}_1(\boldsymbol{\gamma}_2)\|_2 \leq M_1\|\boldsymbol{\gamma}_1 - \boldsymbol{\gamma}_2\|_2$ and $\|\nabla^2\hat{L}_1(\boldsymbol{\gamma}_1) - \nabla^2\hat{L}_1(\boldsymbol{\gamma}_2)\|_2 \leq M_1[1 \vee (d\log(n/d)/\sqrt{n})]\|\boldsymbol{\gamma}_1 - \boldsymbol{\gamma}_2\|_2$ hold for all $\boldsymbol{\gamma}_1, \boldsymbol{\gamma}_2 \in \mathbb{R} \times \mathbb{R}^d$.

Theorem 2 shows that a sample of size $n \gtrsim d$ suffices for the empirical loss to inherit nice geometric properties from its population counterpart. The corollary below illustrates that as long as we can find an approximate second-order stationary point, then the statistical estimation error can be well controlled by the gradient. We defer the proof of this to Appendix G.

**Corollary 1.** *Under the settings in Theorem 2, there exist constants $C, C_1', C_2'$ such that the followings happen with probability exceeding $1 - C_1'(d/n)^{C_2' d} - C_1'\exp(-C_2' n^{1/3})$: for any $\boldsymbol{\gamma} \in \mathbb{R} \times \mathbb{R}^d$ satisfying $\|\nabla\hat{L}_1(\boldsymbol{\gamma})\|_2 \leq \varepsilon$ and $\lambda_{\min}[\nabla^2\hat{L}_1(\boldsymbol{\gamma})] > -\eta$,*

$$\min_{s=\pm 1}\|s\boldsymbol{\gamma} - \boldsymbol{\gamma}^\star\|_2 \leq C\left(\left\|\nabla\hat{L}_1(\boldsymbol{\gamma})\right\|_2 + \sqrt{\frac{d}{n}\log\left(\frac{n}{d}\right)}\right).$$

*As a result, when the event above happens, any local minimizer $\tilde{\boldsymbol{\gamma}}$ of $\hat{L}_1$ satisfies*

$$\min_{s=\pm 1}\|s\tilde{\boldsymbol{\gamma}} - \boldsymbol{\gamma}^\star\|_2 \leq C\sqrt{\frac{d}{n}\log\left(\frac{n}{d}\right)}.$$

### B.4 Step 4: convergence guarantees for perturbed gradient descent

The landscape analysis above shows that all local minimizers of $\hat{L}_1$ are statistically optimal (up to logarithmic factors), and all saddle points are non-degenerate. Then it boils down to finding any $\boldsymbol{\gamma}$ whose gradient size is sufficiently small and Hessian has no significantly negative eigenvalue. Thanks to the Lipschitz smoothness of $\nabla\hat{L}_1$ and $\nabla^2\hat{L}_1$, this can be efficiently achieved by the perturbed gradient descent algorithm (see Algorithm 1) proposed by Jin et al. (2017). Small perturbation is occasionally added to the iterates, helping escape from saddle points efficiently and thus converge towards local minimizers. Theorem 3 provides algorithmic guarantees for CURE on top of that. We defer the proof to Appendix H.

Implementation of the algorithm requires specification of hyperparameters $a$, $b$, $M_1$, $\varepsilon$ and $\eta$. Under the regularity assumptions in Model 1, many structural parameters are well-behaved constants and that helps choose hyperparameters at least in a conservative way. In theory, we can let $b = 2a$; $a$ and $M_1$ be sufficiently large; $\varepsilon$ and $\eta$ be sufficiently small. In our numerical experiments, the algorithm does not appear to be sensitive to choices of hyperparameters. We do not go into much details to avoid distractions.

**Algorithm 1** Perturbed gradient descent   PerturbedGD($\boldsymbol{\gamma}_{\mathrm{pgd}}, \ell, \rho, \varepsilon_{\mathrm{pgd}}, c_{\mathrm{pgd}}, \delta_{\mathrm{pgd}}, \Delta_{\mathrm{pgd}}$)

---

$\chi \leftarrow 3 \max\{\log(d\ell\Delta_{\mathrm{pgd}}/(c_{\mathrm{pgd}}\varepsilon_{\mathrm{pgd}}^2\delta_{\mathrm{pgd}})), 4\}$, $\eta_{\mathrm{pgd}} \leftarrow c_{\mathrm{pgd}}/\ell$, $r \leftarrow \sqrt{c_{\mathrm{pgd}}}\varepsilon_{\mathrm{pgd}}/(\chi^2\ell)$, $g_{\mathrm{thres}} \leftarrow \sqrt{c_{\mathrm{pgd}}}\varepsilon_{\mathrm{pgd}}/\chi^2$, $f_{\mathrm{thres}} \leftarrow c_{\mathrm{pgd}}\varepsilon_{\mathrm{pgd}}^{1.5}/(\chi^3\sqrt{\rho})$, $t_{\mathrm{thres}} \leftarrow \chi\ell/(c_{\mathrm{pgd}}^2\sqrt{\rho\varepsilon_{\mathrm{pgd}}})$, $t_{\mathrm{noise}} \leftarrow -t_{\mathrm{thres}} - 1$.
**Initialize** $\boldsymbol{\gamma}^0 = \boldsymbol{\gamma}_{\mathrm{pgd}}$.
**For** $t = 0, 1, \ldots$ **do**
   **If** $\|\nabla\hat{L}_1(\boldsymbol{\gamma}^t)\|_2 \leq g_{\mathrm{thres}}$ **and** $t - t_{\mathrm{noise}} > t_{\mathrm{thres}}$:
      Update $t_{\mathrm{noise}} \leftarrow t$,
      Perturb $\boldsymbol{\gamma}^t \leftarrow \boldsymbol{\gamma}^t + \boldsymbol{\xi}^t$ with $\boldsymbol{\xi}^t \sim \mathcal{U}(B(\mathbf{0}, r))$
   **If** $t - t_{\mathrm{noise}} = t_{\mathrm{thres}}$ **and** $\hat{L}_1(\boldsymbol{\gamma}^t) - \hat{L}_1(\tilde{\boldsymbol{\gamma}}^{t_{\mathrm{noise}}}) > -f_{\mathrm{thres}}$:
      **Return** $\tilde{\boldsymbol{\gamma}}^{t_{\mathrm{noise}}}$
   **Update** $\boldsymbol{\gamma}^{t+1} \leftarrow \boldsymbol{\gamma}^t - \eta_{\mathrm{pgd}}\nabla\hat{L}_1(\boldsymbol{\gamma}^t)$.

---

**Theorem 3** (Algorithmic guarantees). *Consider the settings in Theorem 2 and adopt the constants $M_1$, $\varepsilon$ and $\eta$ therein. With probability exceeding $1 - C_1[(d/n)^{C_2 d} + e^{-C_2 n^{1/3}} + n^{-10}]$, Algorithm 1 with parameters $\boldsymbol{\gamma}_{\mathrm{pgd}} = \mathbf{0}$, $\ell = M_1$, $\delta_{\mathrm{pgd}} = n^{-11}$, $\rho = M_1\max\{1, d\log(n/d)/\sqrt{n}\}$, $\varepsilon_{\mathrm{pgd}} = \min\{\sqrt{d\log(n/d)/n}, \ell^2/\rho, \eta^2/\rho, \varepsilon\}$ and $\Delta_{\mathrm{pgd}} = 1/4$ terminates within $\tilde{O}(n/d + d^2/n)$ iterations and the output $\hat{\boldsymbol{\gamma}}$ satisfies*

$$\left\|\nabla\hat{L}_1(\hat{\boldsymbol{\gamma}})\right\|_2 \leq \sqrt{\frac{d}{n}\log\left(\frac{n}{d}\right)} \leq \varepsilon \qquad \textit{and} \qquad \lambda_{\min}\left(\nabla^2\hat{L}_1(\hat{\boldsymbol{\gamma}})\right) \geq -\eta.$$

Theorem 3 and Corollary 1 immediately lead to

$$\min_{s=\pm 1}\|s\hat{\boldsymbol{\gamma}} - \boldsymbol{\gamma}^\star\|_2 \lesssim \left\|\nabla\hat{L}_1(\hat{\boldsymbol{\gamma}})\right\|_2 + \sqrt{\frac{d}{n}\log\left(\frac{n}{d}\right)} \lesssim \sqrt{\frac{d}{n}\log\left(\frac{n}{d}\right)},$$

which finishes the proof of Theorem 1.

## C  Preliminaries

Before we start the proof, let us introduce some notations. Recall the definition of the random vector $\boldsymbol{X} = \boldsymbol{\mu}_0 + \boldsymbol{\mu}Y + \boldsymbol{\Sigma}^{1/2}\boldsymbol{Z}$ and the i.i.d. samples $\boldsymbol{X}_1, \ldots, \boldsymbol{X}_n \in \mathbb{R}^d$. Let $\bar{\boldsymbol{X}} = (1, \boldsymbol{X})$, $\bar{\boldsymbol{X}}_i = (1, \boldsymbol{X}_i)$ and $\bar{\boldsymbol{\mu}}_0 = (1, \boldsymbol{\mu}_0)$. For any $\boldsymbol{\gamma} = (\alpha, \boldsymbol{\beta}) \in \mathbb{R} \times \mathbb{R}^d$, define

$$L_\lambda(\boldsymbol{\gamma}) = L(\boldsymbol{\gamma}) + \lambda R(\boldsymbol{\gamma}) \qquad \text{and} \qquad \hat{L}_\lambda(\boldsymbol{\gamma}) = \hat{L}(\boldsymbol{\gamma}) + \lambda\hat{R}(\boldsymbol{\gamma}),$$

where

$$L(\boldsymbol{\gamma}) = \mathbb{E}f(\boldsymbol{\gamma}^\top\bar{\boldsymbol{X}}) = \mathbb{E}f(\alpha + \boldsymbol{\beta}^\top\boldsymbol{X}), \qquad \hat{L}(\boldsymbol{\gamma}) = \frac{1}{n}\sum_{i=1}^n f(\boldsymbol{\gamma}^\top\bar{\boldsymbol{X}}_i) = \frac{1}{n}\sum_{i=1}^n f(\alpha + \boldsymbol{\beta}^\top\boldsymbol{X}_i),$$

$$R(\boldsymbol{\gamma}) = \frac{1}{2}(\alpha + \boldsymbol{\beta}^\top\boldsymbol{\mu}_0)^2 = \frac{1}{2}(\boldsymbol{\gamma}^\top\bar{\boldsymbol{\mu}}_0)^2, \qquad \hat{R}(\boldsymbol{\gamma}) = \frac{1}{2}(\alpha + \boldsymbol{\beta}^\top n^{-1}\sum_{i=1}^n \boldsymbol{X}_i)^2 = \frac{1}{2}(\boldsymbol{\gamma}^\top n^{-1}\sum_{i=1}^n \bar{\boldsymbol{X}}_i)^2.$$

Note that the results stated in Section 3 and B focus on the special case when $\lambda = 1$. The proof in the appendices allows for general choices of $\lambda \geq 1$.

## D Proof of Lemma 1

By direct calculation, one has

$$f'(x) = \begin{cases} h'(x), & |x| \le a \\ [h'(a) + h''(a)(|x| - a) - \frac{h''(a)}{2(b-a)}(|x| - a)^2]\operatorname{sgn}(x), & a < |x| \le b \\ [h'(a) + \frac{b-a}{2}h''(a)]\operatorname{sgn}(x), & |x| > b \end{cases},$$

$$f''(x) = \begin{cases} h''(x), & |x| \le a \\ h''(a)(1 - \frac{|x|-a}{b-a}), & a < |x| \le b \\ 0, & |x| > b. \end{cases},$$

$$f'''(x) = \begin{cases} h'''(x), & |x| < a \\ -\frac{h''(a)}{b-a}\operatorname{sgn}(x), & a < |x| < b \\ 0, & |x| > b \end{cases}.$$

When $a$ is sufficiently large and $b \ge 2a$, we have $F_1 \triangleq \sup_{x \in \mathbb{R}} |f'(x)| = h'(a) + \frac{b-a}{2}h''(a) \le 2a^2 b$, $F_2 \triangleq \sup_{x \in \mathbb{R}} |f''(x)| = h''(a) \le 3a^2$, and $F_3 \triangleq \sup_{|x| \ne a,b} |f'''(x)| = h'''(a) \vee \frac{h''(a)}{b-a} \le 6a$.

In addition, one can also check that when $a < |x| \le b$, we have $|h'(a)| \le |x|^3$ and $|h''(a)| \le 3|x|^2$, thus

$$|f'(x) - h'(x)| \le |f'(x)| + |h'(x)| \le |h'(a)| + |h''(a)|(|x| - a)| + |h''(a)|(|x| - a)^2/(2a)| + |x^3 - x|$$

$$\le |x|^3 + 3|x|^2 + \frac{3}{2}|x|^2 + |x|^3 \le 7|x|^3$$

provided that $b \ge 2a \ge 2$. When $|x| \ge b$, we have

$$|f'(x) - h'(x)| \le |f'(x)| + |h'(x)| \le |h'(a)| + |(b-a)h''(a)/2| + |x^3 - x|$$

$$\le |x|^3 + \frac{3}{2}|x|^2 + |x|^3 \le 4|x|^3.$$

This combined with $f'(x) = h'(x)$ when $|x| \le a$ gives $|f'(x) - h'(x)| \le \mathbf{1}_{\{|x| \ge a\}} 7|x|^3$. Similarly we have $|f''(x) - h''(x)| \le \mathbf{1}_{\{|x| \ge a\}} 9x^2$.

## E Proof of Theorem 1

It suffices to focus on the special case $\boldsymbol{\mu}_0 = \mathbf{0}$ and $\boldsymbol{\Sigma} = \boldsymbol{I}_d$. We first give a theorem that characterizes the landscape of an auxiliary population loss, which serves as a nice starting point of the study of the actual loss functions that we use.

**Theorem 4** (Landscape of the auxillary population loss). *Consider model* (1) *with* $\boldsymbol{\mu}_0 = \mathbf{0}$ *and* $\boldsymbol{\Sigma} = \boldsymbol{I}_d$. *Suppose that* $M_Z > 3$. *Let* $h(x) = (x^2 - 1)^2/4$ *and* $\lambda \ge 1$. *The stationary points of the population loss*

$$L_\lambda^h(\alpha, \boldsymbol{\beta}) = \mathbb{E}h\left(\alpha + \boldsymbol{\beta}^\top \boldsymbol{X}\right) + \frac{\lambda}{2}\alpha^2$$

*are* $\{(\alpha, \boldsymbol{\beta}) : \nabla L_\lambda^h(\alpha, \boldsymbol{\beta}) = \mathbf{0}\} = S_1^h \cup S_2^h$, *where*

*1.* $S_1^h = \{(0, \pm\boldsymbol{\beta}^h)\}$ *consists of global minima, with*

$$\boldsymbol{\beta}^h = \left(\frac{1 + 1/\|\boldsymbol{\mu}\|_2^2}{\|\boldsymbol{\mu}\|_2^4 + 6\|\boldsymbol{\mu}\|_2^2 + M_Z}\right)^{1/2} \boldsymbol{\mu};$$

*2.* $S_2^h = \{(0, \boldsymbol{\beta}) : \boldsymbol{\mu}^\top \boldsymbol{\beta} = 0, \|\boldsymbol{\beta}\|_2^2 = 1/M_Z\} \cup \{\mathbf{0}\}$ *consists of saddle points whose Hessians have negative eigenvalues.*

*We also have the following quantitative results: there exist positive constants* $\varepsilon^h, \delta^h$ *and* $\eta^h$ *determined by* $M_Z, \|\boldsymbol{\mu}\|_2$ *and* $\lambda$ *such that*

1. $\|\nabla L_\lambda^h(\gamma)\|_2 \geq \varepsilon^h$ if $\mathrm{dist}(\gamma, S_1^h \cup S_2^h) \geq \delta^h$;

2. $\nabla^2 L_\lambda^h(\gamma) \succeq \eta^h \boldsymbol{I}$ if $\mathrm{dist}(\gamma, S_1^h) \leq 3\delta^h$, and $\boldsymbol{u}^\top \nabla^2 L_\lambda^h(\gamma) \boldsymbol{u} \leq -\eta^h$ if $\mathrm{dist}(\gamma, S_2^h) \leq 3\delta^h$ where $\boldsymbol{u} = (0, \boldsymbol{\mu}/\|\boldsymbol{\mu}\|_2)$.

*Proof.* See Appendix E.1. □

The following Lemma 2 controls the difference between the landscape of $L_\lambda$ and $L_\lambda^h$ within a compact ball.

**Lemma 2.** *Let $\boldsymbol{X}$ be a random vector in $\mathbb{R}^{d+1}$ with $\|\boldsymbol{X}\|_{\psi_2} \leq M$, $f$ be defined in (8) with $b \geq 2a \geq 4$, $h(x) = (x^2 - 1)^2/4$ for $x \in \mathbb{R}$, $L_\lambda(\gamma) = \mathbb{E}f(\gamma^\top \boldsymbol{X}) + \lambda\alpha^2/2$ and $L_\lambda^h(\gamma) = \mathbb{E}h(\gamma^\top \boldsymbol{X}) + \lambda\alpha^2/2$ for $\gamma \in \mathbb{R}^{d+1}$. There exist constants $C_1, C_2 > 0$ such that for any $R > 0$,*

$$\sup_{\|\gamma\|_2 \leq R} \left\|\nabla L_\lambda(\gamma) - \nabla L_\lambda^h(\gamma)\right\|_2 \leq C_2 R^3 M^4 \exp\left(-\frac{C_1 a^2}{R^2 M^2}\right),$$

$$\sup_{\|\gamma\|_2 \leq R} \left\|\nabla^2 L_\lambda(\gamma) - \nabla^2 L_\lambda^h(\gamma)\right\|_2 \leq C_2 R^2 M^4 \exp\left(-\frac{C_1 a^2}{R^2 M^2}\right).$$

*In addition, when $\mathbb{E}(\boldsymbol{X}\boldsymbol{X}^\top) \succeq \sigma^2 \boldsymbol{I}$ holds for some $\sigma > 0$, there exists $m > 0$ determined by $M$ and $\sigma$ such that $\inf_{\|\gamma\|_2 \geq 3/m} \|\nabla L_\lambda(\gamma)\|_2 \geq m$ and $\inf_{\|\gamma\|_2 \geq 3/m} \|\nabla L_\lambda^h(\gamma)\|_2 \geq m$.*

*Proof.* See Appendix E.2. □

On the one hand, Lemma 2 implies that $\inf_{\|\gamma\|_2 \geq 3/m} \|\nabla L_\lambda(\gamma)\|_2 \geq m$ for some constant $m > 0$. Suppose that

$$\varepsilon^h < m \tag{3}$$

and define $r = 3/\varepsilon^h$. Then

$$\|\nabla L_1(\gamma)\|_2 > \varepsilon^h \qquad \text{if} \qquad \|\gamma\|_2 \geq r. \tag{4}$$

Moreover, we can take $a$ to be sufficiently large such that

$$\sup_{\|\gamma\|_2 \leq r} \left\|\nabla L_1(\gamma) - \nabla L_1^h(\gamma)\right\|_2 \leq \varepsilon^h/2. \tag{5}$$

On the other hand, from Theorem 4 we know that

$$\|\nabla L_\lambda^h(\gamma)\|_2 \geq \varepsilon^h \qquad \text{if} \qquad \mathrm{dist}(\gamma, S_1^h \cup S_2^h) \geq \delta^h. \tag{6}$$

Taking (4), (5) and (6) collectively gives

$$\|\nabla L_\lambda(\gamma)\|_2 \geq \varepsilon^h/2 \qquad \text{if} \qquad \mathrm{dist}(\gamma, S_1^h \cup S_2^h) \geq \delta^h. \tag{7}$$

Hence $\{\gamma : \nabla L_\lambda(\gamma) = \boldsymbol{0}\} \subseteq \{\gamma : \mathrm{dist}(\gamma, S_1^h \cup S_2^h) \leq \delta^h\}$ and it yields a decomposition $\{\gamma : \nabla L_\lambda(\gamma) = \boldsymbol{0}\} = S_1 \cup S_2$, where

$$S_j \subseteq \{\gamma : \mathrm{dist}(\gamma, S_j^h) \leq \delta^h\}, \qquad \forall j = 1, 2. \tag{8}$$

Consequently, for $j = 1, 2$ we have

$$\{\gamma : \mathrm{dist}(\gamma, S_j) \leq 2\delta^h\} \subseteq \{\gamma : \mathrm{dist}(\gamma, S_j^h) \leq 3\delta^h\} \subseteq \{\gamma : \|\gamma\|_2 \leq 3\delta^h + \max_{\gamma' \in S_1^h \cup S_2^h} \|\gamma'\|_2\}. \tag{9}$$

**Now we work on the first proposition in Theorem 1 by characterizing $S_1$.**

**Lemma 3.** *Consider the model in (1) with $\boldsymbol{\mu}_0 = \boldsymbol{0}$ and $\boldsymbol{\Sigma} = \boldsymbol{I}_d$. Suppose that $f \in C^2(\mathbb{R})$ is even, $\lim_{x \to +\infty} xf'(x) = +\infty$ and $f''(0) < 0$. Define*

$$L_\lambda(\alpha, \boldsymbol{\beta}) = \mathbb{E}f(\alpha + \boldsymbol{\beta}^\top \boldsymbol{X}) + \frac{\lambda}{2}\alpha^2, \qquad \forall \alpha \in \mathbb{R}, \ \boldsymbol{\beta} \in \mathbb{R}^d.$$

1. *There exists some $c > 0$ determined by $\|\boldsymbol{\mu}\|_2$, the function $f$, and the distribution of $Z$, such that $(0, \pm c\boldsymbol{\mu})$ are critical points of $L_\lambda$;*

2. *In addition, if $f''$ is piecewise differentible and $|f'''(x)| \le F_3 < \infty$ almost everywhere, we can find $c_0 > 0$ determined by $\|\boldsymbol{\mu}\|_2$, $f''(0)$, $F_3$ and $M$ such that $c > c_0$.*

*Proof.* See Appendix E.3. $\qquad\qquad\qquad\qquad\qquad\qquad\qquad\qquad\qquad\qquad\qquad\qquad\quad$ $\square$

Lemma 3 asserts the existence of two critical points $\pm\boldsymbol{\gamma}^\star = (0, \pm c\boldsymbol{\beta}^h)$ of $L_1$, for some $c$ bounded from below by a constant $c_0 > 0$. If

$$\delta^h < c_0 \|\boldsymbol{\beta}^h\|_2/4, \tag{10}$$

then the property of $S_2^h$ forces

$$\operatorname{dist}(\pm\boldsymbol{\gamma}^\star, S_2^h) \ge \|\boldsymbol{\gamma}^\star\|_2 = c\|\boldsymbol{\beta}^h\|_2 \ge c_0\|\boldsymbol{\beta}^h\|_2 > 4\delta^h > 3\delta^h. \tag{11}$$

It is easily seen from (9) with $j = 2$ that $\operatorname{dist}(\pm\boldsymbol{\gamma}^\star, S_2) > 2\delta^h$ and $\pm\boldsymbol{\gamma}^\star \notin S_2$. Then $\{\boldsymbol{\gamma} : \nabla L_1(\boldsymbol{\gamma}) = \boldsymbol{0}\} = S_1 \cup S_2$ forces

$$\{\boldsymbol{\gamma}^\star, -\boldsymbol{\gamma}^\star\} \subseteq S_1. \tag{12}$$

Let us investigate the curvature near $S_1$. Lemma 2 and (9) with $j = 1$ allow us to take $a$ to be sufficiently large such that

$$\sup_{\operatorname{dist}(\boldsymbol{\gamma}, S_1) \le 2\delta^h} \left\| \nabla^2 L_\lambda(\boldsymbol{\gamma}) - \nabla^2 L_\lambda^h(\boldsymbol{\gamma}) \right\|_2 \le \eta^h/2. \tag{13}$$

Theorem 4 asserts that $\nabla^2 L_\lambda^h(\boldsymbol{\gamma}) \succeq \eta^h \boldsymbol{I}$ if $\operatorname{dist}(\boldsymbol{\gamma}, S_1^h) \le 3\delta^h$. By this, (9) with $j = 1$ and (13),

$$\nabla^2 L_\lambda(\boldsymbol{\gamma}) \succeq (\eta^h/2)\boldsymbol{I} \qquad \text{if} \qquad \operatorname{dist}(\boldsymbol{\gamma}, S_1) \le 2\delta^h. \tag{14}$$

Hence $L_1$ is strongly convex in $\{\boldsymbol{\gamma} : \operatorname{dist}(\boldsymbol{\gamma}, S_1) \le 2\delta^h\}$. Combined with (12), it leads to $S_1 = \{\pm\boldsymbol{\gamma}^\star\}$, and both points therein are local minima.

Let $\boldsymbol{\gamma}^h = (0, \boldsymbol{\beta}^h)$. The fact $S_1^h = \{\pm\boldsymbol{\gamma}^h\}$ and (8) yields

$$|c - 1| \cdot \|\boldsymbol{\beta}^h\|_2 = \|\boldsymbol{\gamma}^\star - \boldsymbol{\gamma}^h\|_2 = \operatorname{dist}(\boldsymbol{\gamma}^\star, S_1^h) \le \delta^h. \tag{15}$$

When

$$\delta^h < \|\boldsymbol{\beta}^h\|_2/2, \tag{16}$$

we have $1/2 < c < 3/2$ as claimed. The global optimality of $\pm\boldsymbol{\gamma}^\star$ is obvious. Without loss of generality, in Theorem 4 we can always take $\delta^h < \|\boldsymbol{\beta}^h\|_2 \min\{c_0/3, 1/2\}$ and then find $\varepsilon^h < m$. In that case, (3), (10) and (16) imply the first proposition in Theorem 1.

**Next, we study the second proposition in Theorem 1.** Let $S = S_2^h$. Given $S_1 = \{\pm\boldsymbol{\gamma}^h\}$ and $S_1 = \{\pm\boldsymbol{\gamma}^\star\}$, from (15) we know that $\operatorname{dist}(\boldsymbol{\gamma}, \{\pm\boldsymbol{\gamma}^\star\} \cup S) \ge 2\delta^h$ implies $\operatorname{dist}(\boldsymbol{\gamma}, S_1^h \cup S_2^h) \ge 2\delta^h$. This combined with (7) immediately gives

$$\|\nabla L_\lambda(\boldsymbol{\gamma})\|_2 \ge \varepsilon^h/2 \qquad \text{if} \qquad \operatorname{dist}(\boldsymbol{\gamma}, \{\pm\boldsymbol{\gamma}^\star\} \cup S) \ge 2\delta^h.$$

Hence the second proposition in Theorem 1 holds if

$$\varepsilon = \varepsilon^h/2 \qquad \text{and} \qquad \delta = 2\delta^h. \tag{17}$$

**Finally, we study the third proposition in Theorem 1.** By (14), the first part of that proposition holds when

$$\eta = \eta^h/2 \qquad \text{and} \qquad \delta = 2\delta^h. \tag{18}$$

It remains to prove the second part. Lemma 2 and (9) with $j = 2$ allow us to take $a$ to be sufficiently large such that

$$\sup_{\operatorname{dist}(\boldsymbol{\gamma}, S) \le 3\delta^h} \left\| \nabla^2 L_\lambda(\boldsymbol{\gamma}) - \nabla^2 L_\lambda^h(\boldsymbol{\gamma}) \right\|_2 \le \eta^h/2. \tag{19}$$

Theorem 4 asserts that $\boldsymbol{u}^\top \nabla^2 L_\lambda^h(\boldsymbol{\gamma})\boldsymbol{u} \le -\eta^h$ for $\boldsymbol{u} = (0, \boldsymbol{\mu}/\|\boldsymbol{\mu}\|_2)$ if $\operatorname{dist}(\boldsymbol{\gamma}, S) \le 3\delta^h$. By this, (9) with $j = 2$ and (19),

$$\nabla^2 L_\lambda(\boldsymbol{\gamma}) \le -\eta^h/2 \qquad \text{if} \qquad \operatorname{dist}(\boldsymbol{\gamma}, S) \le 3\delta^h. \tag{20}$$

Hence (17) suffice for the second part of the third proposition to hold.

According to (17) and (18), Theorem 1 holds with $\varepsilon = \varepsilon^h/2$, $\delta = 2\delta^h$ and $\eta = \eta^h/2$.

## E.1 Proof of Theorem 4

### E.1.1 Part 1: Characterization of stationary points

Note that

$$\nabla L_\lambda^h(\alpha, \boldsymbol{\beta}) = \mathbb{E}\left[\begin{pmatrix} 1 \\ \boldsymbol{X} \end{pmatrix} h'(\alpha + \boldsymbol{\beta}^\top \boldsymbol{X})\right] + \begin{pmatrix} \lambda \\ \mathbf{0} \end{pmatrix}$$

$$= \begin{pmatrix} \mathbb{E}h'(\alpha + \boldsymbol{\beta}^\top \boldsymbol{X}) + \lambda \\ \mathbf{0} \end{pmatrix} + \begin{pmatrix} 0 \\ \mathbb{E}[Y h'(\alpha + \boldsymbol{\beta}^\top \boldsymbol{X})]\boldsymbol{\mu} \end{pmatrix} + \begin{pmatrix} 0 \\ \mathbb{E}[\boldsymbol{Z} h'(\alpha + \boldsymbol{\beta}^\top \boldsymbol{X})] \end{pmatrix}.$$

Now we will expand individual expected values in this sum. For the first term,

$$\mathbb{E}h'(\alpha + \boldsymbol{\beta}^\top \boldsymbol{X}) = \mathbb{E}(\alpha + \boldsymbol{\beta}^\top \boldsymbol{\mu} Y + \boldsymbol{\beta}^\top \boldsymbol{Z})^3 - \mathbb{E}(\alpha + \boldsymbol{\beta}^\top \boldsymbol{\mu} Y + \boldsymbol{\beta}^\top \boldsymbol{Z})$$

$$= \alpha^3 + 3\alpha \mathbb{E}(\boldsymbol{\beta}^\top \boldsymbol{\mu} Y)^2 + 3\alpha \mathbb{E}(\boldsymbol{\beta}^\top \boldsymbol{Z})^2 + \mathbb{E}(\boldsymbol{\beta}^\top \boldsymbol{\mu} Y + \boldsymbol{\beta}^\top \boldsymbol{Z})^3 - \alpha$$

$$= \alpha[\alpha^2 + 3(\boldsymbol{\beta}^\top \boldsymbol{\mu})^2 + 3\|\boldsymbol{\beta}\|_2^2 - 1],$$

where the first line follows since $h'(x) = x^3 - x$, the other two follows from $\mathbb{E}(\boldsymbol{Z}\boldsymbol{Z}^\top) = \boldsymbol{I}$ plus the fact that $Y$ and $\boldsymbol{Z}$ are independent, with zero odd moments due to their symmetry.

Using similar arguments,

$$\mathbb{E}[Y h'(\alpha + \boldsymbol{\beta}^\top \boldsymbol{X})] = \mathbb{E}[Y(\alpha + \boldsymbol{\beta}^\top \boldsymbol{\mu} Y + \boldsymbol{\beta}^\top \boldsymbol{Z})^3] - \mathbb{E}[Y(\alpha + \boldsymbol{\beta}^\top \boldsymbol{\mu} Y + \boldsymbol{\beta}^\top \boldsymbol{Z})]$$

$$= 3\alpha^2 \mathbb{E}\left[Y(\boldsymbol{\beta}^\top \boldsymbol{\mu} Y + \boldsymbol{\beta}^\top \boldsymbol{Z})\right] + \mathbb{E}[Y(\boldsymbol{\beta}^\top \boldsymbol{\mu} Y + \boldsymbol{\beta}^\top \boldsymbol{Z})^3] - \boldsymbol{\beta}^\top \boldsymbol{\mu}$$

$$= 3\alpha^2 \boldsymbol{\beta}^\top \boldsymbol{\mu} + \mathbb{E}[Y(\boldsymbol{\beta}^\top \boldsymbol{\mu} Y)^3] + 3\mathbb{E}[Y(\boldsymbol{\beta}^\top \boldsymbol{\mu} Y)]\mathbb{E}[(\boldsymbol{\beta}^\top \boldsymbol{Z})^2] - \boldsymbol{\beta}^\top \boldsymbol{\mu}$$

$$= \left[3\alpha^2 + (\boldsymbol{\beta}^\top \boldsymbol{\mu})^2 \mathbb{E}Y^4 + 3\|\boldsymbol{\beta}\|_2^2 - 1\right] \boldsymbol{\beta}^\top \boldsymbol{\mu}.$$

To work on $\mathbb{E}[\boldsymbol{Z} h'(\alpha + \boldsymbol{\beta}^\top \boldsymbol{X})] = \mathbb{E}[\boldsymbol{Z} h'(\alpha + \boldsymbol{\beta}^\top \boldsymbol{\mu} Y + \boldsymbol{\beta}^\top \boldsymbol{Z})]$, we define $\bar{\boldsymbol{\beta}} = \boldsymbol{\beta}/\|\boldsymbol{\beta}\|_2$ for $\boldsymbol{\beta} \neq \mathbf{0}$ and $\bar{\boldsymbol{\beta}} = \mathbf{0}$ otherwise. Observe that $(Y, \bar{\boldsymbol{\beta}}\bar{\boldsymbol{\beta}}^\top \boldsymbol{Z}, (\boldsymbol{I} - \bar{\boldsymbol{\beta}}\bar{\boldsymbol{\beta}}^\top)\boldsymbol{Z})$ and $(Y, \bar{\boldsymbol{\beta}}\bar{\boldsymbol{\beta}}^\top \boldsymbol{Z}, -(\boldsymbol{I} - \bar{\boldsymbol{\beta}}\bar{\boldsymbol{\beta}}^\top)\boldsymbol{Z})$ have exactly the same joint distribution. As a result,

$$\mathbb{E}[(\boldsymbol{I} - \bar{\boldsymbol{\beta}}\bar{\boldsymbol{\beta}}^\top)\boldsymbol{Z} h'(\alpha + \boldsymbol{\beta}^\top \boldsymbol{X})] = \mathbb{E}[(\boldsymbol{I} - \bar{\boldsymbol{\beta}}\bar{\boldsymbol{\beta}}^\top)\boldsymbol{Z} h'(\alpha + \boldsymbol{\beta}^\top \boldsymbol{\mu} Y + \boldsymbol{\beta}^\top \boldsymbol{Z})] = \mathbf{0}.$$

Hence,

$$\mathbb{E}[\boldsymbol{Z} h'(\boldsymbol{\beta}^\top \boldsymbol{X})] = \mathbb{E}[\bar{\boldsymbol{\beta}}\bar{\boldsymbol{\beta}}^\top \boldsymbol{Z} h'(\alpha + \boldsymbol{\beta}^\top \boldsymbol{X})] = \mathbb{E}[\bar{\boldsymbol{\beta}}^\top \boldsymbol{Z} h'(\alpha + \boldsymbol{\beta}^\top \boldsymbol{\mu} Y + \boldsymbol{\beta}^\top \boldsymbol{Z})]\bar{\boldsymbol{\beta}}$$

$$= \mathbb{E}[\bar{\boldsymbol{\beta}}^\top \boldsymbol{Z}(\alpha + \boldsymbol{\beta}^\top \boldsymbol{\mu} Y + \boldsymbol{\beta}^\top \boldsymbol{Z})^3]\bar{\boldsymbol{\beta}} - \mathbb{E}[\bar{\boldsymbol{\beta}}^\top \boldsymbol{Z}(\alpha + \boldsymbol{\beta}^\top \boldsymbol{\mu} Y + \boldsymbol{\beta}^\top \boldsymbol{Z})]\bar{\boldsymbol{\beta}}$$

$$= 3\alpha^2 \mathbb{E}[\bar{\boldsymbol{\beta}}^\top \boldsymbol{Z}(\boldsymbol{\beta}^\top \boldsymbol{\mu} Y + \boldsymbol{\beta}^\top \boldsymbol{Z})]\bar{\boldsymbol{\beta}} + \mathbb{E}[\bar{\boldsymbol{\beta}}^\top \boldsymbol{Z}(\boldsymbol{\beta}^\top \boldsymbol{\mu} Y + \boldsymbol{\beta}^\top \boldsymbol{Z})^3]\bar{\boldsymbol{\beta}} - \boldsymbol{\beta}$$

$$= (3\alpha^2 - 1)\boldsymbol{\beta} + 3\mathbb{E}(\boldsymbol{\beta}^\top \boldsymbol{\mu} Y)^2 \boldsymbol{\beta} + \mathbb{E}[\bar{\boldsymbol{\beta}}^\top \boldsymbol{Z}(\boldsymbol{\beta}^\top \boldsymbol{Z})^3]\bar{\boldsymbol{\beta}}$$

$$= [3\alpha^2 + 3(\boldsymbol{\mu}^\top \boldsymbol{\beta})^2 + M_Z \|\boldsymbol{\beta}\|_2^2 - 1]\boldsymbol{\beta},$$

where besides the arguments we have been using we also employed identities $\|\boldsymbol{\beta}\|_2 \bar{\boldsymbol{\beta}} = \boldsymbol{\beta}$ and $\mathbb{E}(\boldsymbol{\gamma}^\top \boldsymbol{Z})^4 = M_Z$ for any unit-norm $\boldsymbol{\gamma}$. Combining all these together, we get

$$\nabla_\alpha L_\lambda^h(\alpha, \boldsymbol{\beta}) = \alpha(\alpha^2 + 3(\boldsymbol{\beta}^\top \boldsymbol{\mu})^2 + 3\|\boldsymbol{\beta}\|^2 + \lambda - 1), \tag{21}$$

$$\nabla_{\boldsymbol{\beta}} L_\lambda^h(\alpha, \boldsymbol{\beta}) = [3\alpha^2 + (\boldsymbol{\beta}^\top \boldsymbol{\mu})^2 + 3\|\boldsymbol{\beta}\|_2^2 - 1](\boldsymbol{\mu}^\top \boldsymbol{\beta})\boldsymbol{\mu} + [3\alpha^2 + 3(\boldsymbol{\mu}^\top \boldsymbol{\beta})^2 + M_Z\|\boldsymbol{\beta}\|_2^2 - 1]\boldsymbol{\beta}. \tag{22}$$

Taking second derivatives,

$$\nabla_{\alpha\alpha}^2 L_\lambda^h(\alpha, \boldsymbol{\beta}) = 3\alpha^2 + 3(\boldsymbol{\beta}^\top \boldsymbol{\mu})^2 + 3\|\boldsymbol{\beta}\|_2^2 + \lambda - 1, \tag{23}$$

$$\nabla_{\boldsymbol{\beta}\alpha}^2 L_\lambda^h(\alpha, \boldsymbol{\beta}) = 6\alpha[(\boldsymbol{\beta}^\top \boldsymbol{\mu})\boldsymbol{\mu} + \boldsymbol{\beta}], \tag{24}$$

$$\nabla_{\boldsymbol{\beta}\boldsymbol{\beta}}^2 L_\lambda^h(\alpha, \boldsymbol{\beta}) = 3(\boldsymbol{\beta}^\top \boldsymbol{\mu})^2 \boldsymbol{\mu}\boldsymbol{\mu}^\top + (3\alpha^2 + 3\|\boldsymbol{\beta}\|_2^2 - 1)\boldsymbol{\mu}\boldsymbol{\mu}^\top + 6\boldsymbol{\mu}\boldsymbol{\mu}^\top \boldsymbol{\beta}\boldsymbol{\beta}^\top$$

$$\qquad + [3\alpha^2 + 3(\boldsymbol{\mu}^\top \boldsymbol{\beta})^2 + M_Z\|\boldsymbol{\beta}\|_2^2 - 1]\boldsymbol{I} + \boldsymbol{\beta}[6(\boldsymbol{\mu}^\top \boldsymbol{\beta})\boldsymbol{\mu}^\top + 2M_Z\boldsymbol{\beta}^\top]$$

$$= [3\alpha^2 + 3(\boldsymbol{\mu}^\top \boldsymbol{\beta})^2 + M_Z\|\boldsymbol{\beta}\|_2^2 - 1]\boldsymbol{I} + [3\alpha^2 + 3(\boldsymbol{\beta}^\top \boldsymbol{\mu})^2 + (3\|\boldsymbol{\beta}\|_2^2 - 1)]\boldsymbol{\mu}\boldsymbol{\mu}^\top$$

$$\qquad + 6(\boldsymbol{\mu}^\top \boldsymbol{\beta})(\boldsymbol{\mu}\boldsymbol{\beta}^\top + \boldsymbol{\beta}\boldsymbol{\mu}^\top) + 2M_Z\boldsymbol{\beta}\boldsymbol{\beta}^\top. \tag{25}$$

Now that we have derived the gradient and Hessian in closed form, we will characterize the lanscape. Let $(\alpha, \boldsymbol{\beta})$ be an arbitrary stationary point, we start by proving that it must satisfy $\alpha = 0$.

**Claim 1.** *If $\lambda \geq 1$ then $\alpha = 0$ holds for any critical point $(\alpha, \boldsymbol{\beta})$.*

*Proof.* Seeking a contradiction assume that $\alpha \neq 0$. We start by assuming $\boldsymbol{\beta} = c\boldsymbol{\mu}$ for some $c \in \mathbb{R}$, then the optimality condition $\nabla_\alpha L_\lambda^h(\alpha, \boldsymbol{\beta}) = 0$ gives $0 < \alpha^2 + 3c^2\|\boldsymbol{\mu}\|_2^2 \left(\|\boldsymbol{\mu}\|_2^2 + 1\right) = 1 - \lambda \leq 0$, yielding a contraction.

Now, let us assume that $\boldsymbol{\mu}$ and $\boldsymbol{\beta}$ are linearly independent, this assumption together with (21) and (22) imply that

$$\alpha^2 + 3(\boldsymbol{\beta}^\top \boldsymbol{\mu})^2 + 3\|\boldsymbol{\beta}\|_2^2 + \lambda - 1 = 0,$$
$$[3\alpha^2 + (\boldsymbol{\beta}^\top \boldsymbol{\mu})^2 + 3\|\boldsymbol{\beta}\|_2^2 - 1]\boldsymbol{\mu}^\top \boldsymbol{\beta} = 0,$$
$$3\alpha^2 + 3(\boldsymbol{\mu}^\top \boldsymbol{\beta})^2 + M_Z\|\boldsymbol{\beta}\|_2^2 - 1 = 0. \tag{26}$$

There are only two possible cases:

*Case 1.* If $\boldsymbol{\beta}^\top \boldsymbol{\mu} = 0$, then the optimality condition for $\alpha$ gives $\alpha^2 + 3\|\boldsymbol{\beta}\|_2^2 = 1 - \lambda \leq 0$, which is a contradiction.

*Case 2.* If $\boldsymbol{\beta}^\top \boldsymbol{\mu} \neq 0$, then $3\alpha^2 + (\boldsymbol{\beta}^\top \boldsymbol{\mu})^2 + 3\|\boldsymbol{\beta}\|_2^2 - 1 = 0$ and by substracting it from (26) we get $0 < 2(\boldsymbol{\beta}^\top \boldsymbol{\mu})^2 + (M_Z - 3)\|\boldsymbol{\beta}\|_2^2 = 0$, yielding a contradiction again.

This completes the proof of the claim. $\square$

This claim directly implies that the Hessian $\nabla^2 L_\lambda^h$, evaluated at any critical point, is a block diagonal matrix with $\nabla^2_{\boldsymbol{\beta}\alpha} L_\lambda^h(\alpha, \boldsymbol{\beta}) = 0$. Furthermore its first block is positive if $\boldsymbol{\beta} \neq \mathbf{0}$, as

$$\nabla^2_{\alpha\alpha} L_\lambda^h(\alpha, \boldsymbol{\beta}) = 3(\boldsymbol{\beta}^\top \boldsymbol{\mu})^2 + 3\|\boldsymbol{\beta}\|_2^2 + \lambda - 1 > \lambda - 1 \geq 0.$$

To prove the results regarding second order information at the critical points, it suffices to look at $\nabla^2_{\boldsymbol{\beta}\boldsymbol{\beta}} L_\lambda^h(\alpha, \boldsymbol{\beta})$.

Following a similar strategy to the one we used for the claim, let us start by assuming that $\boldsymbol{\beta}$ and $\boldsymbol{\mu}$ are linearly independent. Then, (22) yields

$$[(\boldsymbol{\beta}^\top \boldsymbol{\mu})^2 + 3\|\boldsymbol{\beta}\|_2^2 - 1](\boldsymbol{\mu}^\top \boldsymbol{\beta}) = 0, \tag{27}$$
$$3(\boldsymbol{\mu}^\top \boldsymbol{\beta})^2 + M_Z\|\boldsymbol{\beta}\|_2^2 - 1 = 0. \tag{28}$$

Consider two cases:

*Case 1.* If $\boldsymbol{\mu}^\top \boldsymbol{\beta} = 0$, then (28) yields $\|\boldsymbol{\beta}\|_2^2 = 1/M_Z$ and $(0, \boldsymbol{\beta}) \in S_2^h$.

*Case 2.* If $\boldsymbol{\mu}^\top \boldsymbol{\beta} \neq 0$, then (27) forces $(\boldsymbol{\beta}^\top \boldsymbol{\mu})^2 + 3\|\boldsymbol{\beta}\|_2^2 - 1 = 0$. Since $M_Z > 3$, this equation and (28) force $\boldsymbol{\beta} = \mathbf{0}$ and $\boldsymbol{\mu}^\top \boldsymbol{\beta} = 0$, which leads to contradiction.

Therefore, $S_2^h \backslash \{\mathbf{0}\}$ is the collection of all critical points that are linearly independent of $(0, \boldsymbol{\mu})$. For any $(0, \boldsymbol{\beta}) \in S_2^h \backslash \{\mathbf{0}\}$, we have

$$\nabla^2_{\boldsymbol{\beta}\boldsymbol{\beta}} L_\lambda^h(0, \boldsymbol{\beta}) = (3\|\boldsymbol{\beta}\|_2^2 - 1)\boldsymbol{\mu}\boldsymbol{\mu}^\top + 2M_Z\boldsymbol{\beta}\boldsymbol{\beta}^\top,$$
$$\boldsymbol{\mu}^\top \nabla^2_{\boldsymbol{\beta}\boldsymbol{\beta}} L_\lambda^h(0, \boldsymbol{\beta})\boldsymbol{\mu} = (3\|\boldsymbol{\beta}\|_2^2 - 1)\|\boldsymbol{\mu}\|_2^4 = -(1 - 3/M_Z)\|\boldsymbol{\mu}\|_2^4,$$
$$\boldsymbol{u}^\top \nabla^2 L_\lambda^h(0, \boldsymbol{\beta})\boldsymbol{u} \leq -(1 - 3/M_Z)\|\boldsymbol{\mu}\|_2^2 < 0, \tag{29}$$

where $\boldsymbol{u} = (0, \boldsymbol{\mu}/\|\boldsymbol{\mu}\|_2)$. Hence the points in $S_2^h \backslash \{\mathbf{0}\}$ are strict saddles.

Now, suppose that $\boldsymbol{\beta} = c\boldsymbol{\mu}$ and $\nabla L_\lambda^h(0, \boldsymbol{\beta}) = \mathbf{0}$. By (22),

$$\nabla L_\lambda^h(0, \boldsymbol{\beta}) = [(c\|\boldsymbol{\mu}\|_2^2)^3 + (3c^2\|\boldsymbol{\mu}\|_2^2 - 1)c\|\boldsymbol{\mu}\|_2^2]\boldsymbol{\mu} + [3(c\|\boldsymbol{\mu}\|_2^2)^2 + M_Zc^2\|\boldsymbol{\mu}\|_2^2 - 1]c\boldsymbol{\mu}$$
$$= [c^2\|\boldsymbol{\mu}\|_2^6 + (3c^2\|\boldsymbol{\mu}\|_2^2 - 1)\|\boldsymbol{\mu}\|_2^2 + 3c^2\|\boldsymbol{\mu}\|_2^4 + M_Zc^2\|\boldsymbol{\mu}\|_2^2 - 1]c\boldsymbol{\mu}$$
$$= [(\|\boldsymbol{\mu}\|_2^4 + 6\|\boldsymbol{\mu}\|_2^2 + M_Z)\|\boldsymbol{\mu}\|_2^2c^2 - (\|\boldsymbol{\mu}\|_2^2 + 1)]c\boldsymbol{\mu}.$$

It is easily seen that $\nabla L_\lambda^h(\mathbf{0}) = \mathbf{0}$. If $c \neq 0$, then

$$(\|\boldsymbol{\mu}\|_2^4 + 6\|\boldsymbol{\mu}\|_2^2 + M_Z)\|\boldsymbol{\mu}\|_2^2c^2 = \|\boldsymbol{\mu}\|_2^2 + 1. \tag{30}$$

Hence $S_1^h \cup \{\mathbf{0}\}$ is the collection of critical points that live in $\mathrm{span}\{(0, \boldsymbol{\mu})\}$, and $S_1^h \cup S_2^h$ contains all critical points of $L_\lambda^h$.

We first investigate $\{\mathbf{0}\}$. On the one hand,

$$\nabla_{\boldsymbol{\beta}\boldsymbol{\beta}}^2 L_\lambda^h(\mathbf{0}) = -(\boldsymbol{I} + \boldsymbol{\mu}\boldsymbol{\mu}^\top) \prec 0. \tag{31}$$

On the other hand,

$$L_\lambda^h(\alpha, \mathbf{0}) = h(\alpha) + \frac{\lambda}{2}\alpha^2 = \frac{1}{4}(\alpha^2 - 1)^2 + \frac{\lambda}{2}\alpha^2,$$
$$\nabla_\alpha L_\lambda^h(\alpha, \mathbf{0}) = \alpha^3 + (\lambda - 1)\alpha = \alpha(\alpha^2 + \lambda - 1).$$

It follows from $\lambda \geq 1$ that 0 is a local minimum of $L_\lambda^h(\cdot, \mathbf{0})$. Thus $\mathbf{0}$ is a saddle point of $L_\lambda^h$ whose Hessian has negative eigenvalues.

Next, for $(0, \boldsymbol{\beta}) \in S_1$, we derive from (25) that

$$\nabla_{\boldsymbol{\beta}\boldsymbol{\beta}}^2 L_\lambda^h(0, \boldsymbol{\beta}) = [3(c\|\boldsymbol{\mu}\|_2^2)^2 + M_Z c^2 \|\boldsymbol{\mu}\|_2^2 - 1]\boldsymbol{I} + [3(c\|\boldsymbol{\mu}\|_2^2)^2 + 3c^2\|\boldsymbol{\mu}\|_2^2 - 1]\boldsymbol{\mu}\boldsymbol{\mu}^\top$$
$$+ 6c\|\boldsymbol{\mu}\|_2^2 \cdot 2c\boldsymbol{\mu}\boldsymbol{\mu}^\top + 2M_Z c^2 \boldsymbol{\mu}\boldsymbol{\mu}^\top$$
$$= [(3\|\boldsymbol{\mu}\|_2^2 + M_Z)c^2\|\boldsymbol{\mu}\|_2^2 - 1]\boldsymbol{I} + [(3\|\boldsymbol{\mu}\|_2^4 + 15\|\boldsymbol{\mu}\|_2^2 + 2M_Z)c^2 - 1]\boldsymbol{\mu}\boldsymbol{\mu}^\top.$$

From (30) we see that

$$(3\|\boldsymbol{\mu}\|_2^2 + M_Z)c^2\|\boldsymbol{\mu}\|_2^2 - 1 = \frac{(3\|\boldsymbol{\mu}\|_2^2 + M_Z)(\|\boldsymbol{\mu}\|_2^2 + 1)}{\|\boldsymbol{\mu}\|_2^4 + 6\|\boldsymbol{\mu}\|_2^2 + M_Z} - 1 = \frac{2\|\boldsymbol{\mu}\|_2^4 + (M_Z - 3)\|\boldsymbol{\mu}\|_2^2}{\|\boldsymbol{\mu}\|_2^4 + 6\|\boldsymbol{\mu}\|_2^2 + M_Z} > 0,$$
$$(3\|\boldsymbol{\mu}\|_2^4 + 15\|\boldsymbol{\mu}\|_2^2 + 2M_Z)c^2 - 1 \geq 2(\|\boldsymbol{\mu}\|_2^4 + 6\|\boldsymbol{\mu}\|_2^2 + M_Z)c^2 - 1 = \frac{2(\|\boldsymbol{\mu}\|_2^2 + 1)}{\|\boldsymbol{\mu}\|_2^2} - 1 > 0.$$

Hence both points in $S_1$ are local minima because

$$\nabla_{\boldsymbol{\beta}\boldsymbol{\beta}}^2 L_\lambda^h(0, \boldsymbol{\beta}) \succeq \frac{2\|\boldsymbol{\mu}\|_2^4 + (M_Z - 3)\|\boldsymbol{\mu}\|_2^2}{\|\boldsymbol{\mu}\|_2^4 + 6\|\boldsymbol{\mu}\|_2^2 + M_Z}\boldsymbol{I} \succ 0, \qquad \forall (0, \boldsymbol{\beta}) \in S_1, \tag{32}$$

which immediately implies global optimality and finishes the proof.

### E.1.2    Part 2: Quantitative properties of the landscape

1. Lemma 2 implies that we can choose a sufficiently small constant $\varepsilon_1^h > 0$ and a constant $R > 0$ correspondingly such that $\|\nabla L_\lambda^h(\boldsymbol{\gamma})\|_2 \geq \varepsilon_1^h$ when $\|\boldsymbol{\gamma}\|_2 \geq R$. Without loss of generality, we can always take $\delta^h \leq 1$ and $R > 1 + \max_{\boldsymbol{\gamma} \in S_1^h \cup S_2^h} \|\boldsymbol{\gamma}\|_2$. In doing so, we have

$$\mathcal{S} = \{\boldsymbol{\gamma} : \|\boldsymbol{\gamma}\|_2 \leq R, \ \mathrm{dist}(\boldsymbol{\gamma}, S_1^h \cup S_2^h) \geq \delta^h\} \neq \varnothing.$$

We now establish a lower bound for $\inf_{\boldsymbol{\gamma} \in \mathcal{S}} \|\nabla L_\lambda^h(\boldsymbol{\gamma})\|_2$. Define

$$\mathcal{S}_{\boldsymbol{\beta}} = \mathrm{span}\{(0, \boldsymbol{\mu}), (0, \boldsymbol{\beta}), (1, \mathbf{0})\} \cap \mathcal{S}, \qquad \forall \boldsymbol{\beta} \perp \boldsymbol{\mu},$$
$$\varepsilon_{\boldsymbol{\beta}} = \inf_{\boldsymbol{\gamma} \in \mathcal{S}_{\boldsymbol{\beta}}} \|\nabla L_\lambda^h(\boldsymbol{\gamma})\|_2.$$

By symmetry, $\varepsilon_{\boldsymbol{\beta}}$ is the same for all $\boldsymbol{\beta} \perp \boldsymbol{\mu}$. Denote this quantity by $\varepsilon_2^h$. Since $\mathcal{S} = \cup_{\boldsymbol{\beta} \perp \boldsymbol{\mu}} \mathcal{S}_{\boldsymbol{\beta}}$,

$$\inf_{\boldsymbol{\gamma} \in \mathcal{S}} \|\nabla L_\lambda^h(\boldsymbol{\gamma})\|_2 = \inf_{\boldsymbol{\beta} \perp \boldsymbol{\mu}} \inf_{\boldsymbol{\gamma} \in \mathcal{S}_{\boldsymbol{\beta}}} \|\nabla L_\lambda^h(\boldsymbol{\gamma})\|_2 = \inf_{\boldsymbol{\beta} \perp \boldsymbol{\mu}} \varepsilon_{\boldsymbol{\beta}} = \varepsilon_2^h.$$

Take any $\boldsymbol{\beta} \perp \boldsymbol{\mu}$. On the one hand, the nonnegative function $\|\nabla L_\lambda^h(\cdot)\|_2$ is continuous and its zeros are all in $S_1^h \cup S_2^h$. On the other hand, $\mathcal{S}_{\boldsymbol{\beta}}$ is compact and non-empty. Hence $\varepsilon_2^h = \varepsilon_{\boldsymbol{\beta}} > 0$ and it only depends on the function $L_\lambda^h$ restricted to a three-dimensional subspace, i.e. $\mathrm{span}\{(0, \boldsymbol{\mu}), (0, \boldsymbol{\beta}), (1, \mathbf{0})\}$. It is then straightforward to check using the quartic expression of $L_\lambda^h$ and symmetry that $\varepsilon_2^h$ is completely determined by $\|\boldsymbol{\mu}\|_2$, $M_Z$, $\lambda$ and $\delta^h$. From now on we write $\varepsilon_2^h(\delta^h)$ to emphasize its dependence on $\delta^h$, whose value remains to be determined.

To sum up, when $\delta^h \leq 1$ and $\varepsilon^h \leq \min\{\varepsilon_1^h, \varepsilon_2^h(\delta^h)\}$, we have the desired result in the first claim.

2. Given properties (29), (31) and (32) of Hessians at all critical points, it suffices to show that

$$\|\nabla^2 L_\lambda^h(\boldsymbol{\gamma}_1) - \nabla^2 L_\lambda^h(\boldsymbol{\gamma}_2)\|_2 \le C'\|\boldsymbol{\gamma}_1 - \boldsymbol{\gamma}_2\|_2, \qquad \forall \boldsymbol{\gamma}_1,\ \boldsymbol{\gamma}_2 \in B(\mathbf{0}, R) \tag{33}$$

holds for some constant $C'$ determined by $\|\boldsymbol{\mu}\|_2$ and $R$. In that case, we can take sufficiently small $\delta^h$ and $\eta^h$ to finish the proof.

Based on (23), (24) and (25), we first decompose $\nabla^2 L_\lambda^h(\boldsymbol{\gamma})$ into the sum of two matrices $\boldsymbol{I}(\boldsymbol{\gamma})$ and $\boldsymbol{J}(\boldsymbol{\gamma})$ :

$$\nabla^2 L_\lambda^h(\boldsymbol{\gamma}) = \begin{pmatrix} 3\alpha^2 + 3\left(\boldsymbol{\beta}^\top \boldsymbol{\mu}\right)^2 + 3\|\boldsymbol{\beta}\|_2^2 + \lambda - 1 & 6\alpha \left[\left(\boldsymbol{\beta}^\top \boldsymbol{\mu}\right)\boldsymbol{\mu} + \boldsymbol{\beta}\right]^\top \\ 6\alpha \left[\left(\boldsymbol{\beta}^\top \boldsymbol{\mu}\right)\boldsymbol{\mu} + \boldsymbol{\beta}\right] & 3\alpha^2 \left(\boldsymbol{I} + \boldsymbol{\mu}\boldsymbol{\mu}^\top\right) \end{pmatrix}$$
$$+ \begin{pmatrix} 0 & \mathbf{0}^\top \\ \mathbf{0} & \nabla_{\boldsymbol{\beta}\boldsymbol{\beta}}^2 L^h(\boldsymbol{\gamma}) - 3\alpha^2 \left(\boldsymbol{I} + \boldsymbol{\mu}\boldsymbol{\mu}^\top\right) \end{pmatrix}$$
$$= \boldsymbol{I}(\boldsymbol{\gamma}) + \boldsymbol{J}(\boldsymbol{\gamma}).$$

For any $\boldsymbol{\gamma}_1 = (\alpha_1, \boldsymbol{\beta}_1), \boldsymbol{\gamma}_2 = (\alpha_2, \boldsymbol{\beta}_2) \in B(\mathbf{0}, R)$, we have

$$\|\boldsymbol{I}(\boldsymbol{\gamma}_1) - \boldsymbol{I}(\boldsymbol{\gamma}_2)\|_2 \le \left| 3\alpha_1^2 + 3\left(\boldsymbol{\beta}_1^\top \boldsymbol{\mu}\right)^2 + 3\|\boldsymbol{\beta}_1\|_2^2 - 3\alpha_2^2 - 3\left(\boldsymbol{\beta}_2^\top \boldsymbol{\mu}\right)^2 - 3\|\boldsymbol{\beta}_2\|_2^2 \right|$$
$$+ 2\left\|6\alpha_1\left[\left(\boldsymbol{\beta}_1^\top \boldsymbol{\mu}\right)\boldsymbol{\mu} + \boldsymbol{\beta}_1\right] - 6\alpha_2\left[\left(\boldsymbol{\beta}_2^\top \boldsymbol{\mu}\right)\boldsymbol{\mu} + \boldsymbol{\beta}_2\right]\right\|_2$$
$$+ \left\|3\left(\alpha_1^2 - \alpha_2^2\right)\left(\boldsymbol{I} + \boldsymbol{\mu}\boldsymbol{\mu}^\top\right)\right\|_2.$$

Let $\Delta = \|\boldsymbol{\gamma}_1 - \boldsymbol{\gamma}_2\|_2$ and note that $|\alpha_1^2 - \alpha_2^2| \le 2R\Delta$, $|\|\boldsymbol{\beta}_1\|_2^2 - \|\boldsymbol{\beta}_2\|_2^2| \le 2R\Delta$, $|(\boldsymbol{\beta}_1^\top \boldsymbol{\mu})^2 - (\boldsymbol{\beta}_2^\top \boldsymbol{\mu})^2| \le 2R\|\boldsymbol{\mu}\|_2^2\Delta$, $\|\alpha_1\boldsymbol{\beta}_1 - \alpha_2\boldsymbol{\beta}_2\|_2 \le 2R\Delta$ and $|\alpha_1(\boldsymbol{\beta}_1^\top \boldsymbol{\mu}) - \alpha_2(\boldsymbol{\beta}_2^\top \boldsymbol{\mu})| \le 2R\|\boldsymbol{\mu}\|_2\Delta$, we immediately have

$$\|\boldsymbol{I}(\boldsymbol{\gamma}_1) - \boldsymbol{I}(\boldsymbol{\gamma}_2)\|_2 \lesssim (1 + \|\boldsymbol{\mu}\|_2 + \|\boldsymbol{\mu}\|_2^2)R\|\boldsymbol{\gamma}_1 - \boldsymbol{\gamma}_2\|_2.$$

According to (25), $\boldsymbol{J}(\boldsymbol{\gamma})$ depends on $\boldsymbol{\beta}$ but not $\alpha$. Moreover, we have the following decomposition for its bottom right block:

$$\underbrace{\left[3\left(\boldsymbol{\mu}^\top \boldsymbol{\beta}\right)^2 + M_Z\|\boldsymbol{\beta}\|_2^2 - 1\right]\boldsymbol{I}}_{\boldsymbol{J}_1(\boldsymbol{\beta})} + \underbrace{\left[3\left(\boldsymbol{\beta}^\top \boldsymbol{\mu}\right)^2 + \left(3\|\boldsymbol{\beta}\|_2^2 - 1\right)\right]\boldsymbol{\mu}\boldsymbol{\mu}^\top}_{\boldsymbol{J}_2(\boldsymbol{\beta})}$$
$$+ \underbrace{6\left(\boldsymbol{\mu}^\top \boldsymbol{\beta}\right)\left(\boldsymbol{\mu}\boldsymbol{\beta}^\top + \boldsymbol{\beta}^\top \boldsymbol{\mu}\right)}_{\boldsymbol{J}_3(\boldsymbol{\beta})} + \underbrace{2M_Z\boldsymbol{\beta}\boldsymbol{\beta}^\top}_{\boldsymbol{J}_4(\boldsymbol{\beta})}.$$

Similar argument gives $\|\boldsymbol{J}_1(\boldsymbol{\beta}_1) - \boldsymbol{J}_1(\boldsymbol{\beta}_2)\| \lesssim (\|\boldsymbol{\mu}\|_2^2 + M_Z)R\Delta$, $\|\boldsymbol{J}_2(\boldsymbol{\beta}_1) - \boldsymbol{J}_2(\boldsymbol{\beta}_2)\|_2 \lesssim (\|\boldsymbol{\mu}\|_2^4 + \|\boldsymbol{\mu}\|_2^2)R\Delta$, $\|\boldsymbol{J}_3(\boldsymbol{\beta}_1) - \boldsymbol{J}_3(\boldsymbol{\beta}_2)\|_2 \lesssim \|\boldsymbol{\mu}\|_2^2 R\Delta$ and $\|\boldsymbol{J}_4(\boldsymbol{\beta}_1) - \boldsymbol{J}_4(\boldsymbol{\beta}_2)\|_2 \lesssim M_Z R\Delta$. As a result, we have

$$\|\boldsymbol{J}(\boldsymbol{\gamma}_1) - \boldsymbol{J}(\boldsymbol{\gamma}_2)\|_2 \lesssim (\|\boldsymbol{\mu}\|_2^2 + \|\boldsymbol{\mu}\|_2^4 + M_Z)R\|\boldsymbol{\gamma}_1 - \boldsymbol{\gamma}_2\|_2.$$

Hence we finally get (33).

## E.2  Proof of Lemma 2

By definition, $\nabla L_\lambda(\boldsymbol{\gamma}) - \nabla L_\lambda^h(\boldsymbol{\gamma}) = \mathbb{E}\left(\boldsymbol{X}\left[f'\left(\boldsymbol{\gamma}^\top \boldsymbol{X}\right) - h'\left(\boldsymbol{\gamma}^\top \boldsymbol{X}\right)\right]\right)$. From Lemma 1 we obtain that $|f'(x) - h'(x)| \lesssim |x|^3 \mathbf{1}_{\{|x| \ge a\}}$ when $b \ge 2a$ and $a$ is sufficiently large. When $\|\boldsymbol{\gamma}\|_2 \le R$, we have

$$\left\|\nabla L_\lambda(\boldsymbol{\gamma}) - \nabla L_\lambda^h(\boldsymbol{\gamma})\right\|_2 = \sup_{\boldsymbol{u} \in \mathbb{S}^d} \mathbb{E}\left(\boldsymbol{u}^\top \boldsymbol{X}\left[f'(\boldsymbol{\gamma}^\top \boldsymbol{X}) - h'(\boldsymbol{\gamma}^\top \boldsymbol{X})\right]\right)$$
$$\lesssim \sup_{\boldsymbol{u} \in \mathbb{S}^d} \mathbb{E}\left(\left|\boldsymbol{u}^\top \boldsymbol{X}\right|\left|\boldsymbol{\gamma}^\top \boldsymbol{X}\right|^3 \mathbf{1}_{\{|\boldsymbol{\gamma}^\top \boldsymbol{X}| \ge a\}}\right)$$
$$\overset{(i)}{\lesssim} \sup_{\boldsymbol{u} \in \mathbb{S}^d} \mathbb{E}^{1/3}\left|\boldsymbol{u}^\top \boldsymbol{X}\right|^3 \mathbb{E}^{1/3}\left|\boldsymbol{\gamma}^\top \boldsymbol{X}\right|^9 \mathbb{P}^{1/3}\left(\left|\boldsymbol{\gamma}^\top \boldsymbol{X}\right| \ge a\right)$$
$$\overset{(ii)}{\lesssim} \sup_{\boldsymbol{u} \in \mathbb{S}^d} \left\|\boldsymbol{u}^\top \boldsymbol{X}\right\|_{\psi_2} \left\|\boldsymbol{\gamma}^\top \boldsymbol{X}\right\|_{\psi_2}^3 \exp\left(-\frac{C_1 a^2}{\|\boldsymbol{\gamma}^\top \boldsymbol{X}\|_{\psi_2}^2}\right)$$
$$\overset{(iii)}{\le} R^3 M^4 \exp\left(-\frac{C_1 a^2}{R^2 M^2}\right)$$

for some constant $C_1 > 0$. Here (i) uses Hölder's inequality, (ii) comes from sub-Gaussian property (Vershynin, 2010), and (iii) uses $\|\boldsymbol{v}^\top \boldsymbol{X}\|_{\psi_2} \leq \|\boldsymbol{v}\|_2 \|\boldsymbol{X}\|_{\psi_2} = \|\boldsymbol{v}\|_2 M, \forall \boldsymbol{v} \in \mathbb{R}^{d+1}$.

To study the Hessian, we start from $\nabla^2 L_\lambda(\boldsymbol{\gamma}) - \nabla^2 L_\lambda^h(\boldsymbol{\gamma}) = \mathbb{E}\left(\boldsymbol{X}\boldsymbol{X}^\top \left[ f''(\boldsymbol{\gamma}^\top \boldsymbol{X}) - h''(\boldsymbol{\gamma}^\top \boldsymbol{X}) \right]\right)$. Again from Lemma 1 we know that $|f''(x) - h''(x)| \lesssim x^2 \mathbf{1}_{\{|x| \geq a\}}$. When $\|\boldsymbol{\gamma}\|_2 \leq R$, we have

$$
\begin{aligned}
\left\| \nabla^2 L_\lambda(\boldsymbol{\gamma}) - \nabla^2 L_\lambda^h(\boldsymbol{\gamma}) \right\|_2 &= \sup_{\boldsymbol{u} \in \mathbb{S}^d} \boldsymbol{u}^\top \mathbb{E}\left( \boldsymbol{X}\boldsymbol{X}^\top \left[ f''(\boldsymbol{\gamma}^\top \boldsymbol{X}) - h''(\boldsymbol{\gamma}^\top \boldsymbol{X}) \right]\right) \boldsymbol{u} \\
&\lesssim \sup_{\boldsymbol{u} \in \mathbb{S}^d} \mathbb{E}\left( |\boldsymbol{u}^\top \boldsymbol{X}|^2 |\boldsymbol{\gamma}^\top \boldsymbol{X}|^2 \mathbf{1}_{\{|\boldsymbol{\gamma}^\top \boldsymbol{X}| \geq a\}} \right) \\
&\lesssim \sup_{\boldsymbol{u} \in \mathbb{S}^d} \mathbb{E}^{1/3}|\boldsymbol{u}^\top \boldsymbol{X}|^6 \mathbb{E}^{1/3}|\boldsymbol{\gamma}^\top \boldsymbol{X}|^6 \mathbb{P}^{1/3}\left( |\boldsymbol{\gamma}^\top \boldsymbol{X}| \geq a \right) \\
&\lesssim \sup_{\boldsymbol{u} \in \mathbb{S}^d} \|\boldsymbol{u}^\top \boldsymbol{X}\|_{\psi_2}^2 \|\boldsymbol{\gamma}^\top \boldsymbol{X}\|_{\psi_2}^2 \exp\left( -\frac{C_1 a^2}{\|\boldsymbol{\gamma}^\top \boldsymbol{X}\|_{\psi_2}^2} \right) \\
&\leq R^2 M^4 \exp\left( -\frac{C_1 a^2}{R^2 M^2} \right)
\end{aligned}
$$

for some constant $C_1 > 0$.

We finally work on the lower bound for $\|\nabla L_\lambda(\boldsymbol{\gamma})\|_2$. From $b \geq 2a \geq 4$ we get $f(x) = h(x)$ for $|x| \leq a$; $f'(x) \geq 0$ and $f''(x) \geq 0$ for all $x \geq 1$. Since $f'$ is odd,

$$
\inf_{x \in \mathbb{R}} xf'(x) = \inf_{|x| \leq 1} xf'(x) = \inf_{|x| \leq 1} xh'(x) = \inf_{|x| \leq 1} \{x^4 - x^2\} \geq -1,
$$

$$
\inf_{|x| \geq 2} f'(x) \operatorname{sgn}(x) = \inf_{x \geq 2} f'(x) \geq f'(2) = h'(2) = 2^3 - 2 = 6.
$$

Taking $a = 2$, $b = 1$ and $c = 6$ in Lemma 8, we get

$$
\|L_\lambda(\boldsymbol{\gamma})\|_2 \geq 6 \inf_{\boldsymbol{u} \in \mathbb{S}^d} \mathbb{E}|\boldsymbol{u}^\top \boldsymbol{X}| - \frac{12 + 1}{\|\boldsymbol{\gamma}\|_2} \geq 6\varphi(\|\boldsymbol{X}\|_{\psi_2}, \lambda_{\min}[\mathbb{E}(\boldsymbol{X}\boldsymbol{X}^\top)]) - \frac{13}{\|\boldsymbol{\gamma}\|_2} \geq 6\varphi(M, \sigma^2) - \frac{13}{\|\boldsymbol{\gamma}\|_2}
$$

for $\boldsymbol{\gamma} \neq \mathbf{0}$. Here $\varphi$ is the function in Lemma 9. If we let $m = \varphi(M, \sigma^2)$, then $\inf_{\|\boldsymbol{\gamma}\|_2 \geq 3/m} \|L_\lambda(\boldsymbol{\gamma})\|_2 \geq m$. Follow a similar argument, we can show that $\inf_{\|\boldsymbol{\gamma}\|_2 \geq 3/m} \|L_\lambda^h(\boldsymbol{\gamma})\|_2 \geq m$ also holds for the same $m$.

### E.3 Proof of Lemma 3

To prove the first part, we define $\bar{\boldsymbol{\mu}} = \boldsymbol{\mu}/\|\boldsymbol{\mu}\|_2$ and seek for $c > 0$ determined by $\|\boldsymbol{\mu}\|_2$, the function $f$, and the distribution of $Z$ such that $\nabla L_1(0, \pm c\bar{\boldsymbol{\mu}}) = \mathbf{0}$.

By the chain rule, for any $(\alpha, \boldsymbol{\beta}, t) \in \mathbb{R} \times \mathbb{R}^d \times \mathbb{R}$ we have

$$
\nabla L_\lambda(\alpha, \boldsymbol{\beta}) = \begin{pmatrix} \mathbb{E}f'(\alpha + \boldsymbol{\beta}^\top \boldsymbol{X}) + \lambda\alpha \\ \mathbb{E}[\boldsymbol{X}f'(\alpha + \boldsymbol{\beta}^\top \boldsymbol{X})] \end{pmatrix} \quad \text{and} \quad \nabla L_1(0, t\bar{\boldsymbol{\mu}}) = \begin{pmatrix} \mathbb{E}f'(t\bar{\boldsymbol{\mu}}^\top \boldsymbol{X}) \\ \mathbb{E}[\boldsymbol{X}f'(t\bar{\boldsymbol{\mu}}^\top \boldsymbol{X})] \end{pmatrix}.
$$

Since $f$ is even, $f'$ is odd and $t\bar{\boldsymbol{\mu}}^\top \boldsymbol{X}$ has symmetric distribution with respect to 0, we have $\mathbb{E}f'(t\bar{\boldsymbol{\mu}}^\top \boldsymbol{X}) = 0$. It follows from $(\boldsymbol{I} - \bar{\boldsymbol{\mu}}\bar{\boldsymbol{\mu}}^\top)\boldsymbol{X} = (\boldsymbol{I} - \bar{\boldsymbol{\mu}}\bar{\boldsymbol{\mu}}^\top)\boldsymbol{Z}$ that

$$
(\boldsymbol{I} - \bar{\boldsymbol{\mu}}\bar{\boldsymbol{\mu}}^\top)\mathbb{E}[\boldsymbol{X}f'(t\bar{\boldsymbol{\mu}}^\top \boldsymbol{X})] = \mathbb{E}[(\boldsymbol{I} - \bar{\boldsymbol{\mu}}\bar{\boldsymbol{\mu}}^\top)\boldsymbol{Z}f'(t\bar{\boldsymbol{\mu}}^\top \boldsymbol{X})] = \mathbb{E}[(\boldsymbol{I} - \bar{\boldsymbol{\mu}}\bar{\boldsymbol{\mu}}^\top)\boldsymbol{Z}f'(t\|\boldsymbol{\mu}\|_2 Y + t\bar{\boldsymbol{\mu}}^\top \boldsymbol{Z})].
$$

Thanks to the independence between $Y$ and $\boldsymbol{Z}$ as well as the spherical symmetry of $\boldsymbol{Z}$, $(Y, \bar{\boldsymbol{\mu}}^\top \boldsymbol{Z}, (\boldsymbol{I} - \bar{\boldsymbol{\mu}}\bar{\boldsymbol{\mu}}^\top)\boldsymbol{Z})$ and $(Y, \bar{\boldsymbol{\mu}}^\top \boldsymbol{Z}, -(\boldsymbol{I} - \bar{\boldsymbol{\mu}}\bar{\boldsymbol{\mu}}^\top)\boldsymbol{Z})$ share the same distribution. Then

$$
(\boldsymbol{I} - \bar{\boldsymbol{\mu}}\bar{\boldsymbol{\mu}}^\top)\mathbb{E}[\boldsymbol{X}f'(t\bar{\boldsymbol{\mu}}^\top \boldsymbol{X})] = \mathbf{0} \quad \text{and} \quad \mathbb{E}[\boldsymbol{X}f'(t\bar{\boldsymbol{\mu}}^\top \boldsymbol{X})] = \bar{\boldsymbol{\mu}}\bar{\boldsymbol{\mu}}^\top \mathbb{E}[\boldsymbol{X}f'(t\bar{\boldsymbol{\mu}}^\top \boldsymbol{X})].
$$

As a result,

$$
\nabla L_\lambda(0, t\bar{\boldsymbol{\mu}}) = \mathbb{E}[\bar{\boldsymbol{\mu}}^\top \boldsymbol{X}f'(t\bar{\boldsymbol{\mu}}^\top \boldsymbol{X})] \begin{pmatrix} 0 \\ \bar{\boldsymbol{\mu}} \end{pmatrix}.
$$

Define $W = \bar{\boldsymbol{\mu}}^{\top}\boldsymbol{X} = \|\boldsymbol{\mu}\|_2 Y + \bar{\boldsymbol{\mu}}^{\top}\boldsymbol{Z}$ and $\varphi(t) = \mathbb{E}[Wf'(tW)]$ for $t \in \mathbb{R}$. The fact that $f$ is even yields $f'(0) = 0$ and $\varphi(0) = \mathbb{E}[Wf'(0)] = 0$. On the one hand, $f''(0) < 0$ forces

$$\varphi'(0) = \mathbb{E}[W^2 f''(tW)]|_{t=0} = f''(0)\mathbb{E}W^2 = f''(0)(\|\boldsymbol{\mu}\|_2^2 + 1) < 0. \tag{34}$$

Hence there exists $t_1 > 0$ such that $\varphi(t_1) < 0$. On the other hand, $\lim_{x \to +\infty} xf'(x) = +\infty$ leads to $\lim_{t \to +\infty} x\varphi(x) = \mathbb{E}[tWf'(tW)] = +\infty$. Then there exists $t_2 > 0$ such that $\varphi(t_2) > 0$. By the continuity of $\varphi$, we can find some $c > 0$ such that $\varphi(c) = 0$. Consequently,

$$\nabla \hat{L}_1(0, c\bar{\boldsymbol{\mu}}) = \varphi(c)\begin{pmatrix} 0 \\ \bar{\boldsymbol{\mu}} \end{pmatrix} = \boldsymbol{0}.$$

In addition, from

$$\varphi(-c) = \mathbb{E}[Wf'(-cW)] = -\mathbb{E}[Wf'(cW)] = -\varphi(c) = 0$$

we get $\nabla L(0, -c\bar{\boldsymbol{\mu}}) = \boldsymbol{0}$. It is easily seen that $t_1$, $t_2$ and $c$ are purely determined by properties of $f$ and $W$, where the latter only depends on $\|\boldsymbol{\mu}\|_2$ and the distribution of $Z$. This finishes the first part.

To prove the second part, we first observe that

$$|\varphi''(t)| = |\mathbb{E}[W^3 f'''(tW)]| \le F_3 \mathbb{E}|W|^3 = F_3(3^{-1/2}\mathbb{E}^{1/3}|W|^3)^3 \cdot 3^{3/2} \le 3^{3/2}F_3 M, \qquad \forall t \in \mathbb{R}.$$

Let $c_0 = -f''(0)(\|\boldsymbol{\mu}\|_2^2 + 1)/(3^{3/2}F_3 M)$. In view of (34),

$$\varphi'(t) \le \varphi'(0) + t\sup_{s \in \mathbb{R}}|\varphi''(s)| \le f''(0)(\|\boldsymbol{\mu}\|_2^2 + 1) + 3^{3/2}F_3 M t < 0, \qquad \forall t \in [0, c_0).$$

Thus $\varphi(t) < \varphi(0) = 0$ in the same interval, forcing $c > c_0$.

## F    Proof of Theorem 2

It suffices to prove the bound on the exceptional probability for each claim.

1. Claim 1 can be derived from Lemma 4, Theorem 1 and concentration of gradients within a ball (cf. Lemma 6).

   **Lemma 4.** *Let $\{\boldsymbol{X}_i\}_{i=1}^n$ be i.i.d. random vectors in $\mathbb{R}^{d+1}$ with $\|\boldsymbol{X}_i\|_{\psi_2} \le 1$ and $\mathbb{E}(\boldsymbol{X}_i \boldsymbol{X}_i^{\top}) \succeq \sigma^2 \boldsymbol{I}$ for some $\sigma > 0$, $f$ be defined in (8) with $b \ge 2a \ge 4$, and*

   $$\hat{L}_{\lambda}(\boldsymbol{\gamma}) = \frac{1}{n}\sum_{i=1}^n f(\boldsymbol{\gamma}^{\top}\boldsymbol{X}_i) + \frac{\lambda}{2}(\boldsymbol{\gamma}^{\top}\hat{\boldsymbol{\mu}})^2$$

   *with $\hat{\boldsymbol{\mu}} = \frac{1}{n}\sum_{i=1}^n \boldsymbol{X}_i$ and $\lambda \ge 0$. There exist positive constants $C, C_1, C_2, R$ and $\varepsilon_1$ determined by $\sigma$ such that when $n/d \ge C$,*

   $$\mathbb{P}\left(\inf_{\|\boldsymbol{\gamma}\|_2 \ge R}\|\nabla \hat{L}_{\lambda}(\boldsymbol{\gamma})\|_2 > \varepsilon_1\right) > 1 - C_1(d/n)^{C_2 d}.$$

   *Proof.* See Appendix F.1. $\qquad \square$

   Let $R$ and $\varepsilon$ be the constants stated in Lemma 4 and Theorem 1, respectively. Lemma 6 asserts that

   $$\mathbb{P}\left(\sup_{\boldsymbol{\gamma} \in B(\boldsymbol{0}, R)}\left\|\nabla \hat{L}_{\lambda}(\boldsymbol{\gamma}) - \nabla L_{\lambda}(\boldsymbol{\gamma})\right\|_2 < \frac{\varepsilon}{2}\right) > 1 - C_1(d/n)^{C_2 d}$$

   for some constant $C_1, C_2 > 0$, provided that $n/d$ is large enough. From Theorem 1 we know that $\|\nabla L_{\lambda}(\boldsymbol{\gamma})\|_2 \ge \varepsilon$ if $\mathrm{dist}(\boldsymbol{\gamma}, \{\pm\boldsymbol{\gamma}^{\star}\} \cup S) \ge \delta$. The triangle inequality immediately gives

   $$\mathbb{P}\left(\inf_{\boldsymbol{\gamma}:\,\mathrm{dist}(\boldsymbol{\gamma},\{\pm\boldsymbol{\gamma}^{\star}\}\cup S)\ge\delta}\|\nabla \hat{L}_{\lambda}(\boldsymbol{\gamma})\|_2 > \varepsilon/2\right) < 1 - C_1'(d/n)^{C_2' d},$$

   for some constants $C_1'$ and $C_2'$.

2. We invoke the following Lemma 5 to prove Claim 2.

**Lemma 5.** *Let $\{\boldsymbol{X}_i\}_{i=1}^n$ be i.i.d. random vectors in $\mathbb{R}^{d+1}$ with $\|\boldsymbol{X}_i\|_{\psi_2} \leq 1$; $\boldsymbol{u} \in \mathbb{S}^d$ be deterministic; $R > 0$ be a constant. Let $f$ be defined in (8) with constants $b \geq 2a \geq 4$, and*

$$\hat{L}_\lambda(\boldsymbol{\gamma}) = \frac{1}{n} \sum_{i=1}^n f(\boldsymbol{\gamma}^\top \boldsymbol{X}_i) + \frac{\lambda}{2}(\boldsymbol{\gamma}^\top \hat{\boldsymbol{\mu}})^2$$

*with $\hat{\boldsymbol{\mu}} = \frac{1}{n}\sum_{i=1}^n \boldsymbol{X}_i$ and $\lambda \geq 0$. Suppose that $n/d \geq e$. There exist positive constants $C_1, C_2, C$ and $N$ such that when $n > N$,*

$$\mathbb{P}\left( \sup_{\boldsymbol{\gamma}_1 \neq \boldsymbol{\gamma}_2} \frac{\|\nabla \hat{L}_\lambda(\boldsymbol{\gamma}_1) - \nabla \hat{L}_\lambda(\boldsymbol{\gamma}_2)\|_2}{\|\boldsymbol{\gamma}_1 - \boldsymbol{\gamma}_2\|_2} < C \right) > 1 - C_1 e^{-C_2 n},$$

$$\mathbb{P}\left( \sup_{\boldsymbol{\gamma}_1 \neq \boldsymbol{\gamma}_2} \frac{\|\nabla^2 \hat{L}_\lambda(\boldsymbol{\gamma}_1) - \nabla^2 \hat{L}_\lambda(\boldsymbol{\gamma}_2)\|_2}{\|\boldsymbol{\gamma}_1 - \boldsymbol{\gamma}_2\|_2} < C \max\{1, d\log(n/d)/\sqrt{n}\} \right) > 1 - C_1 (d/n)^{C_2 d},$$

$$\mathbb{P}\left( \sup_{\|\boldsymbol{\gamma}\|_2 \leq R} |\boldsymbol{u}^\top [\nabla^2 \hat{L}_\lambda(\boldsymbol{\gamma}) - \nabla^2 L_\lambda(\boldsymbol{\gamma})]\boldsymbol{u}| < C\sqrt{d\log(n/d)/n} \right) > 1 - C_1(d/n)^{C_2 d} - C_1 e^{-C_2 n^{1/3}}.$$

*Proof.* See Appendix F.2. □

From Theorem 1 we know that $\boldsymbol{u}^\top \nabla^2 L_\lambda(\boldsymbol{\gamma})\boldsymbol{u} \leq -\eta$ if $\mathrm{dist}(\boldsymbol{\gamma}, S) \leq \delta$. Lemma 5 (after proper rescaling) asserts that

$$\mathbb{P}\left( \sup_{\|\boldsymbol{\gamma}\|_2 \leq R} |\boldsymbol{u}^\top [\nabla^2 \hat{L}_\lambda(\boldsymbol{\gamma}) - \nabla^2 L_\lambda(\boldsymbol{\gamma})]\boldsymbol{u}| < \frac{\eta}{2} \right) > 1 - C_1(d/n)^{C_2 d} - C_1 e^{-C_2 n^{1/3}}$$

provided that $n/d$ is sufficiently large. Then Claim 2 follows from the triangle's inequality.

3. Claim 3 follows from Lemma 5 with proper rescaling.

### F.1 Proof of Lemma 4

It is shown in Lemma 2 that when $b \geq 2a \geq 4$, we have $\inf_{x \in \mathbb{R}} xf'(x) \geq -1$ and $\inf_{|x| \geq 2} f'(x)\,\mathrm{sgn}(x) \geq 6$. Using an empirical version of Lemma 8,

$$\nabla \hat{L}_\lambda(\boldsymbol{\gamma}) \geq \inf_{\boldsymbol{u} \in \mathbb{S}^d} \frac{1}{n}\sum_{i=1}^n |\boldsymbol{u}^\top \boldsymbol{X}_i| - \frac{13}{\|\boldsymbol{\gamma}\|_2}, \qquad \forall \boldsymbol{\gamma} \in \mathbb{R}^d.$$

Define $S_n(\boldsymbol{u}) = \frac{1}{n}\sum_{i=1}^n (|\boldsymbol{u}^\top \boldsymbol{X}_i| - \mathbb{E}|\boldsymbol{u}^\top \boldsymbol{X}_i|)$ for $\boldsymbol{u} \in \mathbb{S}^d$. By the triangle inequality,

$$\hat{L}_\lambda(\boldsymbol{\gamma}) \geq \inf_{\boldsymbol{u} \in \mathbb{S}^d} \mathbb{E}|\boldsymbol{u}^\top \boldsymbol{X}_1| - \sup_{\boldsymbol{u} \in \mathbb{S}^d} |S_n(\boldsymbol{u})| - \frac{13}{\|\boldsymbol{\gamma}\|_2}, \qquad \forall \boldsymbol{\gamma} \in \mathbb{R}^d.$$

According to Lemma 9, $\inf_{\boldsymbol{u} \in \mathbb{S}^d} \mathbb{E}|\boldsymbol{u}^\top \boldsymbol{X}_1| > \varphi$ for some constant $\varphi > 0$ determined by $\sigma$. Then it suffices to prove

$$\sup_{\boldsymbol{u} \in \mathbb{S}^d} |S_n(\boldsymbol{u})| = O_\mathbb{P}(\sqrt{d\log(n/d)/n};\ d\log(n/d)). \tag{35}$$

We will use Theorem 1 in Wang (2019) to get there.

1. Since $\|\boldsymbol{X}_i\|_{\psi_2} \leq 1$, the Hoeffding-type inequality in Proposition 5.10 of Vershynin (2010) asserts the existence of a constant $c > 0$ such that

$$\mathbb{P}(|S_n(\boldsymbol{u})| \geq t) \leq e \cdot e^{-cnt^2}, \qquad \forall t \geq 0.$$

Then $\{S_n(\boldsymbol{u})\}_{\boldsymbol{u} \in \mathbb{S}^d} = O_\mathbb{P}(\sqrt{d\log(n/d)/n};\ d\log(n/d))$.

2. Let $\varepsilon_n = \sqrt{d/n}$. According to Lemma 5.2 in Vershynin (2010), there exists an $\varepsilon_n$-net $\mathcal{N}_n$ of $\mathbb{S}^d$ with cardinality at most $(1 + 2R/\varepsilon_n)^d$. When $n/d$ is large, $\log|\mathcal{N}_n| = d\log(1 + \sqrt{n/d}) \lesssim d\log(n/d)$.

3. Define $M_n = \sup_{\boldsymbol{u} \in \mathbb{S}^d, \boldsymbol{v} \in \mathbb{S}^d, \boldsymbol{u} \neq \boldsymbol{v}} \{|S_n(\boldsymbol{u}) - S_n(\boldsymbol{v})|/\|\boldsymbol{u} - \boldsymbol{v}\|_2\}$. By Cauchy-Schwarz inequality,

$$\left| \frac{1}{n} \sum_{i=1}^{n} |\boldsymbol{u}^\top \boldsymbol{X}_i| - \frac{1}{n} \sum_{i=1}^{n} |\boldsymbol{v}^\top \boldsymbol{X}_i| \right| \leq \frac{1}{n} \sum_{i=1}^{n} |(\boldsymbol{u} - \boldsymbol{v})^\top \boldsymbol{X}_i| \leq \left( \frac{1}{n} \sum_{i=1}^{n} |(\boldsymbol{u} - \boldsymbol{v})^\top \boldsymbol{X}_i|^2 \right)^{1/2}$$

$$\leq \|\boldsymbol{u} - \boldsymbol{v}\|_2 \sup_{\boldsymbol{w} \in \mathbb{S}^d} \left( \frac{1}{n} \sum_{i=1}^{n} |\boldsymbol{w}^\top \boldsymbol{X}_i|^2 \right)^{1/2}$$

$$= \|\boldsymbol{u} - \boldsymbol{v}\|_2 \cdot O_{\mathbb{P}}(1; \, n),$$

where the last equality follows from Lemma 11. Similarly,

$$\left| \mathbb{E}|\boldsymbol{u}^\top \boldsymbol{X}_1| - \mathbb{E}|\boldsymbol{v}^\top \boldsymbol{X}_1| \right| \leq \|\boldsymbol{u} - \boldsymbol{v}\|_2 \|\mathbb{E}(\boldsymbol{X}_1 \boldsymbol{X}_1^\top)\|_2 \lesssim \|\boldsymbol{u} - \boldsymbol{v}\|_2.$$

Hence $M_n = O_{\mathbb{P}}(1; \, n)$.

Then Theorem 1 in Wang (2019) yields (35).

### F.2 Proof of Lemma 5

It follows from Example 6 in Wang (2019) that $\|n^{-1} \sum_{i=1}^{n} \boldsymbol{X}_i - \boldsymbol{\mu}_0\|_2 = O_{\mathbb{P}}(1; \, n)$. As a result $\|n^{-1} \sum_{i=1}^{n} \boldsymbol{X}_i\|_2 = O_{\mathbb{P}}(1; \, n)$. This combined with Lemma 8 and Lemma 11 gives

$$\sup_{\boldsymbol{\gamma}_1 \neq \boldsymbol{\gamma}_2} \frac{\|\nabla \hat{L}_\lambda(\boldsymbol{\gamma}_1) - \nabla \hat{L}_\lambda(\boldsymbol{\gamma}_2)\|_2}{\|\boldsymbol{\gamma}_1 - \boldsymbol{\gamma}_2\|_2} = O_{\mathbb{P}}(1; \, n),$$

$$\sup_{\boldsymbol{\gamma}_1 \neq \boldsymbol{\gamma}_2} \frac{|\boldsymbol{u}^\top [\nabla^2 \hat{L}_\lambda(\boldsymbol{\gamma}_1) - \nabla^2 \hat{L}_\lambda(\boldsymbol{\gamma}_2)]\boldsymbol{u}|}{\|\boldsymbol{\gamma}_1 - \boldsymbol{\gamma}_2\|_2} = O_{\mathbb{P}}(1; \, n^{1/3}),$$

$$\sup_{\boldsymbol{\gamma}_1 \neq \boldsymbol{\gamma}_2} \frac{\|\nabla^2 \hat{L}_\lambda(\boldsymbol{\gamma}_1) - \nabla^2 \hat{L}_\lambda(\boldsymbol{\gamma}_2)\|_2}{\|\boldsymbol{\gamma}_1 - \boldsymbol{\gamma}_2\|_2} = O_{\mathbb{P}}(\max\{1, d\log(n/d)/\sqrt{n}\}; \, d\log(n/d))$$

given $F_2 \leq 3a^2 \lesssim 1$ and $F_3 \leq 6a \lesssim 1$, provided that $n/d$ is sufficiently large. It is easily seen that there exist universal constants $(c_1, c_2, N) \in (0, +\infty)^3$ and a non-decreasing function $f : [c_2, +\infty) \to (0, +\infty)$ with $\lim_{x \to \infty} f(x) = \infty$, such that

$$\mathbb{P}\left( \sup_{\boldsymbol{\gamma}_1 \neq \boldsymbol{\gamma}_2} \frac{\|\nabla \hat{L}_\lambda(\boldsymbol{\gamma}_1) - \nabla \hat{L}_\lambda(\boldsymbol{\gamma}_2)\|_2}{\|\boldsymbol{\gamma}_1 - \boldsymbol{\gamma}_2\|_2} \geq t \right) \leq c_1 e^{-nf(t)}, \tag{36}$$

$$\mathbb{P}\left( \sup_{\boldsymbol{\gamma}_1 \neq \boldsymbol{\gamma}_2} \frac{|\boldsymbol{u}^\top [\nabla^2 \hat{L}_\lambda(\boldsymbol{\gamma}_1) - \nabla^2 \hat{L}_\lambda(\boldsymbol{\gamma}_2)]\boldsymbol{u}|}{\|\boldsymbol{\gamma}_1 - \boldsymbol{\gamma}_2\|_2} \geq t \right) \leq c_1 e^{-n^{1/3} f(t)}, \tag{37}$$

$$\mathbb{P}\left( \sup_{\boldsymbol{\gamma} \neq \boldsymbol{\gamma}} \frac{\|\nabla^2 \hat{L}_\lambda(\boldsymbol{\gamma}_1) - \nabla^2 \hat{L}_\lambda(\boldsymbol{\gamma}_2)\|_2}{\|\boldsymbol{\gamma}_1 - \boldsymbol{\gamma}_2\|_2} \geq t \max\{1, d\log(n/d)/\sqrt{n}\} \right) \leq c_1 e^{-d\log(n/d)f(t)} = c_1(d/n)^{df(t)}, \tag{38}$$

as long as $n \geq N_1$ and $t \geq c_2$. We prove the first two inequalities in Lemma 5 by (36), (38) and choosing proper constants.

Let

$$X_n(\boldsymbol{\gamma}) = \boldsymbol{u}^\top [\nabla^2 \hat{L}_\lambda(\boldsymbol{\gamma}) - \nabla^2 L_\lambda(\boldsymbol{\gamma})]\boldsymbol{u} = \boldsymbol{u}^\top [\nabla^2 \hat{L}(\boldsymbol{\gamma}) - \nabla^2 L(\boldsymbol{\gamma})]\boldsymbol{u},$$

$\mathcal{S}_n = B(\boldsymbol{0}, R)$ and $m = \log(n/d)$. We will invoke Theorem 1 in Wang (2019) to control $\sup_{\boldsymbol{\gamma} \in \mathcal{S}_n} |X_n(\boldsymbol{\gamma})|$ and prove the remaining claim.

1. By definition, $X_n(\boldsymbol{\gamma}) = \frac{1}{n} \sum_{i=1}^{n} \{(\boldsymbol{u}^\top \boldsymbol{X}_i)^2 f''(\boldsymbol{\gamma}^\top \boldsymbol{X}_i) - \mathbb{E}[(\boldsymbol{u}^\top \boldsymbol{X}_i)^2 f''(\boldsymbol{\gamma}^\top \boldsymbol{X}_i)]\}$ and

$$\|(\boldsymbol{u}^\top \boldsymbol{X}_i)^2 f''(\boldsymbol{\gamma}^\top \boldsymbol{X}_i)\|_{\psi_1} \leq F_2 \|(\boldsymbol{u}^\top \boldsymbol{X}_i)^2\|_{\psi_1} \lesssim F_2 \|\boldsymbol{u}^\top \boldsymbol{X}_i\|_{\psi_2}^2 \lesssim 1.$$

By the Bernstein-type inequality in Proposition 5.16 of Vershynin (2010), there is a constant $c'$ such that

$$\mathbb{P}(|X_n(\boldsymbol{\gamma})| \geq t) \leq 2e^{-c'n[t^2 \wedge t]}, \qquad \forall t \geq 0, \, \boldsymbol{\gamma} \in \mathbb{R}^d.$$

When $t = s\sqrt{md/n}$ for $s \geq 1$, we have $nt^2 = s^2 md \geq smd$. Since $n/d \geq e$, we have

$$m = \log(n/d) = \log[1 + (n/d - 1)] \leq n/d - 1 \leq n/d,$$

$n \geq md$ and $nt = s\sqrt{nmd} \geq smd$. This gives

$$\mathbb{P}(|X_n(\boldsymbol{\gamma})| \geq s\sqrt{md/n}) \leq 2e^{-c'mds}, \qquad \forall s \geq 1, \ \boldsymbol{\gamma} \in \mathbb{R}^d.$$

Hence $\{X_n(\boldsymbol{\gamma})\}_{\boldsymbol{\gamma} \in \mathcal{S}_n} = O_{\mathbb{P}}(\sqrt{md/n}; \ md)$.

2. Let $\varepsilon_n = 2R\sqrt{d/n}$. According to Lemma 5.2 in Vershynin (2010), there exists an $\varepsilon_n$-net $\mathcal{N}_n$ of $\mathcal{S}_n$ with cardinality at most $(1 + 2R/\varepsilon_n)^d$. Since $n/d \geq e$, $\log|\mathcal{N}_n| = d\log(1 + \sqrt{n/d}) \lesssim d\log(n/d) = md$.

3. Define $M_n = \sup_{\boldsymbol{\gamma}_1 \neq \boldsymbol{\gamma}_2}\{|X_n(\boldsymbol{\gamma}_1) - X_n(\boldsymbol{\gamma}_2)|/\|\boldsymbol{\gamma}_1 - \boldsymbol{\gamma}_2\|_2\}$. Observe that by Lemma 8 and $\|\boldsymbol{X}_i\|_{\psi_2} \leq 1$,

$$\sup_{\boldsymbol{\gamma}_1 \neq \boldsymbol{\gamma}_2} \frac{|\boldsymbol{u}^\top[\nabla^2 L\lambda(\boldsymbol{\gamma}_1) - \nabla^2 L_\lambda(\boldsymbol{\gamma}_2)]\boldsymbol{u}|}{\|\boldsymbol{\gamma}_1 - \boldsymbol{\gamma}_2\|_2} \leq \sup_{\boldsymbol{\gamma}_1 \neq \boldsymbol{\gamma}_2} \frac{\|\nabla^2 L(\boldsymbol{\gamma}_1) - \nabla^2 L(\boldsymbol{\gamma}_2)\|_2}{\|\boldsymbol{\gamma}_1 - \boldsymbol{\gamma}_2\|_2}$$
$$\leq F_3 \sup_{\boldsymbol{u} \in \mathbb{S}^d} \mathbb{E}|\boldsymbol{u}^\top \boldsymbol{X}|^3 \leq (\sqrt{3})^3 F_3 \lesssim 1.$$

From this and (37) we obtain that $M_n = O_{\mathbb{P}}(1; \ n^{1/3})$.

Based on these, Theorem 1 Wang (2019) implies that

$$\sup_{\boldsymbol{\gamma} \in \mathcal{S}_n} |X_n(\boldsymbol{\gamma})| = O_{\mathbb{P}}(\sqrt{md/n} + \varepsilon_n; \ md \wedge n^{1/3}) = O_{\mathbb{P}}(\sqrt{\log(n/d)d/n}; \ d\log(n/d) \wedge n^{1/3}).$$

As a result, there exist absolute constants $(c_1', c_2', N_1') \in (0, +\infty)^3$ and a non-decreasing function $g : [c_2', +\infty) \to (0, +\infty)$ such that

$$\mathbb{P}\left(\sup_{\boldsymbol{\gamma} \in \mathcal{S}_n} |X_n(\boldsymbol{\gamma})| \geq t\sqrt{\log(n/d)d/n}\right) \leq c_1' e^{-(md \wedge n^{1/3})g(t)} \leq c_1'(e^{-mdg(t)} + e^{-n^{1/3}g(t)})$$
$$\leq c_1'(d/n)^{dg(t)} + c_1' e^{-n^{1/3}g(t)}, \qquad \forall n \geq N_1', \ t \geq c_2'.$$

The proof is finished by taking $t = c_2'$ and re-naming some constants above.

## G   Proof of Corollary 1

From Claim 1 in the second item of Theorem 2, we know that $\|\nabla\hat{L}_1(\boldsymbol{\gamma})\|_2 \leq \varepsilon$ implies $\mathrm{dist}(\boldsymbol{\gamma}, \{\pm\boldsymbol{\gamma}^\star\} \cup S) < \delta$. On the other side, since $\lambda_{\min}[\nabla^2\hat{L}_1(\boldsymbol{\gamma})] > -\eta$, we have $\boldsymbol{v}^\top\nabla^2\hat{L}_1(\boldsymbol{\gamma})\boldsymbol{v} > -\eta$ for any unit vector $\boldsymbol{v}$. Then in view of Claim 2 of Theorem 2, we know that $\mathrm{dist}(\boldsymbol{\gamma}, S) > \delta$. Therefore we arrive at $\mathrm{dist}(\boldsymbol{\gamma}, \{\pm\boldsymbol{\gamma}^\star\}) < \delta$. According to Theorem 1, $\nabla^2 L_1(\boldsymbol{\gamma}') \succeq \eta\boldsymbol{I}$ so long as $\mathrm{dist}(\boldsymbol{\gamma}', S_1) \leq \delta$. This and $\nabla L_1(\boldsymbol{\gamma}^\star) = \boldsymbol{0}$ lead to

$$\min_{s = \pm 1} \|s\boldsymbol{\gamma} - \boldsymbol{\gamma}^\star\|_2 \leq \frac{1}{\eta}\|\nabla L_1(\boldsymbol{\gamma}) - \nabla L_1(\boldsymbol{\gamma}^\star)\|_2 = \frac{1}{\eta}\|\nabla L_1(\boldsymbol{\gamma})\|_2$$
$$\leq \frac{1}{\eta}\|\nabla\hat{L}_1(\boldsymbol{\gamma})\|_2 + \frac{1}{\eta}\|\nabla\hat{L}_1(\boldsymbol{\gamma}) - \nabla L_1(\boldsymbol{\gamma})\|_2. \qquad (39)$$

All of these hold with probability exceeding $1 - C_1(d/n)^{C_2 d} - C_1\exp(-C_2 n^{1/3})$.

The desired result is a product of (39) and Lemma 6 below.

**Lemma 6.** *For any constant $R > 0$, there exists a constant $C > 0$ such that when $n \geq Cd$ for all $n$,*

$$\sup_{\|\boldsymbol{\gamma}\|_2 \leq R} \left\|\nabla\hat{L}_1(\boldsymbol{\gamma}) - \nabla L_1(\boldsymbol{\gamma})\right\|_2 = O_{\mathbb{P}}\left(\sqrt{\frac{d}{n}\log\left(\frac{n}{d}\right)}; \ d\log\left(\frac{n}{d}\right)\right) \qquad (40)$$

*Proof.* See Appendix G.1. $\qquad\square$

### G.1 Proof of Lemma 6

Let $\boldsymbol{\gamma} = (\alpha, \boldsymbol{\beta})$, $\hat{L}(\boldsymbol{\gamma}) = \frac{1}{n}\sum_{i=1}^{n} f(\alpha + \boldsymbol{\beta}^{\top}\boldsymbol{X}_i)$, $L(\boldsymbol{\gamma}) = \mathbb{E}f(\alpha + \boldsymbol{\beta}^{\top}\boldsymbol{X})$, $\hat{R}(\boldsymbol{\gamma}) = \frac{1}{2}(\alpha + \boldsymbol{\beta}^{\top}\hat{\boldsymbol{\mu}}_0)^2$ and $R(\boldsymbol{\gamma}) = \frac{1}{2}(\alpha + \boldsymbol{\beta}^{\top}\boldsymbol{\mu}_0)^2$. Since $|f'(0)| = 0$, $\sup_{x \in \mathbb{R}}|f''(x)| = h'(a) + (b - a)h''(a) \leq 3a^2 b \lesssim 1$ and $\|\boldsymbol{X}_i\|_{\psi_2} \leq M \lesssim 1$, from Theorem 2 in Wang (2019) we get

$$\sup_{\|\boldsymbol{\gamma}\|_2 \leq R}\left\|\nabla\hat{L}\left(\boldsymbol{\gamma}\right) - \nabla L\left(\boldsymbol{\gamma}\right)\right\|_2 = O_{\mathbb{P}}\left(\sqrt{\frac{d}{n}\log\left(\frac{n}{d}\right)}; \, d\log\left(\frac{n}{d}\right)\right).$$

Then it boils down to proving uniform convergence of $\|\nabla\hat{R}(\boldsymbol{\gamma}) - \nabla R(\boldsymbol{\gamma})\|$. Let $\bar{\boldsymbol{X}}_i = (1, \boldsymbol{X}_i)$, $\tilde{\boldsymbol{\mu}}_0 = (1, \frac{1}{n}\sum_{i=1}^{n}\boldsymbol{X}_i)$ and $\bar{\boldsymbol{\mu}}_0 = (1, \boldsymbol{\mu}_0)$. By definition,

$$\nabla\hat{R}\left(\boldsymbol{\gamma}\right) = \left(\boldsymbol{\gamma}^{\top}\tilde{\boldsymbol{\mu}}_0\right)\tilde{\boldsymbol{\mu}}_0 \qquad \text{and} \qquad \nabla R\left(\boldsymbol{\gamma}\right) = \left(\boldsymbol{\gamma}^{\top}\bar{\boldsymbol{\mu}}_0\right)\bar{\boldsymbol{\mu}}_0,$$

Since $\|\bar{\boldsymbol{X}}_i - \bar{\boldsymbol{\mu}}_0\|_{\psi_2} \lesssim \|\bar{\boldsymbol{X}}_i\|_{\psi_2} \lesssim 1$, we know that $\|\tilde{\boldsymbol{\mu}}_0 - \bar{\boldsymbol{\mu}}_0\|_{\psi_2} \lesssim 1/\sqrt{n}$. In view of Example 6 Wang (2019) and $\|\boldsymbol{\mu}_0\|_2 \lesssim 1$, we know that $\|\tilde{\boldsymbol{\mu}}_0 - \boldsymbol{\mu}_0\|_2 = O_{\mathbb{P}}(\sqrt{d/n\log(n/d)}; \, d\log(n/d))$ and $\|\tilde{\boldsymbol{\mu}}_0\|_2 = O_{\mathbb{P}}(1; \, d\log(n/d))$. As a result,

$$\begin{aligned}
\sup_{\|\boldsymbol{\gamma}\|_2 \leq R}\left\|\nabla\hat{R}\left(\boldsymbol{\gamma}\right) - \nabla R\left(\boldsymbol{\gamma}\right)\right\|_2 &\leq \sup_{\|\boldsymbol{\gamma}\|_2 \leq R}\left\{\left|\boldsymbol{\gamma}^{\top}\left(\tilde{\boldsymbol{\mu}}_0 - \bar{\boldsymbol{\mu}}_0\right)\right|\|\tilde{\boldsymbol{\mu}}_0\|_2 + \left|\boldsymbol{\gamma}^{\top}\bar{\boldsymbol{\mu}}_0\right|\|\tilde{\boldsymbol{\mu}}_0 - \bar{\boldsymbol{\mu}}_0\|_2\right\} \\
&\leq R\|\tilde{\boldsymbol{\mu}}_0 - \bar{\boldsymbol{\mu}}_0\|_2\left(\|\tilde{\boldsymbol{\mu}}_0\|_2 + \|\bar{\boldsymbol{\mu}}_0\|_2\right) \\
&= O_{\mathbb{P}}\left(\sqrt{\frac{d}{n}\log\left(\frac{n}{d}\right)}; \, d\log\left(\frac{n}{d}\right)\right).
\end{aligned}$$

## H  Proof of Theorem 3

To prove Theorem 3, we invoke the convergence guarantees for perturbed gradiend descent in Jin et al. (2017).

**Theorem 5** (Theorem 3 of Jin et al. (2017)). *Assume that $F(\cdot)$ is $\ell$-smooth and $\rho$-Hessian Lipschitz. Then there exists an absolute constant $c_{\max}$ such that, for any $\delta_{\mathrm{pgd}} > 0$, $\varepsilon_{\mathrm{pgd}} \leq \ell^2/\rho$, $\Delta_{\mathrm{pgd}} \geq F(\boldsymbol{\gamma}_{\mathrm{pgd}}) - \inf_{\boldsymbol{\gamma} \in \mathbb{R}^{d+1}} F(\boldsymbol{\gamma})$ and constant $c_{\mathrm{pgd}} \leq c_{\max}$, with probability exceeding $1 - \delta_{\mathrm{pgd}}$, Algorithm 1 terminates within*

$$T \lesssim \frac{\ell\left[F\left(\boldsymbol{\gamma}_{\mathrm{pgd}}\right) - \inf_{\boldsymbol{\gamma} \in \mathbb{R}^{d+1}} F(\boldsymbol{\gamma})\right]}{\varepsilon_{\mathrm{pgd}}^2}\log^4\left(\frac{d\ell\Delta_{\mathrm{pgd}}}{\varepsilon_{\mathrm{pgd}}^2 \delta_{\mathrm{pgd}}}\right)$$

*iterations and the output $\boldsymbol{\gamma}^T$ satisfies*

$$\left\|\nabla F\left(\boldsymbol{\gamma}^T\right)\right\|_2 \leq \varepsilon_{\mathrm{pgd}} \qquad \text{and} \qquad \lambda_{\min}\left(\nabla^2 F\left(\boldsymbol{\gamma}\right)\right) \geq -\sqrt{\rho\varepsilon_{\mathrm{pgd}}}.$$

Let $\mathcal{A}$ denote this event where all of the geometric properties in Theorem 2 holds. When $\mathcal{A}$ happens, $\hat{L}_1$ is $\ell$-smooth and $\rho$-Hessian Lipschitz with

$$\ell = M_1 \qquad \text{and} \qquad \rho = M_1\left(1 \vee \frac{d\log(n/d)}{\sqrt{n}}\right).$$

Let $\boldsymbol{\gamma}_{\mathrm{pgd}} = \boldsymbol{0}$ and $\Delta_{\mathrm{pgd}} = 1/4$. Since $\inf_{\boldsymbol{\gamma} \in \mathbb{R} \times \mathbb{R}^d}\hat{L}_1(\boldsymbol{\gamma}) \geq 0$, we have

$$\Delta_{\mathrm{pgd}} = \hat{L}_1\left(\boldsymbol{\gamma}_{\mathrm{pgd}}\right) \geq \hat{L}_1\left(\boldsymbol{\gamma}_{\mathrm{pgd}}\right) - \inf_{\boldsymbol{\gamma} \in \mathbb{R} \times \mathbb{R}^d}\hat{L}_1\left(\boldsymbol{\gamma}\right).$$

In addition, we take $\delta^{\mathrm{pgd}} = n^{-11}$ and let

$$\varepsilon_{\mathrm{pgd}} = \sqrt{\frac{d}{n}\log\left(\frac{n}{d}\right)} \wedge \frac{\ell^2}{\rho} \wedge \frac{\eta^2}{\rho} \wedge \varepsilon.$$

Here $\varepsilon$ and $\eta$ are the constants defined in Theorem 2.

Recall that $M_1, \eta, \varepsilon \asymp 1$. Conditioned on the event $\mathcal{A}$, Theorem 5 asserts that with probability exceeding $1 - n^{-10}$, Algorithm 1 with parameters $\boldsymbol{\gamma}_{\mathrm{pgd}}, \ell, \rho, \varepsilon_{\mathrm{pgd}}, c_{\mathrm{pgd}}, \delta_{\mathrm{pgd}}$, and $\Delta_{\mathrm{pgd}}$ terminates within

$$T \lesssim \left(\frac{n}{d\log(n/d)} + \frac{d^2}{n}\log^2\left(\frac{n}{d}\right)\right)\log^4(nd) = \tilde{O}\left(\frac{n}{d} + \frac{d^2}{n}\right)$$

iterations, and the output $\hat{\gamma}$ satisfies

$$\left\|\nabla \hat{L}_1(\hat{\gamma})\right\|_2 \leq \varepsilon_{\mathrm{pgd}} \leq \sqrt{\frac{d}{n} \log\left(\frac{n}{d}\right)} \qquad \text{and} \qquad \lambda_{\min}\left(\nabla^2 \hat{L}_1(\hat{\gamma})\right) \geq -\sqrt{\rho \varepsilon_{\mathrm{pgd}}} \geq -\eta.$$

Then the desired result follows directly from $\mathbb{P}(\mathcal{A}) \geq 1 - C_1(d/n)^{C_2 d} - C_1 \exp(-C_2 n^{1/3})$ in Theorem 2.

# I  Proof of Corollary 1

Throughout the proof we suppose that the high-probability event

$$\min_{s=\pm 1}\left\|s\hat{\gamma} - c\gamma^{\mathrm{Bayes}}\right\|_2 \lesssim \sqrt{\frac{d}{n} \log\left(\frac{n}{d}\right)}$$

in Theorem 1 happens. Write $\hat{\gamma} = (\hat{\alpha}, \hat{\beta})$ and $\gamma^\star = (\alpha^\star, \beta^\star) = c\gamma^{\mathrm{Bayes}}$. Without loss of generality, assume that $\mu_0 = \mathbf{0}$, $\Sigma = I_d$, $\arg\min_{s=\pm 1}\|s\hat{\gamma} - \gamma^\star\|_2 = 1$ and $\hat{\beta}^\top \mu > 0$. Let $F$ be the cumulative distribution function of $Z = e_1^\top Z$.

For any $\gamma = (\alpha, \beta)$ with $\beta^\top \mu > 0$, we use $X = \mu Y + Z$ and the symmetry of $Z$ to derive that

$$\begin{aligned}
\mathcal{R}(\gamma) &= \frac{1}{2}\mathbb{P}\left(\alpha + \beta^\top(\mu + Z) < 0\right) + \frac{1}{2}\mathbb{P}\left(\alpha + \beta^\top(-\mu + Z) > 0\right) \\
&= \frac{1}{2}\mathbb{P}\left(\beta^\top Z < -\alpha - \beta^\top \mu\right) + \frac{1}{2}\mathbb{P}\left(\beta^\top Z > -\alpha + \beta^\top \mu\right) \\
&= \frac{1}{2}F\left(-\alpha/\|\beta\|_2 - (\beta/\|\beta\|_2)^\top \mu\right) + \frac{1}{2}F\left(\alpha/\|\beta\|_2 - (\beta/\|\beta\|_2)^\top \mu\right).
\end{aligned}$$

Define $\gamma_0 = (\alpha_0, \beta_0)$ with $\alpha_0 = \hat{\alpha}/\|\hat{\beta}\|_2$ and $\beta_0 = \hat{\beta}/\|\hat{\beta}\|_2$; $\gamma_1 = (\alpha_1, \beta_1)$ with $\alpha_1 = 0$ and $\beta_1 = \mu/\|\mu\|_2$. Recall that $\gamma^{\mathrm{Bayes}} = c(0, \mu)$ for some constant $c > 0$. We have

$$\mathcal{R}(\hat{\gamma}) - \mathcal{R}\left(\gamma^{\mathrm{Bayes}}\right) = \underbrace{\frac{1}{2}F\left(-\alpha_0 - \beta_0^\top \mu\right) - \frac{1}{2}F\left(-\alpha_1 - \beta_1^\top \mu\right)}_{E_1} + \underbrace{\frac{1}{2}F\left(\alpha_0 - \beta_0^\top \mu\right) - \frac{1}{2}F\left(\alpha_1 - \beta_1^\top \mu\right)}_{E_2}.$$

Using Taylor's Theorem, $\|p'\|_\infty \lesssim 1$ and $\|\mu\|_2 \lesssim 1$, one can arrive at

$$\left|E_1 - p\left(-\alpha_1 - \beta_1^\top \mu\right)\left(\alpha_1 - \alpha_0 + (\beta_1 - \beta_0)^\top \mu\right)\right| \lesssim \|\gamma_0 - \gamma_1\|_2^2,$$

$$\left|E_2 - p\left(\alpha_1 - \beta_1^\top \mu\right)\left(\alpha_0 - \alpha_1 + (\beta_1 - \beta_0)^\top \mu\right)\right| \lesssim \|\gamma_0 - \gamma_1\|_2^2,$$

From $\alpha_1 = 0$, $\beta_1 = \mu/\|\mu\|_2$ and $\|p\|_\infty \lesssim 1$ we obtain that

$$\begin{aligned}
\mathcal{R}(\hat{\gamma}) - \mathcal{R}\left(\gamma^{\mathrm{Bayes}}\right) &\lesssim |p(-\beta_1^\top \mu)[-\alpha_0 + (\beta_1 - \beta_0)^\top \mu] + p(-\beta_1^\top \mu)[\alpha_0 + (\beta_1 - \beta_0)^\top \mu]| + \|\gamma_0 - \gamma_1\|_2^2 \\
&\lesssim |(\beta_1 - \beta_0)^\top \beta_1| + \|\gamma_0 - \gamma_1\|_2^2.
\end{aligned}$$

Since $\beta_0$ and $\beta_1$ are unit vectors,

$$\|\beta_1 - \beta_0\|_2^2 = \|\beta_0\|_2^2 - 2\beta_0^\top \beta_1 + \|\beta_1\|_2^2 = 2(1 - \beta_0^\top \beta_1) = 2(\beta_1 - \beta_0)^\top \beta_1,$$

$$\mathcal{R}(\hat{\gamma}) - \mathcal{R}\left(\gamma^{\mathrm{Bayes}}\right) \lesssim \|\beta_1 - \beta_0\|_2^2 + \|\gamma_0 - \gamma_1\|_2^2 \lesssim \|\gamma_0 - \gamma_1\|_2^2. \qquad (41)$$

Note that $\|\hat{\beta} - \beta^\star\|_2 \leq \|\hat{\gamma} - \gamma^\star\|_2 \lesssim \sqrt{d/n \log(n/d)}$ and $\|\beta^\star\|_2 \asymp 1$. When $n/d$ is sufficiently large, we have $\|\hat{\beta}\|_2 \asymp 1$ and

$$\begin{aligned}
\|\beta_1 - \beta_0\|_2 &= \left\|\hat{\beta}/\|\hat{\beta}\|_2 - \beta^\star/\|\beta^\star\|_2\right\|_2 \lesssim \left\|\|\beta^\star\|_2 \hat{\beta} - \|\hat{\beta}\|_2 \beta^\star\right\|_2 \\
&\leq \left|\|\beta^\star\|_2 - \|\hat{\beta}\|_2\right|\|\hat{\beta}\|_2 + \|\hat{\beta}\|_2\|\hat{\beta} - \beta^\star\|_2 \lesssim \|\hat{\beta} - \beta^\star\|_2.
\end{aligned}$$

In addition, we also have $|\alpha_0 - \alpha_1| = |\alpha_0| = |\hat{\alpha}|/\|\hat{\beta}\|_2 \lesssim |\hat{\alpha}| = |\hat{\alpha} - \alpha^\star|$. As a result, $\|\gamma_0 - \gamma_1\|_2 \lesssim |\hat{\alpha} - \alpha^\star| + \|\beta_1 - \beta_0\|_2 \lesssim \|\hat{\gamma} - \gamma^\star\|_2$. Plugging these bounds into (41), we get

$$\mathcal{R}(\hat{\gamma}) - \mathcal{R}(\gamma^\star) \lesssim \|\hat{\gamma} - \gamma^\star\|_2^2 \lesssim \frac{d}{n} \log\left(\frac{n}{d}\right).$$

## J  Technical lemmas

**Lemma 7.** *Let $X$ be a random vector in $\mathbb{R}^{d+1}$ with $\mathbb{E}\|X\|_2^3 < \infty$. Then*

$$\sup_{u,v \in \mathbb{S}^d} \mathbb{E}(|u^\top X|^2 |v^\top X|) = \sup_{u \in \mathbb{S}^d} \mathbb{E}|u^\top X|^3.$$

*Proof.* It is easily seen that $\sup_{u,v \in \mathbb{S}^d} \mathbb{E}(|u^\top X|^2 |v^\top X|) \geq \sup_{u \in \mathbb{S}^d} \mathbb{E}|u^\top X|^3$. To prove the other direction, we first use Cauchy-Schwarz inequality to get

$$\mathbb{E}(|u^\top X|^2 |v^\top X|) = \mathbb{E}[|u^\top X|^{3/2}(|u^\top X|^{1/2}|v^\top X|)] \leq \mathbb{E}^{1/2}|u^\top X|^3 \cdot \mathbb{E}^{1/2}(|u^\top X| \cdot |v^\top X|^2).$$

By taking suprema we prove the claim. $\qquad\square$

**Lemma 8.** *Let $X$ be a random vector in $\mathbb{R}^{d+1}$ and $f \in C^2(\mathbb{R})$. Suppose that $\mathbb{E}\|X\|_2^3 < \infty$, $\sup_{x \in \mathbb{R}} |f''(x)| = F_2 < \infty$ and $f''$ is $F_3$-Lipschitz. Define $\bar{\mu} = \mathbb{E}X$. Then*

$$L_\lambda(\gamma) = \mathbb{E}f(\gamma^\top X) + \lambda(\gamma^\top \bar{\mu})^2/2$$

*exists for all $\gamma \in \mathbb{R}^{d+1}$ and $\lambda \geq 0$, and*

$$\sup_{\gamma_1 \neq \gamma_2} \frac{\|\nabla L_\lambda(\gamma_1) - \nabla L_\lambda(\gamma_2)\|_2}{\|\gamma_1 - \gamma_2\|_2} \leq F_2 \sup_{u \in \mathbb{S}^d} \mathbb{E}|u^\top X|^2 + \lambda\|\bar{\mu}\|_2^2,$$

$$\sup_{\gamma_1 \neq \gamma_2} \frac{|u^\top[\nabla^2 L_\lambda(\gamma_1) - \nabla^2 L_\lambda(\gamma_2)]u|}{\|\gamma_1 - \gamma_2\|_2} \leq F_3 \sup_{v \in \mathbb{S}^d} \mathbb{E}[(u^\top X)^2|v^\top X|], \qquad \forall u \in \mathbb{S}^{d-1},$$

$$\sup_{\gamma_1 \neq \gamma_2} \frac{\|\nabla^2 L_\lambda(\gamma_1) - \nabla^2 L_\lambda(\gamma_2)\|_2}{\|\gamma_1 - \gamma_2\|_2} \leq F_3 \sup_{u \in \mathbb{S}^d} \mathbb{E}|u^\top X|^3.$$

*In addition, if there exist nonnegative numbers $a, b$ and $c$ such that $\inf_{x \in \mathbb{R}} xf'(x) \geq -b$ and $\inf_{|x| \geq a} f'(x)\,\mathrm{sgn}(x) \geq c$, then*

$$\|\nabla L_\lambda(\gamma)\|_2 \geq c \inf_{u \in \mathbb{S}^d} \mathbb{E}|u^\top X| - \frac{ac+b}{\|\gamma\|_2}, \qquad \forall \gamma \neq \mathbf{0}.$$

*Proof.* Let $L(\gamma) = \mathbb{E}f(\gamma^\top X)$ and $R(\gamma) = (\gamma^\top \bar{\mu})^2/2$. Since $L_\lambda = L + \lambda R$, $\nabla^2 L(\gamma) = \mathbb{E}[XX^\top f''(\gamma^\top X)]$ and $\nabla^2 R(\gamma) = \bar{\mu}\bar{\mu}^\top$,

$$\sup_{\gamma_1 \neq \gamma_2} \frac{\|\nabla L_\lambda(\gamma_1) - \nabla L_\lambda(\gamma_2)\|_2}{\|\gamma_1 - \gamma_2\|_2} = \sup_{\gamma \in \mathbb{R}^{d+1}} \|\nabla^2 L_\lambda(\gamma)\|_2 = \sup_{\gamma \in \mathbb{R}^{d+1}} \sup_{u \in \mathbb{S}^d} u^\top \nabla^2 L_\lambda(\gamma) u$$

$$\leq F_2 \sup_{u \in \mathbb{S}^d} \mathbb{E}(u^\top X)^2 + \lambda\|\bar{\mu}\|_2^2.$$

For any $u \in \mathbb{S}^d$,

$$|u^\top[\nabla^2 L_\lambda(\gamma_1) - \nabla^2 L_\lambda(\gamma_2)]u| = \left|\mathbb{E}[(u^\top X)^2 f''(\gamma_1^\top X)] - \mathbb{E}[(u^\top X)^2 f''(\gamma_2^\top X)]\right|$$

$$\leq \mathbb{E}[(u^\top X)^2|f''(\gamma_1^\top X) - f''(\gamma_2^\top X)|]$$

$$\leq F_3\mathbb{E}[(u^\top X)^2|(\gamma_1 - \gamma_2)^\top X|]$$

$$\leq F_3\|\gamma_1 - \gamma_2\|_2 \sup_{v \in \mathbb{S}^d} \mathbb{E}[(u^\top X)^2|v^\top X|].$$

As a result,

$$\sup_{\gamma_1 \neq \gamma_2} \frac{\|\nabla^2 L_\lambda(\gamma_1) - \nabla^2 L_\lambda(\gamma_2)\|_2}{\|\gamma_1 - \gamma_2\|_2} = \sup_{\gamma_1 \neq \gamma_2} \frac{\sup_{u \in \mathbb{S}^d}|u^\top[\nabla^2 L_\lambda(\gamma_1) - \nabla^2 L_\lambda(\gamma_2)]u|}{\|\gamma_1 - \gamma_2\|_2}$$

$$= \sup_{u \in \mathbb{S}^d} \sup_{\gamma_1 \neq \gamma_2} \frac{|u^\top[\nabla^2 L_\lambda(\gamma_1) - \nabla^2 L_\lambda(\gamma_2)]u|}{\|\gamma_1 - \gamma_2\|_2}$$

$$\leq \sup_{u \in \mathbb{S}^d}\{F_3 \sup_{v \in \mathbb{S}^d} \mathbb{E}[(u^\top X)^2|v^\top X|]\} = F_3 \sup_{u \in \mathbb{S}^d} \mathbb{E}|u^\top X|^3,$$

where the last equality follows from Lemma 7.

We finally come to the lower bound on $\|\nabla L_\lambda(\boldsymbol{\gamma})\|_2$. Note that $\|\nabla L_\lambda(\boldsymbol{\gamma})\|_2\|\boldsymbol{\gamma}\|_2 \geq \langle \boldsymbol{\gamma}, \nabla L_\lambda(\boldsymbol{\gamma})\rangle$, $\nabla L(\boldsymbol{\gamma}) = \mathbb{E}[\boldsymbol{X}f'(\boldsymbol{X}^\top\boldsymbol{\gamma})]$ and $\nabla R(\boldsymbol{\gamma}) = (\boldsymbol{\gamma}^\top\bar{\boldsymbol{\mu}})\bar{\boldsymbol{\mu}}$. The condition $\inf_{|x|\geq a} f'(x)\,\mathrm{sgn}(x) \geq c$ implies that $xf'(x) \geq c|x|$ when $|x| \geq a$. By this and $\inf_{x\in\mathbb{R}} xf'(x) \geq -b$,

$$
\begin{aligned}
\langle \boldsymbol{\gamma}, \nabla L(\boldsymbol{\gamma})\rangle = \mathbb{E}[\boldsymbol{X}^\top\boldsymbol{\gamma}f'(\boldsymbol{X}^\top\boldsymbol{\gamma})] &= \mathbb{E}[\boldsymbol{X}^\top\boldsymbol{\gamma}f'(\boldsymbol{X}^\top\boldsymbol{\gamma})\mathbf{1}_{\{|\boldsymbol{X}^\top\boldsymbol{\gamma}|\geq a\}}] + \mathbb{E}[\boldsymbol{X}^\top\boldsymbol{\gamma}f'(\boldsymbol{X}^\top\boldsymbol{\gamma})\mathbf{1}_{\{|\boldsymbol{X}^\top\boldsymbol{\gamma}|<a\}}] \\
&\geq c\mathbb{E}(|\boldsymbol{X}^\top\boldsymbol{\gamma}|\mathbf{1}_{\{|\boldsymbol{X}^\top\boldsymbol{\gamma}|\geq a\}}) - b = c\mathbb{E}|\boldsymbol{X}^\top\boldsymbol{\gamma}| - c\mathbb{E}(|\boldsymbol{X}^\top\boldsymbol{\gamma}|\mathbf{1}_{\{|\boldsymbol{X}^\top\boldsymbol{\gamma}|<a\}}) - b \\
&\geq c\mathbb{E}|\boldsymbol{X}^\top\boldsymbol{\gamma}| - (ac+b) \geq \|\boldsymbol{\gamma}\|_2 c \inf_{\boldsymbol{u}\in\mathbb{S}^d}\mathbb{E}|\boldsymbol{u}^\top\boldsymbol{X}| - (ac+b).
\end{aligned}
$$

In addition, we also have $\langle \boldsymbol{\gamma}, \nabla R(\boldsymbol{\gamma})\rangle = (\boldsymbol{\gamma}^\top\bar{\boldsymbol{\mu}})^2 \geq 0$. Then the lower bound directly follows. $\square$

**Lemma 9.** *There exists a continuous function $\varphi : (0, +\infty)^2 \to (0, +\infty)$ that is non-increasing in the first argument and non-decreasing in the second argument, such that for any nonzero sub-Gaussian random variable $X$, $\mathbb{E}|X| \geq \varphi(\|X\|_{\psi_2}, \mathbb{E}X^2)$.*

*Proof.* For any $t > 0$,

$$
\mathbb{E}|X| \geq \mathbb{E}(|X|\mathbf{1}_{\{|X|\leq t\}}) \leq t^{-1}\mathbb{E}(X^2\mathbf{1}_{\{|X|\leq t\}}) = t^{-1}[\mathbb{E}X^2 - \mathbb{E}(X^2\mathbf{1}_{\{|X|>t\}})].
$$

By Cauchy-Schwarz inequality and the sub-Gaussian property (Vershynin, 2010), there exist constants $C_1, C_2 > 0$ such that

$$
\mathbb{E}(X^2\mathbf{1}_{\{|X|>t\}}) \leq \mathbb{E}^{1/2}X^4 \cdot \mathbb{P}^{1/2}(|X| > t) \leq C_1\|X\|_{\psi_2}^2 e^{-C_2 t^2/\|X\|_{\psi_2}^2}.
$$

By taking $\varphi(\|X\|_{\psi_2}, \mathbb{E}X^2) = \sup_{t>0} t^{-1}(\mathbb{E}X^2 - C_1\|X\|_{\psi_2}^2 e^{-C_2 t^2/\|X\|_{\psi_2}^2})$ we finish the proof, as the required monotonicity is obvious. $\square$

**Lemma 10.** *Let $\{X_{ni}\}_{n\geq 1, i\in[n]}$ be an array of random variables where for any $n$, $\{X_{ni}\}_{i=1}^n$ are i.i.d. sub-Gaussian random variables with $\|X_{n1}\|_{\psi_2} \leq 1$. Fix some constant $a \geq 2$, define $S_n = \frac{1}{n}\sum_{i=1}^n |X_{ni}|^a$ and let $\{r_n\}_{n=1}^\infty$ be a deterministic sequence satisfying $\log n \leq r_n \leq n$. We have*

$$
\begin{aligned}
S_n - \mathbb{E}|X_{n1}|^a &= O_{\mathbb{P}}(r_n^{(a-1)/2}/\sqrt{n};\; r_n), \\
S_n &= O_{\mathbb{P}}(\max\{1, r_n^{(a-1)/2}/\sqrt{n}\};\; r_n).
\end{aligned}
$$

*Proof.* Define $R_{nt} = t\sqrt{r_n}$ and $S_{nt} = \frac{1}{n}\sum_{i=1}^n |X_{ni}|^a\mathbf{1}_{\{|X_{ni}|\leq R_{nt}\}}$ for $n, t \geq 1$. For any $p \geq 1$, we have $2p \geq 2 > 1$ and $(2p)^{-1/2}\mathbb{E}^{1/(2p)}|X_{ni}|^{2p} \leq \|X_{ni}\|_{\psi_2} \leq 1$. Hence

$$
\begin{aligned}
\mathbb{E}(|X_{ni}|^a\mathbf{1}_{\{|X_{ni}|\leq R_{nt}\}})^p = \mathbb{E}(|X_{ni}|^{ap}\mathbf{1}_{\{|X_{ni}|\leq R_{nt}\}}) &= \mathbb{E}(|X_{ni}|^{2p}|X_{ni}|^{(a-2)p}\mathbf{1}_{\{|X_{ni}|\leq R_{nt}\}}) \\
&\leq \mathbb{E}|X_{ni}|^{2p}R_{nt}^{(a-2)p} \leq [(2p)^{1/2}\|X_{ni}\|_{\psi_2}]^{2p}R_{nt}^{(a-2)p} \leq (2pR_{nt}^{a-2})^p
\end{aligned}
$$

and $\||X_{ni}|^a\mathbf{1}_{\{|X_{ni}|\leq R_{nt}\}}\|_{\psi_1} \leq 2R_{nt}^{a-2}$. By the Bernstein-type inequality in Proposition 5.16 of Vershynin (2010), there exists a constant $c$ such that

$$
\mathbb{P}(|S_{nt} - \mathbb{E}S_{nt}| \geq s) \leq 2\exp\left[-cn\left(\frac{s^2}{R_{nt}^{2(a-2)}} \wedge \frac{s}{R_{nt}^{a-2}}\right)\right], \qquad \forall t \geq 0,\; s \geq 0. \qquad (42)
$$

Take $t \geq 1$ and $s = t^{a-1}r_n^{(a-1)/2}/\sqrt{n}$. We have

$$
\begin{aligned}
\frac{s}{R_{nt}^{a-2}} &= \frac{t^{a-1}r_n^{(a-1)/2}/\sqrt{n}}{t^{a-2}r_n^{(a-2)/2}} = t\sqrt{r_n/n}, \\
\frac{s^2}{R_{nt}^{2(a-2)}} \wedge \frac{s}{R_{nt}^{a-2}} &= \frac{t^2 r_n}{n} \wedge \frac{t\sqrt{r_n}}{\sqrt{n}} \geq \frac{tr_n}{n},
\end{aligned}
$$

where the last inequality is due to $r_n/n \leq 1 \leq t$. By (42),

$$
\mathbb{P}(|S_{nt} - \mathbb{E}S_{nt}| \geq t^{a-1}r_n^{(a-1)/2}/\sqrt{n}) \leq 2e^{-cr_n t}, \qquad \forall t \geq 1. \qquad (43)
$$

By Cauchy-Schwarz inequality and $\|X_{n1}\|_{\psi_2} \le 1$, there exist $C_1, C_2 > 0$ such that

$$0 \le \mathbb{E}S_n - \mathbb{E}S_{nt} = \mathbb{E}(|X_{n1}|^a \mathbf{1}_{\{|X_{n1}|>t\sqrt{r_n}\}}) \le \mathbb{E}^{1/2}|X_{n1}|^{2a} \cdot \mathbb{P}^{1/2}(|X_{n1}| > t\sqrt{r_n}) \le C_1 e^{-C_2 t^2 r_n}$$

holds for all $t \ge 0$. Since $r_n \ge \log n$, there exists a constant $C > 0$ such that $C_1 e^{-C_2 t^2 r_n} \le t^{a-1} r_n^{(a-1)/2}/\sqrt{n}$ as long as $t \ge C$. Hence (43) forces

$$\mathbb{P}(|S_{nt} - \mathbb{E}S_n| \ge 2t^{a-1} r_n^{(a-1)/2}/\sqrt{n}) \le \mathbb{P}(|S_{nt} - \mathbb{E}S_{nt}| + |\mathbb{E}S_{nt} - \mathbb{E}S_n| \ge 2t^{a-1} r_n^{(a-1)/2}/\sqrt{n})$$

$$\le \mathbb{P}(|S_{nt} - \mathbb{E}S_{nt}| \ge t^{a-1} r_n^{(a-1)/2}/\sqrt{n}) \le 2e^{-cr_n t}, \qquad \forall t \ge C.$$

Note that

$$\mathbb{P}(|S_n - \mathbb{E}S_n| \ge 2t^{a-1} r_n^{(a-1)/2}/\sqrt{n}) \tag{44}$$

$$\le \mathbb{P}(|S_n - \mathbb{E}S_n| \ge 2t^{a-1} r_n^{(a-1)/2}/\sqrt{n}, \; S_n = S_{nt}) + \mathbb{P}(S_n \ne S_{nt})$$

$$\le \mathbb{P}(|S_{nt} - \mathbb{E}S_n| \ge 2qt^{a-1} r_n^{(a-1)/2}/\sqrt{n}) + \mathbb{P}(S_n \ne S_{nt})$$

$$\le 2e^{-cr_n t} + \mathbb{P}\left(\max_{i \in [n]} |X_{ni}| > t\sqrt{r_n}\right), \qquad \forall t \ge C. \tag{45}$$

Since $\|X_{ni}\|_{\psi_2} \le 1$, there exist constants $C_1', C_2' > 0$ such that

$$\mathbb{P}(|X_{ni}| \ge t) \le C_1' e^{-C_2' t^2}, \qquad \forall n \ge 1, \; i \in [n], \; t \ge 0.$$

By union bounds,

$$\mathbb{P}\left(\max_{i \in [n]} |X_{ni}| > t\sqrt{r_n}\right) \le n C_1' e^{-C_2' t^2 r_n} = C_1' e^{\log n - C_2' t^2 r_n}, \qquad \forall t \ge 0.$$

When $t \ge \sqrt{2/C_2'}$, we have $C_2' t^2 r_n \ge 2r_n \ge 2\log n$ and thus $\log n - C_2' t^2 r_n \le -C_2' t^2 r_n/2$. Then (45) leads to

$$\mathbb{P}(|S_n - \mathbb{E}S_n| \ge 2t^{a-1} r_n^{(a-1)/2}/\sqrt{n}) \le 2e^{-cr_n t} + C_1' e^{-C_2' r_n t^2/2}, \qquad \forall t \ge C \vee \sqrt{2/C_2'}.$$

This shows $S_n - \mathbb{E}|X_{n1}|^a = S_n - \mathbb{E}S_n = O_{\mathbb{P}}(r_n^{(a-1)/2}/\sqrt{n}; \; r_n)$. The proof is finished by $\mathbb{E}|X_{n1}|^a \lesssim 1$. $\qquad\square$

**Lemma 11.** *Suppose that $\{X_i\}_{i=1}^n \subseteq \mathbb{R}^{d+1}$ are independent random vectors, $\max_{i \in [n]} \|X_i\|_{\psi_2} \le 1$ and $n \ge md \ge \log n$ for some $m \ge 1$. We have*

$$\sup_{\boldsymbol{u} \in \mathbb{S}^d} \frac{1}{n} \sum_{i=1}^n |\boldsymbol{u}^\top \boldsymbol{X}_i|^2 = O_{\mathbb{P}}(1; \; n),$$

$$\sup_{\boldsymbol{u} \in \mathbb{S}^d} \frac{1}{n} \sum_{i=1}^n (\boldsymbol{v}^\top \boldsymbol{X}_i)^2 |\boldsymbol{u}^\top \boldsymbol{X}_i| = O_{\mathbb{P}}(1; \; n^{1/3}), \qquad \forall \boldsymbol{v} \in \mathbb{S}^d,$$

$$\sup_{\boldsymbol{u} \in \mathbb{S}^d} \frac{1}{n} \sum_{i=1}^n |\boldsymbol{u}^\top \boldsymbol{X}_i|^3 = O_{\mathbb{P}}\left(\max\{1, \; md/\sqrt{n}\}; \; md\right).$$

*Proof.* From $2^{-1/2} \mathbb{E}^{1/2}(\boldsymbol{u}^\top \boldsymbol{X})^2 \le \|\boldsymbol{u}^\top \boldsymbol{X}\|_{\psi_2} \le 1, \forall \boldsymbol{u} \in \mathbb{S}^d$ we get $\mathbb{E}(\boldsymbol{X}\boldsymbol{X}^\top) \preceq 2\boldsymbol{I}$. Since $n \ge d+1$, Remark 5.40 in Vershynin (2010) asserts that

$$\sup_{\boldsymbol{u} \in \mathbb{S}^d} \frac{1}{n} \sum_{i=1}^n |\boldsymbol{u}^\top \boldsymbol{X}_i|^2 = \left\|\frac{1}{n} \sum_{i=1}^n \boldsymbol{X}_i \boldsymbol{X}_i^\top\right\|_2 \le \left\|\frac{1}{n} \sum_{i=1}^n \boldsymbol{X}_i \boldsymbol{X}_i^\top - \mathbb{E}(\boldsymbol{X}\boldsymbol{X}^\top)\right\|_2 + \|\mathbb{E}(\boldsymbol{X}\boldsymbol{X}^\top)\|_2 = O_{\mathbb{P}}(1; \; n).$$

For any $\boldsymbol{u}, \boldsymbol{v} \in \mathbb{S}^d$, the Cauchy-Schwarz inequality forces

$$\frac{1}{n} \sum_{i=1}^n (\boldsymbol{v}^\top \boldsymbol{X}_i)^2 |\boldsymbol{u}^\top \boldsymbol{X}_i| \le \left(\frac{1}{n} \sum_{i=1}^n (\boldsymbol{v}^\top \boldsymbol{X}_i)^4\right)^{1/2} \left(\frac{1}{n} \sum_{i=1}^n (\boldsymbol{u}^\top \boldsymbol{X}_i)^2\right)^{1/2},$$

$$\sup_{\boldsymbol{u} \in \mathbb{S}^d} \frac{1}{n} \sum_{i=1}^n (\boldsymbol{v}^\top \boldsymbol{X}_i)^2 |\boldsymbol{u}^\top \boldsymbol{X}_i| \le \left(\frac{1}{n} \sum_{i=1}^n (\boldsymbol{v}^\top \boldsymbol{X}_i)^4\right)^{1/2} O_{\mathbb{P}}(1; \; n).$$

Since $\{\boldsymbol{v}^\top \boldsymbol{X}_i\}_{i=1}^n$ are i.i.d. sub-Gaussian random variables and $\|\boldsymbol{v}^\top \boldsymbol{X}_i\|_{\psi_2} \leq 1$, Lemma 10 with $a = 4$ and $r_n = n^{1/3}$ yields $\frac{1}{n}\sum_{i=1}^n (\boldsymbol{v}^\top \boldsymbol{X}_i)^4 = O_{\mathbb{P}}(1; \ n^{1/3})$. Hence $\sup_{\boldsymbol{u}\in\mathbb{S}^d} \frac{1}{n}\sum_{i=1}^n (\boldsymbol{v}^\top \boldsymbol{X}_i)^2 |\boldsymbol{u}^\top \boldsymbol{X}_i| = O_{\mathbb{P}}(1; \ n^{1/3})$.

To prove the last equation in Lemma 11, define $\boldsymbol{Z}_i = \boldsymbol{X}_i - \mathbb{E}\bar{\boldsymbol{X}}_i$. From $\|\boldsymbol{Z}_i\|_{\psi_2} = \|\boldsymbol{X}_i - \mathbb{E}\bar{\boldsymbol{X}}_i\|_{\psi_2} \leq 2\|\boldsymbol{X}_i\|_{\psi_2} \leq 2$ we get $\sup_{\boldsymbol{u}\in\mathbb{S}^d} \frac{1}{n}\sum_{i=1}^n |\boldsymbol{u}^\top \boldsymbol{Z}_i|^2 = O_{\mathbb{P}}(1; \ n)$. For $\boldsymbol{u} \in \mathbb{S}^d$,

$$
\begin{aligned}
|\boldsymbol{u}^\top \boldsymbol{X}_i|^3 &= |\boldsymbol{u}^\top \boldsymbol{Z}_i|^3 + (|\boldsymbol{u}^\top \boldsymbol{X}_i| - |\boldsymbol{u}^\top \boldsymbol{Z}_i|)(|\boldsymbol{u}^\top \boldsymbol{X}_i|^2 + |\boldsymbol{u}^\top \boldsymbol{X}_i|\cdot|\boldsymbol{u}^\top \boldsymbol{Z}_i| + |\boldsymbol{u}^\top \boldsymbol{Z}_i|^2) \\
&\leq |\boldsymbol{u}^\top \boldsymbol{Z}_i|^3 + |\boldsymbol{u}^\top(\boldsymbol{X}_i - \boldsymbol{Z}_i)|(|\boldsymbol{u}^\top \boldsymbol{X}_i|^2 + |\boldsymbol{u}^\top \boldsymbol{X}_i|\cdot|\boldsymbol{u}^\top \boldsymbol{Z}_i| + |\boldsymbol{u}^\top \boldsymbol{Z}_i|^2) \\
&\leq |\boldsymbol{u}^\top \boldsymbol{Z}_i|^3 + |\boldsymbol{u}^\top \mathbb{E}\bar{\boldsymbol{X}}_i| \cdot \frac{3}{2}(|\boldsymbol{u}^\top \boldsymbol{X}_i|^2 + |\boldsymbol{u}^\top \boldsymbol{Z}_i|^2) \leq |\boldsymbol{u}^\top \boldsymbol{Z}_i|^3 + \frac{3}{2}(|\boldsymbol{u}^\top \boldsymbol{X}_i|^2 + |\boldsymbol{u}^\top \boldsymbol{Z}_i|^2),
\end{aligned}
$$

where the last inequality is due to $|\boldsymbol{u}^\top \mathbb{E}\bar{\boldsymbol{X}}_i| \leq \|\mathbb{E}\bar{\boldsymbol{X}}_i\|_2 \leq \|\boldsymbol{X}_i\|_{\psi_2} \leq 1$. Hence

$$
\sup_{\boldsymbol{u}\in\mathbb{S}^d} \frac{1}{n}\sum_{i=1}^n |\boldsymbol{u}^\top \boldsymbol{X}_i|^3 \leq \sup_{\boldsymbol{u}\in\mathbb{S}^d} \frac{1}{n}\sum_{i=1}^n |\boldsymbol{u}^\top \boldsymbol{Z}_i|^3 + O_{\mathbb{P}}(1; \ n). \tag{46}
$$

Define $S(\boldsymbol{u}) = \frac{1}{n}\sum_{i=1}^n |\boldsymbol{u}^\top \boldsymbol{Z}_i|^3$ for $\boldsymbol{u} \in \mathbb{S}^d$. We will invoke Theorem 1 in Wang (2019) to control $\sup_{\boldsymbol{u}\in\mathbb{S}^d} S(\boldsymbol{u})$.

1. For any $\boldsymbol{u} \in \mathbb{S}^d$, $\{\boldsymbol{u}^\top \boldsymbol{Z}_i\}_{i=1}^n$ are i.i.d. and $\|\boldsymbol{u}^\top \boldsymbol{Z}_i\|_{\psi_2} \leq 1$. Lemma 10 with $a = 3$ and $r_n = md$ yields

$$
\{S(\boldsymbol{u})\}_{\boldsymbol{u}\in\mathbb{S}^d} = O_{\mathbb{P}}(\max\{1, md/\sqrt{n}\}; \ md).
$$

2. According to Lemma 5.2 in Vershynin (2010), for $\varepsilon = 1/6$ there exists an $\varepsilon$-net $\mathcal{N}$ of $\mathbb{S}^d$ with cardinality at most $(1 + 2/\varepsilon)^d = 13^d$. Hence $\log|\mathcal{N}| \lesssim md$.

3. For any $x, y \in \mathbb{R}$, we have $||x| - |y|| \leq |x - y|$, $2|xy| \leq x^2 + y^2$ and

$$
\left||x|^3 - |y|^3\right| \leq ||x| - |y||\,(x^2 + |xy| + y^2) \leq \frac{3}{2}|x - y|(x^2 + y^2).
$$

Hence for any $\boldsymbol{u}, \boldsymbol{v} \in \mathbb{S}^d$,

$$
\begin{aligned}
|S(\boldsymbol{u}) - S(\boldsymbol{v})| &\leq \frac{1}{n}\sum_{i=1}^n \left||\boldsymbol{u}^\top \boldsymbol{Z}_i|^3 - |\boldsymbol{v}^\top \boldsymbol{Z}_i|^3\right| \leq \frac{3}{2}\cdot\frac{1}{n}\sum_{i=1}^n |(\boldsymbol{u}-\boldsymbol{v})^\top \boldsymbol{Z}_i|(|\boldsymbol{u}^\top \boldsymbol{Z}_i|^2 + |\boldsymbol{v}^\top \boldsymbol{Z}_i|^2) \\
&\leq 3\|\boldsymbol{u}-\boldsymbol{v}\|_2 \sup_{\boldsymbol{w}_1, \boldsymbol{w}_2 \in \mathbb{S}^d} \frac{1}{n}\sum_{i=1}^n |\boldsymbol{w}_1^\top \boldsymbol{Z}_i| \cdot |\boldsymbol{w}_2^\top \boldsymbol{Z}_i|^2 = \frac{1}{2\varepsilon}\|\boldsymbol{u}-\boldsymbol{v}\|_2 \sup_{\boldsymbol{w}\in\mathbb{S}^d} S(\boldsymbol{w}).
\end{aligned}
$$

where the last inequality follows from $\varepsilon = 1/6$ and Lemma 7.

Theorem 1 in Wang (2019) then asserts that $\sup_{\boldsymbol{u}\in\mathbb{S}^d} S(\boldsymbol{u}) = O_{\mathbb{P}}(\max\{1, md/\sqrt{n}\}; \ md)$. We finish the proof using (46). $\qquad \square$