[Reviews · NeurIPS 2020]

Review 1

Summary and Contributions: This paper introduces an unsupervised method for estimating the optimal separator for a 2-component ellipsoidal mixture. The authors propose a sensible objective for learning the coefficient vector \beta which is orthogonal to the optimal separator, based on the reasoning that the projection of the underlying random variable on \beta concentrates around the points +-1 (after scaling and translating). They go on to show estimation guarantees under a balanced model for the output of a modified gradient descent algorithm. Some brief experiments on a subset of the MNIST-fashion data are presented

Strengths: The strengths of the work lie in the theoretical aspects of the proposed approach. Although the model they address theoretically is somewhat restrictive, statistical estimation guarantees of related works are scarce if even existent. Furthermore, the approach taken by the authors in their theory is intuitive and could likely be used to extend their result to some other projection methods for clustering.

Weaknesses: The main weakness of the paper is the limited empirical documentation of the practical performance of the method. The authors show that their method outperforms a few alternatives which are not expected to perform well on the single test bed considered. Although the proposed method substantially outperforms those other methods, a more relevant comparison would be with [1]; [2]; [3]; another projection based method; or a mixture model capable of handling elongated clusters such as [4]. [1] Peña, Daniel, and Francisco J. Prieto. "Cluster identification using projections." Journal of the American Statistical Association 96.456 (2001): 1433-1445. [2] Pavlidis, Nicos G., David P. Hofmeyr, and Sotiris K. Tasoulis. "Minimum density hyperplanes." The Journal of Machine Learning Research 17.1 (2016): 5414-5446. [3] Ding, Chris, and Tao Li. "Adaptive dimension reduction using discriminant analysis and k-means clustering." Proceedings of the 24th international conference on Machine learning. 2007. [4] Fraley, Chris, and Adrian E. Raftery. "MCLUST: Software for model-based cluster analysis." Journal of classification 16.2 (1999): 297-306.

Correctness: The theoretical claims seem to be correct, however the empirical methodology is arguably incorrect in the manners discussed above.

Clarity: The paper is nicely and clearly written

Relation to Prior Work: For the most part the review of relevant literature is good, however aside from [1] in "Weaknesses:..." above the authors seem to have missed the literature on projection pursuit methods for clustering, which seem to be very closely related.

Reproducibility: Yes

Additional Feedback: EDIT: The authors convincingly addressed the concerns I had regarding the relevance of the empirical results. While I would prefer more than a single example if possible, I appreciate that space is at a premium in such papers and there is considerable space required to cover the theoretical aspects. Overall I like the paper and I think the approach taken to the theory is nice. Using a version of gradient descent which allows the algorithm to escape subtle local minima in the empirical loss surface which are not present in the true loss surface is, as far as I can tell, novel in the context of projection methods for clustering. I do think further and more relevant empirical results would vastly improve the paper, though, and encourage the authors to investigate other methods for comparison and also more than a single data set.


Review 2

Summary and Contributions: The paper proposes a non-convex program for clustering an elliptical mixture model with two components. It also provides a perturbed gradient descent method for solving the optimization problem. At last, it provides theoretical guarantees for the convergence rate and accuracy.

Strengths: The theoretical contribution is valuable for the NeurIPS community and distinguishes this work from other works that try to address this particular problem, which is of direct interest to the NeurIPS community. Furthermore, the objective function used in the paper is interesting and the choice of perturbed gradient descent is natural. The overall methodology seems novel to me and could potentially impact other applications (for example, the paper mentions a relationship to phase retrieval). I find the approach fresh and creative.

Weaknesses: The experimental part can be improved (though the strength of the paper is in its theory). First of all, the Fashion MNIST dataset may not be the best choice for showing the effectiveness of the proposed method. It is unclear to me why the model described by (1) is natural for this dataset. Second of all, the paper only compares CURE with very simple methods. More benchmarks should be considered. There are also methods that do not tend to solve (1) but can perform well; for instance, t-SNE can provide good embedding for Fashion-MNIST.

Correctness: The idea of the proof seems convincing and interesting. However, the details of the long proof are in the appendix and I did not have the time to verify all of them.

Clarity: Generally, the writing is clear. The motivation for the method is nicely explained and the idea of the proof is properly introduced. Section 2.3 is a bit vague and not so convincing. The theoretical part does not seem to fit to this general setting and there is no empirical evidence for this part of the work. The authors may spend more effort explaining this generalization (with a bit more details) and may think of a relevant numerical experiment. I also believe that some more details (possibly with references to supplemental material) can help with broader readership. For example, more details on Bayes optimality and exemplifying it for the simple example will be useful (even though it is known to the expert, more details will help a larger audience). Also, it is helpful to explain more clearly the general setting (Section 2.3) and show with detailed explanation (possibly also in appendix) why the main case is a special case of the general one. While I could use the few provided details to conclude the latter fact, I believe that more details are needed for the general reader. For example, mu_1 and mu_2 in this part were never defined (though I guessed what they were by looking at the unnumbered equation after line 169 of the earlier section, but it may not be clear at all to a general reader).

Relation to Prior Work: Generally, the paper nicely surveys prior related works. However, it seems that there is a bit of a simplistic approach when comparing with other works. For example, I mentioned the issue of comparing with few simple methods, while there are also other benchmarks. Another example, is the discussion about the problem with the transformation by PCA, however, one may use more robust linear embeddings (see e.g., 10.1109/JPROC.2018.2853141) or nonlinear embeddings (such as t-SNE).

Reproducibility: Yes

Additional Feedback: One may scale the function f or equivalently add the regularization parameter lambda in front of the second term of the energy function. According to the appendix, there is a preference for lambda geq 1. Are there any other restrictions on lambda and why do you formulate the result with only lambda = 1? In practice, is there any advantage to tuning lambda? Misc comments: Line 32: discriminantive -> discriminative Line 129: sub-gaussian -> sub-Gaussian in some places language can be smoothed, e.g., "which answers for the failure of PCA" In the appendix, the parameter lambda has different meanings (regularization parameter of two different methods as well as eigenvalues) I also mentioned above that mu_1 and mu_2 were not defined. I like the paper, there is something very creative about it. I have various comments for improving it, but they do not change my strong impression of the original and creative perspective. I also think that the main contribution is the theory, so one can be more forgiving with other issues. Additional comments after review: The authors wrote a nice rebuttal. My initial review realized that despite the weaknesses I previously indicated the paper has a nice and creative theoretical contribution and took into account that in the future more experiments and generalization of the theory can be done and that the presentation will be improved. Thus I don't think I should further upgrade my score. I am also unsure how to rank it among previous NeurIPS papers (I have a preference to carefully reading substantial journal papers), but it is definitely a good paper and should be accepted.


Review 3

Summary and Contributions: I thank the authors for their feedback. --- The authors present a tractable algorithm for binary clustering that learns the Bayes optimal clustering/classification rule. The authors present a novel loss function for binary clustering which is consistent, and analyzes the empirical loss function's landscape, showing that perturbed gradient descent converges to the global optimum.

Strengths: The authors present a non-convex optimization problem for binary clustering which is tractable. - The authors present a novel loss function for binary clustering which is consistent. - They then analyze the empirical loss function's landscape, and show that perturbed gradient descent converges to the global optimum.

Weaknesses: How would we extend this work beyond binary clustering? For example, what if we want to learn Bayes optimal classification rules in k-clustering (k >= 3)? In classification this is easy to extend, but seems much harder if we have >= 3 clusters.

Correctness: Yes

Clarity: Yes

Relation to Prior Work: Yes

Reproducibility: Yes

Additional Feedback:


Review 4

Summary and Contributions: This paper presents a discriminative model for binary clustering, named CURE. The goal is to cluster the data without requiring the clusters to be spherical. The authors propose to learn an affine transform and project the data to a one-dimensional point cloud, where the projected data is concentrated around -1 and 1 (for binary clustering), with regularization on the balanced size of two clusters. The proposed non-convex problem is optimized via previous methods (Jin et al., 2017). ================ [After reading the rebuttal] I have read the authors' feedback and other reviews. The rebuttal addressed my concerns on empirical evaluation. I updated my score to positive.

Strengths: The idea of learning an affine transform of the data to address the non-spherical clustering is valid. The authors provide theoretical analysis of the convergence and misclassification rate of the proposed method.

Weaknesses: It would be helpful to include some discussion on more up-to-date related methods. The methods discussed in the related works are before 2017. There are recent works on clustering of non-spherical data, such as Kushnir, Dan, Shirin Jalali, and Iraj Saniee. "Towards Clustering High-dimensional Gaussian Mixture Clouds in Linear Running Time." The 22nd International Conference on Artificial Intelligence and Statistics. 2019. In the experiments, the proposed method is compared with K-means and spectral methods. Some more recent related works should be included in the comparison. The claims are not well supported with the empirical results. For example, it would be helpful to include results on computational time to support the efficiency of the proposed method.

Correctness: The claims look correct to me. The empirical section needs some more results to support the claims. See above for details.

Clarity: The paper is well written.

Relation to Prior Work: More recent related works should be discussed in the paper.

Reproducibility: No

Additional Feedback:

[Author Response · NeurIPS 2020]

We thank all the reviewers for their very helpful comments and suggestions. Please find below our responses.

**Literature review**

We will rewrite the existing paragraph on Projection Pursuit (PP) and discuss PP-based clustering methods that have
direct relevance. In particular, we will review criteria for cluster identification including kurtosis (Peña and Prieto, 2001
and follow-up works), first absolute moment and skewness (Verzelen and Arias-Castro, 2017) and relate them to CURE.
We will also discuss algorithms proposed in those papers. In addition, we will review more recent clustering methods
for non-spherical data, such as the one (Kushnir et al., 2019) mentioned by Reviewer 4 based on random projections.

**Generalization to multi-class nonlinear clustering**

While our theories focus on the two-component elliptical mixture model, the idea of CURE generalizes to multi-class
scenarios, allowing for nonlinear discriminant functions (i.e. feature mappings). We will revise the brief and somewhat
abstract discussion in Section 2.3 and provide more intuitions to the general audience. As is pointed out by Reviewer 2,
we will elaborate why the linear CURE approach is a special case. Due to space constraints, we will defer the most of
these details to the supplementary.

We will add a new numerical example (to the supplementary) with the first 4
classes in Fashion-MNIST (T-shirt/top, Trouser, Pullover, Dress). All details
will be included in the final version. In short, we use Wasserstein-1 distance as
the discrepancy measure $D$ for uncoupled regression and compare two classes
of feature mappings: linear functions and fully-connected neural networks
with one hidden layer that has 100 nodes. The learning curves in Figure 1
shows the advantage of neural network and demonstrates the flexibility of
CURE with nonlinear function classes.

Achieving Bayes optimality in multi-class clustering is indeed very challeng-
ing. Under parametric models (e.g. Gaussian mixtures), one may construct

Figure 1: 4-class Fashion-MNIST.

suitable loss functions for CURE based on likelihood functions. We will comment on this in the discussion section.

**Numerical experiments**

A main motivation for this paper is to deal with stretched (elongated) clusters where directions with the largest
variabilities of data may not be informative for clustering at all. Instead, one should aim for directions onto which the
projected data exhibit cluster structures. This explains why we choose T-shirts/tops and Pullovers in the Fashion-MNIST
dataset for demonstration: the bulk of a image corresponds to the belly part of clothing with different grayscales, logos
and hence contributes to the most of variability. However, T-shirts and Pullovers are distinguished by sleeves. Hence
the two classes can be separated by a linear function that is not related to the leading principle component of data.

We conducted new experiments comparing CURE with more algorithms: [1] Model-based clustering (Mclust) in Fraley
and Raftery (1999); [2] Projection Pursuit (PP) in Peña and Prieto (2001); [3] alternations between linear discriminant
analysis and K-means (LDA + Kmeans) in Ding and Li (2007); [4] Minimum Density Hyperplane (MDH) in Pavlidis
et al. (2016). [1], [2] and [4] are implemented using open-source R packages with default settings. While there
is no implementation of [3] available publicly, we did it ourselves following the instructions in the paper. Table 1
shows the results. For randomized algorithms we do 50 independen runs and report means and standard deviations of
misclassification rates. Again, CURE outperforms all the competitors. On a Macbook Pro it takes less than 10 seconds
to converge while others usually require one minute or more. Currently, we are also exploring other datasets.

Table 1: Misclassification rates of CURE and other methods.

| $N_1 : N_2$ / Method | $1 : 1$ | $2 : 1$ | $3 : 1$ | $4 : 1$ |
|---|---|---|---|---|
| CURE (ours) | $5.2 \pm 0.2\%$ | $7.1 \pm 0.4\%$ | $9.3 \pm 0.7\%$ | $11.3 \pm 1.1\%$ |
| Mclust [1] | $48.7 \pm 1.3\%$ | $39.1 \pm 4.8\%$ | $34.1 \pm 8.0\%$ | $28.2 \pm 7.8\%$ |
| Projection Pursuit [2] | $36.9 \pm 9.8\%$ | $37.4 \pm 9.6\%$ | $39.7 \pm 6.9\%$ | $40.6 \pm 7.3\%$ |
| LDA + Kmeans [3] | $45.9\%$ | $49.0\%$ | $45.6\%$ | $44.3\%$ |
| MDH [4] | $48.6\%$ | $43.1\%$ | $38.3\%$ | $35.2\%$ |

It is worth pointing out that CURE learns a classification rule that readily predicts labels for any new data. This is an
advantage over many existing approaches for clustering and embedding, including spectral methods and t-SNE where
out-of-sample extensions are not so straightforward. We will highlight this in the paper.

**Other issues.** For the balanced two-class problem we have an explicit construction of $f$ and set the regularization
parameter $\lambda$ to be 1. Our theory goes through as long as $\lambda \geq 1$, and our experimental results are not sensitive to the
choice of $\lambda$. So we choose $\lambda = 1$ to reduce the need of tuning and simplify statements of theoretical results.

[Meta-Review · NeurIPS 2020]

All four reviewers support acceptance for the contributions, notably for the theoretical aspects of the proposed approach and for the overall methodology being useful and novel; I also recommend acceptance. However, please consider revising your paper to incorporate additional references and to address reviewers R1, R2 and R4's concerns that the comparison with other works is rather simplistic. More experimental support (in e.g. Supplementary Materials) would also strengthen the paper.